# Quantifying how post-transcriptional noise and gene copy number variation bias transcriptional parameter inference from mRNA distributions

Xiaoming Fu[1,2,3†], Heta P Patel[4†], Stefano Coppola[4], Libin Xu[1], Zhixing Cao[1]*, Tineke L Lenstra[4]*, Ramon Grima[2]*

[1]Key Laboratory of Smart Manufacturing in Energy Chemical Process, East China University of Science and Technology, Shanghai, China; [2]School of Biological Sciences, University of Edinburgh, Edinburgh, United Kingdom; [3]Center for Advanced Systems Understanding, Helmholtz-Zentrum Dresden-Rossendorf, Görlitz, Germany; [4]The Netherlands Cancer Institute, Oncode Institute, Division of Gene Regulation, Amsterdam, Netherlands

*For correspondence:
zcao@ecust.edu.cn (ZC);
t.lenstra@nki.nl (TLL);
ramon.grima@ed.ac.uk (RG)

[†]These authors contributed equally to this work

Competing interest: The authors declare that no competing interests exist.

**Abstract** Transcriptional rates are often estimated by fitting the distribution of mature mRNA numbers measured using smFISH (single molecule fluorescence *in situ* hybridization) with the distribution predicted by the telegraph model of gene expression, which defines two promoter states of activity and inactivity. However, fluctuations in mature mRNA numbers are strongly affected by processes downstream of transcription. In addition, the telegraph model assumes one gene copy but in experiments, cells may have two gene copies as cells replicate their genome during the cell cycle. While it is often presumed that post-transcriptional noise and gene copy number variation affect transcriptional parameter estimation, the size of the error introduced remains unclear. To address this issue, here we measure both mature and nascent mRNA distributions of *GAL10* in yeast cells using smFISH and classify each cell according to its cell cycle phase. We infer transcriptional parameters from mature and nascent mRNA distributions, with and without accounting for cell cycle phase and compare the results to live-cell transcription measurements of the same gene. We find that: (i) correcting for cell cycle dynamics decreases the promoter switching rates and the initiation rate, and increases the fraction of time spent in the active state, as well as the burst size; (ii) additional correction for post-transcriptional noise leads to further increases in the burst size and to a large reduction in the errors in parameter estimation. Furthermore, we outline how to correctly adjust for measurement noise in smFISH due to uncertainty in transcription site localisation when introns cannot be labelled. Simulations with parameters estimated from nascent smFISH data, which is corrected for cell cycle phases and measurement noise, leads to autocorrelation functions that agree with those obtained from live-cell imaging.

## Editor's evaluation

This important paper tackles a core problem in systems biology – how to quantify kinetic parameters from incomplete and noisy experimental data. The study makes a convincing methodological contribution to the field, and its potential usefulness is demonstrated using experimental data in yeast.

## Introduction

Transcription in single cells occurs in stochastic bursts (*Suter et al., 2011*; *Larsson et al., 2019*). Although the first observation of bursting occurred more than 40 years ago (*McKnight and Miller, 1977*), the precise mechanisms behind this phenomenon are still under active investigation (*Nicolas et al., 2017*; *Tunnacliffe and Chubb, 2020*). The direct measurement of the dynamic properties of bursting employs live-cell imaging approaches, which allow visualization of bursts as they occur in living cells (*Donovan et al., 2019*). However, in practice, such live-cell measurements are challenging because they are low-throughput and require genome-editing (*Brouwer et al., 2020*; *Lenstra and Larson, 2016*). To circumvent this, one can exploit the fact that bursting creates heterogeneity in a population. In this case, it is relatively straightforward to obtain a steady-state distribution of the number of mRNAs per cell from smFISH or single-cell sequencing experiments. These distributions have been used to infer dynamics by comparison to theoretical models. The simplest mathematical model describing bursting is the telegraph (or two-state) model (*Peccoud and Ycart, 1995*; *Raj et al., 2006*). In this model, promoters switch between an active and inactive state, where initiation occurs during the active promoter state. The model makes the further simplifying assumption that the gene copy number is one and that all the reactions are effectively first-order. The mRNA in this model can be interpreted as cellular (mature) mRNA since its removal via various decay pathways in the cytoplasm is known to follow single-exponential (first-order) decay kinetics in eukaryotic cells (*Wang et al., 2002*; *Herzog et al., 2017*). The solution of the telegraph model for the steady-state distribution of mRNA numbers has been fitted to experimental mature mRNA number distributions to estimate the transcriptional parameters (*Raj et al., 2006*; *Kim and Marioni, 2013*; *Suter et al., 2011*; *Larsson et al., 2019*).

However, the reliability of the estimates of transcriptional parameters from mRNA distributions is questionable because the noise in mature mRNA (and consequently the shape of the mRNA distribution) is affected by a wide variety of factors. Recent extensions of the telegraph model have carefully investigated how mRNA fluctuations are influenced by the number of promoter states (*Zhou and Zhang, 2012*; *Ham et al., 2020b*), polymerase dynamics (*Cao et al., 2020*), cell-to-cell variability in the rate parameter values (*Dattani and Barahona, 2017*; *Ham et al., 2020a*), replication and binomial partitioning due to cell division (*Cao and Grima, 2020*), nuclear export (*Singh and Bokes, 2012*) and cell cycle duration variability (*Perez-Carrasco et al., 2020*). One way to avoid noise from various post-transcriptional sources is to measure distributions of nascent mRNA rather than mature mRNA, and then fit these to the distributions predicted by an appropriate mathematical model. A nascent mRNA (*Zenklusen et al., 2008*; *Larson et al., 2009*) is an mRNA that is being actively transcribed, that is it is still tethered to an RNA polymerase II (Pol II) moving along a gene during transcriptional elongation. Fluctuations in nascent mRNA numbers thus directly reflect the process of transcription. Because nascent mRNA removal is not first-order, an extension of the telegraph model has been developed (the delay telegraph model) (*Xu et al., 2016*).

However, nascent mRNA data still suffers from other sources of noise due to cell-to-cell variability. For example in an asynchronous population of dividing cells, cells can have either one or two gene copies. In the absence of a molecular mechanism that compensates for the increase in gene copy number upon replication, cells with two gene copies which cannot be spatially resolved will have a different distribution of nascent mRNA numbers (one with higher mean) than cells with one gene copy. The importance of the cell cycle is illustrated by the finding (*Zopf et al., 2013*) that noisy transcription from the synthetic TetO promoter in *S. cerevisiae* is dominated by its dependence on the cell cycle. The estimation of transcriptional parameters from nascent mRNA data for pre- and post-replication phases of the cell cycle has, to the best of our knowledge, only been reported in *Skinner et al., 2016*.

Interestingly, all the studies that estimate transcriptional parameters from nascent mRNA data (*Skinner et al., 2016*; *Xu et al., 2015*; *Zoller et al., 2018*; *Senecal et al., 2014*; *Fritzsch et al., 2018*) do not compare them with transcriptional parameters estimated from cellular (mature) mRNA data measured in the same experiment. Similarly, a quantitative comparison between inference from cell-cycle-specific data and data which contains information from all cell cycle phases is lacking. Likely, this is because it is considered evident that quantifying fluctuations earlier in the gene expression process and adjusted for the cell-cycle will improve estimates. However, nascent mRNA distributions are technically more challenging to acquire than mature mRNA distributions; and inference from nascent mRNA distributions is substantially more complex (*Xu et al., 2016*). Thus, it still needs to be shown

that acquiring nascent mRNA data is a necessity from a parameter inference point of view, i.e. that it leads to significantly different and more robust estimates than using mature mRNA data. We also note that current studies report parameter inference from organisms where it is possible to label introns to identify mRNA located at the transcription site. This is not possible in many yeast genes and other microorganisms, and in these cases it is unclear how to correct parameter estimates for uncertainty in the transcription site location.

In this paper, we sought to understand the precise impact of post-transcriptional noise and cell-to-cell variability on the accuracy of transcriptional parameters inferred from mature mRNA data. The fitting algorithms (for mature and nascent mRNA data) were first tested on simulated data, where limitations of the algorithms were uncovered in accurately estimating the transcriptional parameters in certain regions of parameter space. The algorithms were then applied to four independent experimental data sets, each measuring *GAL10* mature and nascent mRNA data from smFISH in galactose-induced budding yeast, conditional on the stage of the cell cycle (G1 or G2) for thousands of cells. Comparison of the transcriptional parameter estimates allowed us to separate the influence of ignoring cell cycle variability from that of post-transcriptional noise (mature vs nascent mRNA data). We found that only fitting of nascent cell-cycle data, corrected for measurement noise (due to uncertainty in the transcription site location), provided good agreement with measurements from live-cell data. Cell-cycle specific analysis also revealed that upon transition from G1 to G2, yeast cells show dosage compensation by reducing burst frequency, similar to mammalian cells (*Padovan-Merhar et al., 2015*). Our systematic comparison highlights the challenges of obtaining kinetic information from static data, and provides insight into potential biases when inferring transcriptional parameters from smFISH distributions.

## Results
### Inference from mature mRNA data vs inference from nascent mRNA data: testing inference accuracy using synthetic data

To understand the accuracy of the inference algorithms from nascent and mature mRNA data, in various regions of parameter space, (i) we generated synthetic data using stochastic simulations with certain known values of the parameters; (ii) applied the inference algorithms to estimate the parameters from the synthetic data; (iii) compared the true and inferred kinetic parameter values.

The generation of synthetic mature mRNA data (mature mRNA measurements in each of $10^4$ cells) using stochastic simulations of the telegraph model (*Figure 1a*) is described in Methods Sections Mathematical model and Generation of synthetic mature mRNA data. The inference algorithm is described in detail in Methods Section Steps of the algorithm to estimate parameters from mature mRNA data. It is based on a maximization of the likelihood of observing the single cell mature mRNA numbers measured in a population of cells. The likelihood of observing a certain number of mature mRNA numbers from a given cell is given by evaluating the telegraph model's steady-state mature mRNA count probability distribution.

For nascent RNA data, we used stochastic simulations of the delay telegraph model (*Figure 1b*) to generate the position of bound Pol II molecules from which we constructed the synthetic smFISH signal in each of $10^4$ cells (Methods Section Generation of synthetic nascent mRNA data). An inference algorithm estimates the parameters, based on a maximization of the likelihood of observing the single cell total fluorescence intensity measured in a population of cells (Methods Section Steps of the algorithm to estimate parameters from nascent mRNA data). Note that the likelihood of observing a certain fluorescence signal intensity from a cell is given by extension of the delay telegraph model (but not directly by the delay telegraph model itself) to account for the smFISH probe positions.

This extension takes into account that the experimental fluorescence data used in this manuscript was acquired from smFISH of PP7-*GAL10* in budding yeast, where probes were hybridized to the PP7 sequences. Because the PP7 sequences are positioned at the 5' of the *GAL10* gene, the fluorescence intensity of a single mRNA on the DNA locus resembles a trapezoidal pulse (see *Figure 1* for an illustration). As the Pol II molecule travels through the 14 repeats of the PP7 loops, the fluorescence intensity increases as the fluorescent probes binds to the nascent mRNA (this is the linear part of the trapezoidal pulse). However, once all 14 loops on the nascent mRNA are bound by the fluorescent probes, the intensity of a single mRNA reaches maximal intensity and the plot plateaus as the RNA

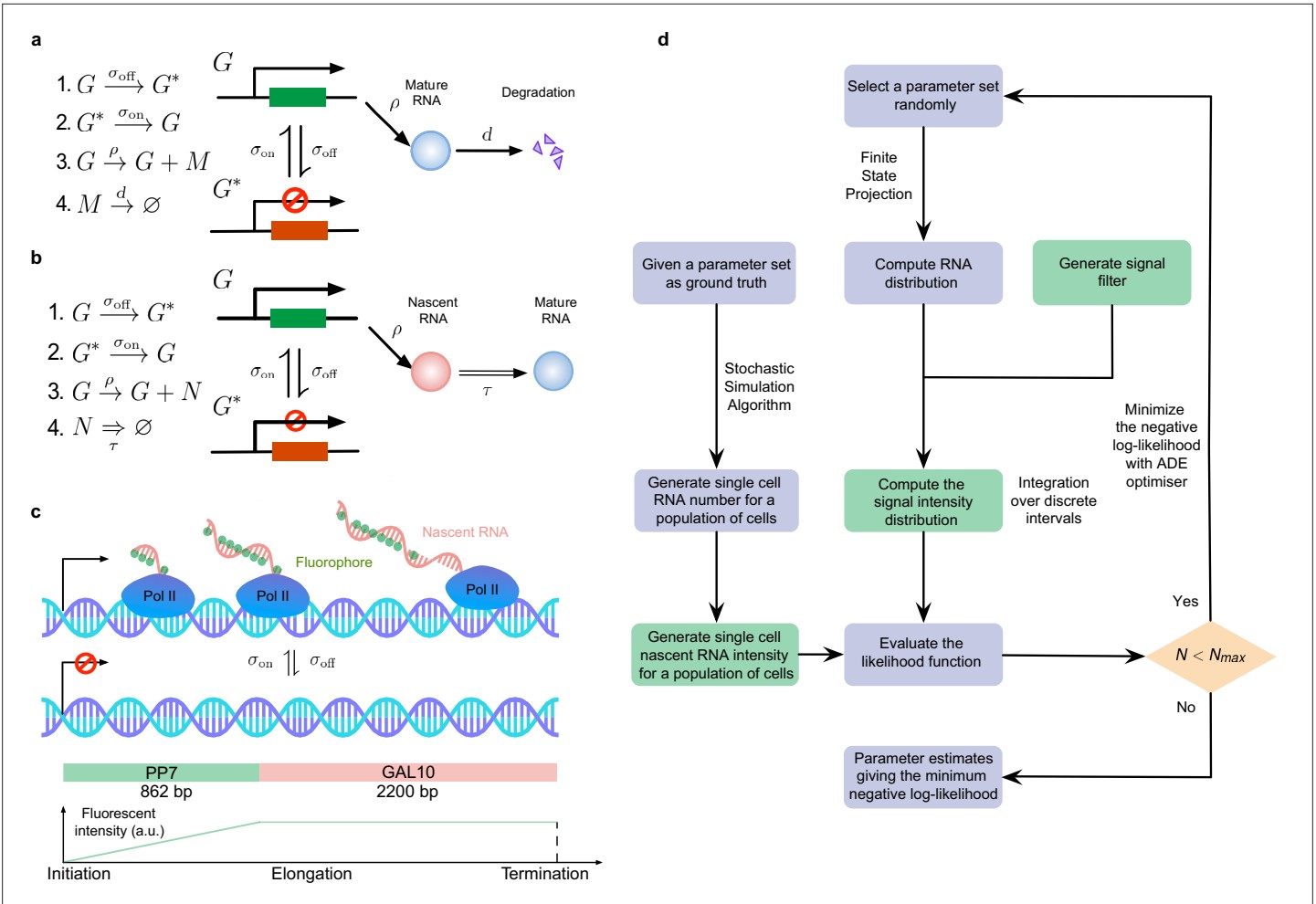

**Figure 1.** Overview of the inference of transcriptional parameters from synthetic data. (**a**) A schematic illustration of the telegraph model. (**b**). A schematic of the delay telegraph model. The double horizontal line for nascent mRNA removal indicates this is a delayed reaction. (**c**) Illustration showing promoter switching between two states, Pol II binding to the promoter in the ON state and subsequently undergoing productive elongation. Note that the length of the nascent mRNA tail increases until Pol II terminates at the end of the gene. As Pol II travels through the 14 repeats of the PP7 loops, the intensity of the mRNA increases due to fluorescent probe binding to the mRNA; intensity saturates as Pol II enters the *GAL10* gene body. (**d**) Illustration of the algorithms to generate synthetic data and to perform inference from mature and nascent mRNA data. The green boxes are only applicable for the inference of the fluorescence signal intensity of nascent mRNAs; note that in nascent mRNA inference, the "RNA number" in the flow chart should be interpreted as the number of bound Pol II molecules on the gene. A large iteration step $N_{\max}$ ($\geq 10^4$) is chosen as the termination condition for the optimizer.

elongates through the *GAL10* gene body before termination and release. The total fluorescent signal density function is hence given by

$$p(s; \theta) = \sum_{k=0}^{\infty} p(s|k)P(k; \theta), \tag{1}$$

where $p(s|k)$ is the density function of the signal $s$ given there are $k$ bound Pol II molecules and $P(k; \theta)$ is the steady-state solution of the delay telegraph model giving the probability of observing $k$ bound Pol II molecules for the parameter set $\theta$. In Methods Section Mathematical model, we show how $p(s|k)$ can be approximately calculated for the trapezoidal pulse. Hence *Equation (1)* represents the extension of the delay telegraph model to predict the smFISH fluorescent signal of the transcription site. Note that both of these inference algorithms were used to infer the promoter switching and initiation rate parameters. The degradation rate and the elongation time were not estimated but assumed to be known. The inference and synthetic data generation procedures are summarised and illustrated in *Figure 1d*.

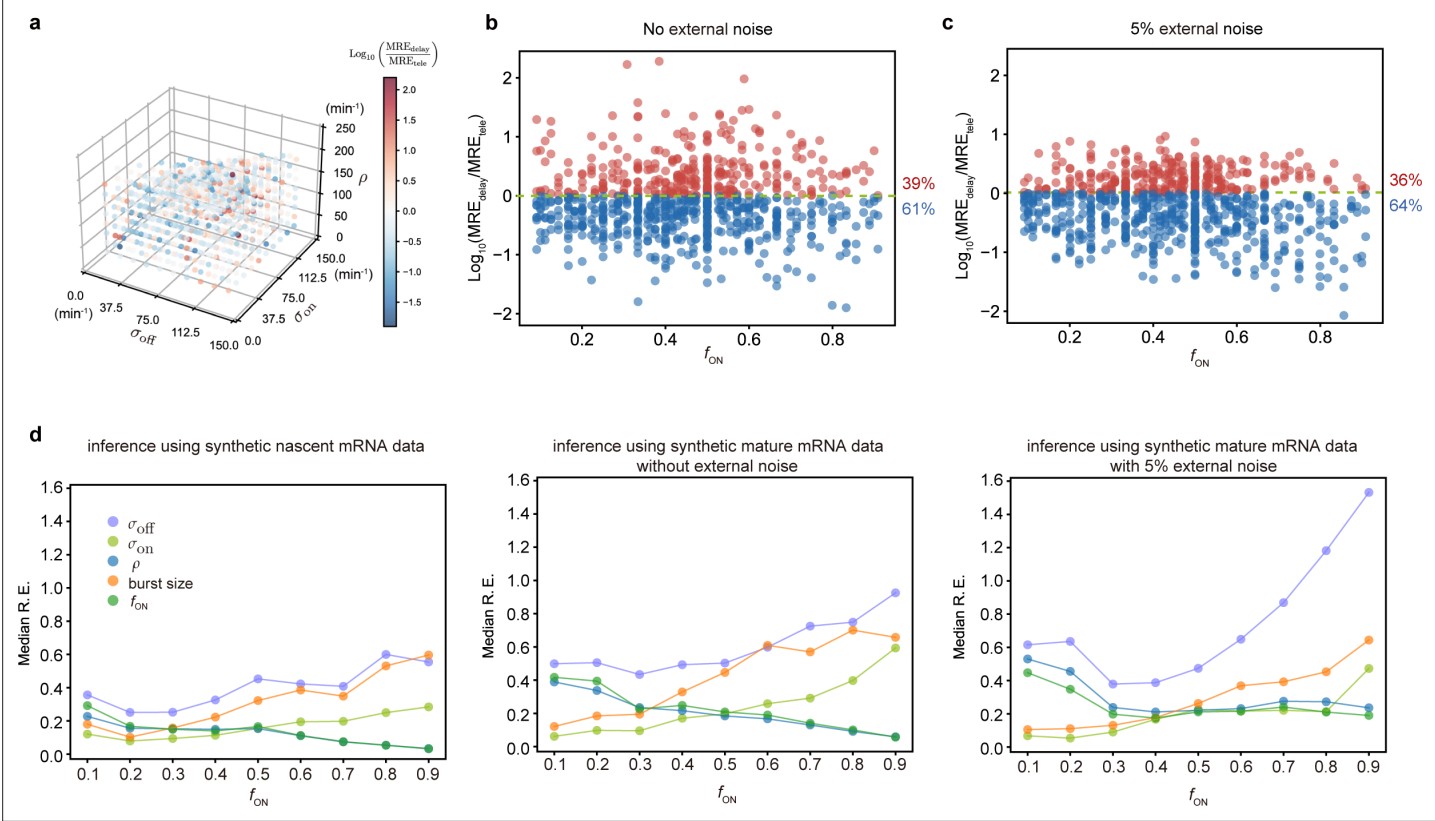

**Figure 2.** Accuracy of the inferred kinetic parameters from synthetic mature and nascent mRNA data using the telegraph and delay telegraph model, respectively. (**a**) 3D scatter plot showing the ratio of the mean relative error from nascent mRNA data (using delay telegraph model $\mathrm{MRE_{delay}}$) and the mean relative error from mature mRNA data (using the telegraph model $\mathrm{MRE_{tele}}$) for 789 independent parameter sets sampled on a grid. Red data points indicate parameter sets with lower relative errors for mature data compared to nascent data, blue datapoints indicate parameter sets with lower relative error for nascent data compared to mature data (**b**) Same data as (**a**) but shown as a function of the fraction of ON time, $f_{\mathrm{ON}}$. For $\approx 61\%$ of the parameters, the inference accuracy is higher when using nascent mRNA data. (**c**) Sampling from the same parameter space, we then add log-normal distributed noise (size 5%) to the initiation rate $\rho$ (see text for details) to mimic external noise due to post-transcriptional processing that is only present in mature mRNA. $\mathrm{Log_{10}}$ of the ratio of the median relative error (MRE) using perturbed mature mRNA data against $\mathrm{Log_{10}}$ MRE using nascent mRNA data is shown as a function of the true fraction of ON time, $f_{\mathrm{ON}}$. For $\approx 64\%$ of the parameters, the inference accuracy is higher when using nascent mRNA data. (**d**) The median relative error of each transcriptional parameter as a function of the fraction of ON time, using synthetic nascent mRNA, synthetic mature mRNA data and synthetic mature mRNA with external noise. Inference from nascent data is generally more accurate than using mature mRNA data.

The accuracy of inference was first calculated as the mean of the relative error in the estimated parameters $\sigma_{\mathrm{off}}$, $\sigma_{\mathrm{on}}$, and $\rho$ (for its definition see Methods, Equation (6)); note that this error measures deviations from the known ground truth values. *Figure 2a* shows, by means of a 3D scatter plot, the ratio of the mean relative error from nascent mRNA data (using delay telegraph model) and the mean relative error from mature mRNA data (using the telegraph model) for 789 independent parameter sets sampled on a grid (for each of these sets, we simulated $10^4$ cells). The overall bluish hue of the plot suggested that the mean relative error from nascent mRNA data was typically less than the error from mature mRNA data. This was confirmed in *Figure 2b* where the same data was plotted but now as a function of the fraction of ON time (defined as $f_{\mathrm{ON}} = \sigma_{\mathrm{on}}/(\sigma_{\mathrm{off}} + \sigma_{\mathrm{on}})$). Out of 789 parameter sets, for 483 of them ($\approx 61\%$) the inference accuracy was higher when using nascent mRNA data.

Thus far, we have implicitly assumed that fluctuations in both nascent and mature mRNA are due to transcriptional bursting. However, it is clear that mature mRNA data exhibit a higher degree of noise due to post-transcriptional processing. For example, it has been shown that transcriptional noise is typically amplified during mRNA nuclear export (*Hansen et al., 2018*). In addition, cell-to-cell variation in the number of nuclear pore complexes has recently been identified as the source of heterogeneity in nuclear export rates within isogenic yeast populations (*Durrieu et al., 2022*). To take into

account these additional noise sources, which we call external noise, we added noise to the initiation rate $\rho$ in the telegraph model since this rate implicitly models all processes between the synthesis of the transcript and the appearance of mature mRNA in the cytoplasm. Specifically, for each of the 789 parameter sets previously used, we changed $\rho$ to $\rho'$ where the latter is a log-normal distributed random variable such that its mean is $\rho$ and its standard deviation is equal to 0.05 of the mean (5% external noise). Note that this implies that at the time of measurement, each cell in the population had a different value of the initiation rate. Simulations with this perturbed set of parameters led to a synthetic mature mRNA data set from which we re-inferred parameters using the telegraph model. In *Figure 2c* we show the ratio of mean relative error from nascent mRNA data and the mean relative error from perturbed mature mRNA data as a function of the fraction of ON time, $f_{ON}$. The percentage of parameters where nascent mRNA is more accurate is slightly increased compared to the data without noise (64% versus 61% of the parameters; compare *Figure 2c* and *Figure 2b*). However, the addition of even more noise (10% external noise added to the initiation rate) increases the inference accuracy for 91% of the parameter sets when the nascent mRNA data is used (Appendix 1 and *Appendix 1—figure 1*).

To obtain more insight into the accuracy of the individual parameters, we next plotted the median relative error of transcriptional parameters $\sigma_{off}, \sigma_{on}, \rho$, burst size and the inferred fraction of ON time, as a function of the true fraction of ON time (*Figure 2d*). We compared the results using synthetic nascent mRNA, synthetic mature mRNA data and synthetic mature mRNA with 5% external noise. The median of the relative error for each transcriptional parameter (as given by the second equation of Equation 8) was obtained for the subset of the 789 parameter sets for which the true fraction of ON time $f_{ON}$ falls into the interval $[x - 0.05, x + 0.05]$ where $x = 0.1, 0.2, \ldots, 0.9$. From the plots, the following can be deduced: (i) the errors in $\sigma_{on}$ (the burst frequency), $\sigma_{off}$ and the burst size $\rho/\sigma_{off}$ tend to increase with $f_{ON}$ while the rest of the parameters ($\rho$ and the estimated value of $f_{ON}$) decrease; (ii) for small $f_{ON}$, the best estimated parameters are the burst frequency and size while for large $f_{ON}$, it was $\rho$ and the estimated value of $f_{ON}$. The worst estimated parameter was $\sigma_{off}$, independent of the value of $f_{ON}$; (iii) the addition of external noise to mature mRNA data had a small impact on inference for small $f_{ON}$; in contrast, for large $f_{ON}$ the noise appreciably increased the relative error in $\sigma_{off}$ and to a lesser extent the error in the other parameters too.

Additionally, in Appendices 1 and 2 we show that (i) independent of the accuracy of parameter estimation, the best fit distributions accurately matched the ground truth distributions (Appendix 1 and *Appendix 1—figure 2*); (ii) the parameters ordered by relative error were in agreement with the parameters ordered by sample variability (Appendix 1 and *Appendix 1—table 1*) and by profile likelihood error (*Kreutz et al., 2013*) (Appendix 1, *Appendix 1—tables 2 and 3*). Since from experimental data, only the sample variability and the profile likelihood error are available, it follows that the results of our synthetic data study in *Figure 2* based on relative error from the ground truth have wide practical applicability; (iii) stochastic perturbation of the mature or nascent mRNA data (due to errors in the measurement of the number of spots and the fluorescent intensity) had little effect on the inference quality, unless the gene spent a large proportion of time in the OFF state (*Appendix 1—tables 4 and 5*); (iv) if one utilized the conventional telegraph model to fit the nascent data generated by the delay telegraph model, it was possible to obtain a distribution fitting as good as the delay telegraph model but with low-fidelity parameter estimation (Appendix 2, *Appendix 2—figure 1* and *Appendix 2—table 1*). Analytically, the telegraph model is only an accurate approximation of the delay telegraph model when the promoter switching timescales are much longer than the time spent by Pol II on a gene or the off switching rates are very small such that gene expression is nearly constitutive.

In summary, by means of synthetic experiments, we have clarified how the accuracy of the parameter inference strongly depends on the type of data (nascent or mature mRNA) and the fraction of time spent in the ON state (which determines the mode of gene expression).

## Applications to experimental yeast mRNA data

Now that we have introduced the inference algorithms and tested them thoroughly using synthetic data, we applied the algorithms to experimental data (see Method Section Experimental data acquisition and processing for details of the data acquisition). Note that in what follows, delay telegraph model refers to the extended delay telegraph model that accounts for the smFISH probe positions that was used to predict the smFISH fluorescent signal of the transcription site.

## Inference from mature mRNA data: experimental artifacts

We have four independent datasets from which we determined mRNA count and nascent RNA distributions. *Figure 3a* shows an example cell with mature single RNAs in the cytoplasm, and a bright nuclear spot representing the site of nascent transcription. Spots and cell outlines were identified using automated pipelines. Importantly, to obtain an accurate estimation of transcriptional parameters, the experimental input distributions of mRNA count and nascent RNAs require high accuracy. We therefore first determined how technical artifacts in the analysis affects the inference estimates.

First, if the number of mRNA transcripts per cell is high, accurate determination of the number of transcripts may be challenging, as transcripts may overlap. To determine if this occurred in our datasets, we analyzed the distributions of intensities of the cytoplasmic spots, which revealed unimodal distributions where ~90% of the detected spots fell in the range 0.5× median – 1.5× median (*Figure 4a*). We therefore concluded that overlapping spots are not a large confounder in our data. In fact, in our experiments, the number of detected mature mRNA transcripts per cell was lower than expected, based on the number of nascent transcripts (compare *Figure 3* with *Figure 4*). This discrepancy between nascent and mature transcripts likely arises because the addition of the PP7 loops to the *GAL10* RNA destabilizes the RNA, resulting in faster mRNA turnover compared to most endogenous RNAs (*Miller et al., 2011*; *Wang et al., 2002*; *Holstege et al., 1998*; *Geisberg et al., 2014*). Previously, both shorter and longer mRNA half-lives from the addition of stem loops have been observed, which may be caused because changes in the 5' UTR length or sequence affect its recognition by the mRNA degradation machinery (*Heinrich et al., 2017*; *Tutucci et al., 2018*; *Garcia and Parker, 2015*). In our case, we note that such high turnover should aid transcriptional parameter estimates, as it closely reflects transcriptional activity.

A second possible source of error is cell segmentation. To test how cell segmentation errors contribute to the mature mRNA distribution and the transcriptional bursting estimates, we compared two independent segmentation tools, where segmentation 1 often resulted in missed spots (*Figure 3b*), resulting in an underestimation of the mean mRNA count and of the variance (compare *Figure 3b and c*). We inferred the transcriptional parameters using the algorithm described in Methods Section Steps of the algorithm to estimate parameters from mature mRNA data. In the absence of an experimental measurement of the degradation rate, we could only estimate the three transcriptional parameters normalised by $d$. The best fits of dataset 1 are shown in (*Figure 3b and c*) and the transcriptional parameters (for all four datasets) are summarized in (*Figure 3e*). Note that the estimated parameters for all four datasets, using both segmentations, are shown in *Appendix 3—table 1* and the associated best fit distributions in *Appendix 3—figure 1a*. Notably the segmentation algorithms led to similar estimates for the burst frequency but considerably different estimates for the rest of the parameters. In particular segmentation 1 suggested that burst expression is infrequent (≈20% of the time) whereas segmentation 2 was consistent with burst expression occurring half of the time. Given that accurate cell segmentation remains challenging, this analysis illustrates that parameter estimation from mature mRNA counts may be affected by technical errors. For the remainder of the mature mRNA analysis, we have used only segmentation 2 data.

Lastly, it may be challenging to distinguish the nascent transcription site from a mature RNA, especially if few nascent RNAs are being produced. Either one can decide to include all cellular spots in the total mRNA count, including the transcription site, with the result that the number of mature transcripts is overestimated with one RNA for cells which show a transcription site. Or conversely, one can decide to exclude the transcription site by subtracting one spot from each cell, with the result that the number of mature mRNAs may be underestimated by one RNA for cells that are transcriptionally silent. To understand how this choice affects the accuracy of parameter inference, we compared both options in (*Figure 3c, d and e*), where seg2 included all spots, and seg2-TS excluded transcription sites (by subtracting 1 from each cell). The estimated parameters for all four datasets are shown in *Appendix 3—table 1* and the associated best fit distributions in *Appendix 3—figure 1a*. Although the mean was lower when transcription sites were excluded, all the parameters except the burst frequency $\sigma_{on}$ were within the error, indicating that the choice of whether or not to include the transcription site in the mature mRNA count had a small influence on parameter estimation. For the remainder of the analysis, we included all spots, and counted the transcription site as one RNA.

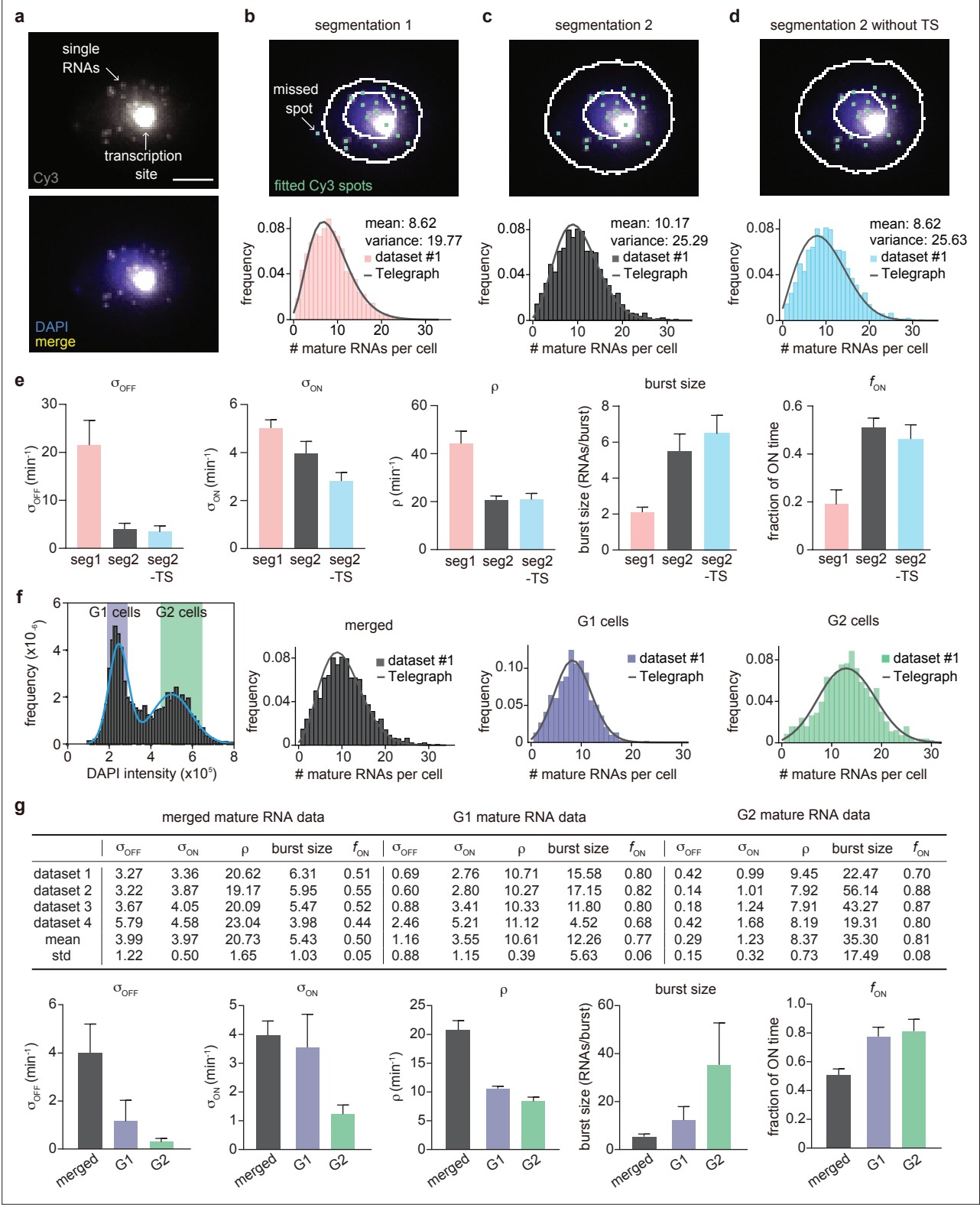

**Figure 3.** Inference results using four mature mRNA data sets with sample sizes of 2333, 6366, 4550 and 3163 cells, respectively. (**a**) Representative smFISH image of a yeast cell with *PP7-GAL10* RNAs labeled with Cy3 and the nucleus labeled with DAPI. (**b**) The DAPI and Cy3 signals were used to determine the nuclear and cellular mask, respectively. Detected and fitted spots are indicated in green. Mature RNA count distribution (pink) for segmentation method 1 with a best fit obtained from the telegraph model (gray curve). Scale bar is 5 μm(**c-d**) The DAPI and Cy3 signals were used to determine the nuclear and cellular mask using a second independent segmentation tool (segmentation 2). Mature RNA count distribution (gray and

*Figure 3 continued on next page*

*Figure 3 continued*

cyan) with/without counting the transcription site (TS) for segmentation method 2 with a best fit obtained from the telegraph model (gray curves). (**e**) Bar graphs of inferred transcriptional parameters (merged mature RNA data) from fitting the distributions of the two segmentation methods ('seg1' and 'seg2') as well as the distribution of mature RNAs only ('seg2 -TS' which indicates the exclusion of one spot in each cell that represents the transcription site). The burst size was computed as $\rho/\sigma_{off}$ and the fraction of ON time as $\sigma_{on}/(\sigma_{on} + \sigma_{off})$. Error bars indicate standard deviation computed over the four datasets. (**f**) Distribution of the integrated DAPI intensity for each cell. Cyan line represents a Gaussian bimodal fit with highlighted regions indicating the intensity-based classification of G1 and G2 cells. Distributions of the mature RNA count for all cells (merged) and cell-cycle classified cells (G1 cells and G2 cells). (**g**) Tables and bar graphs of inferred parameters for merged and cell-cycle-specific data. Note that the transcriptional parameters $\sigma_{on}, \sigma_{off}, \rho$ are normalised by the degradation rate and hence dimensionless. For the cell-cycle-specific data, parameters were inferred per gene copy.

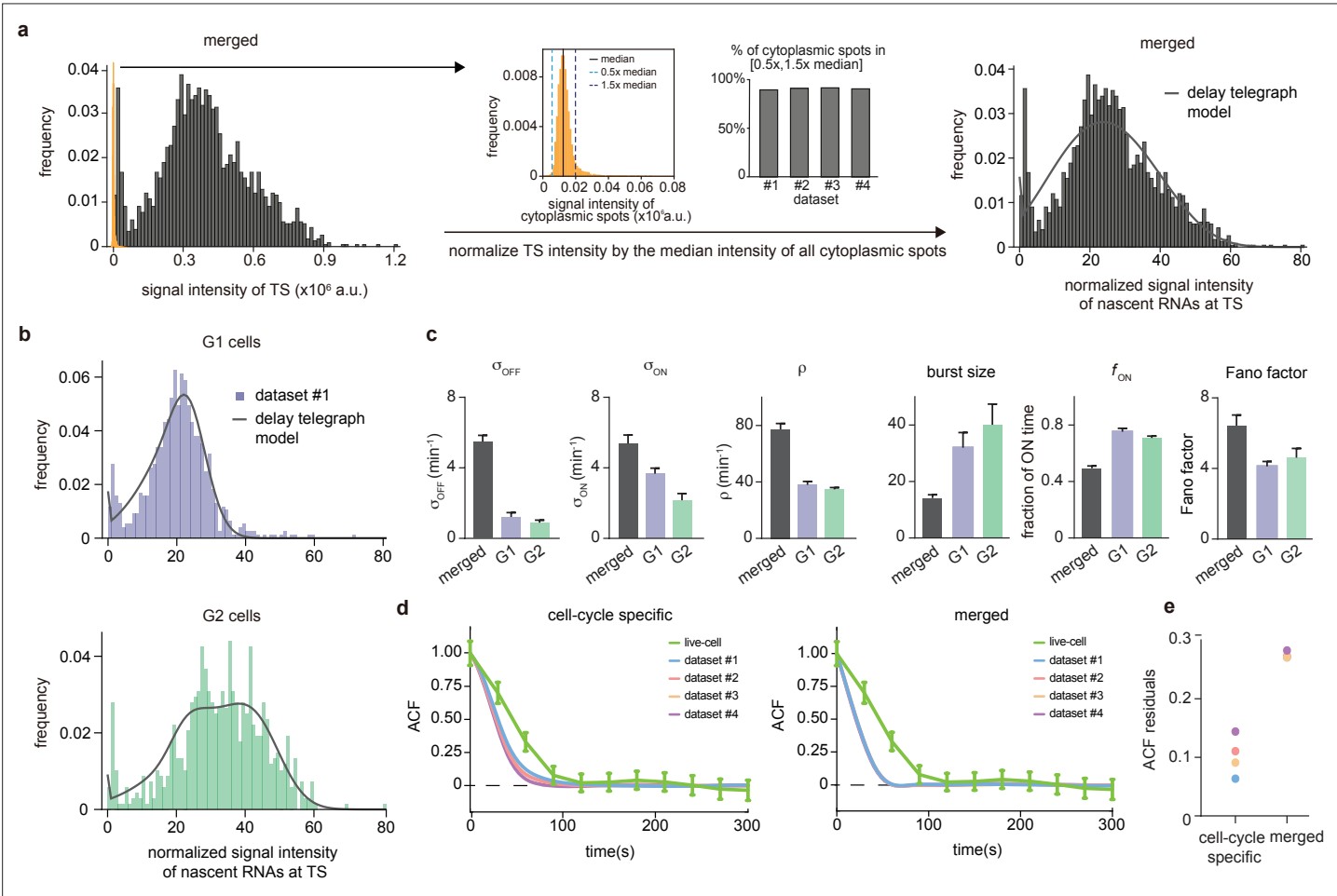

**Figure 4.** Inference from the normalized nascent mRNA distributions for merged and cell-cycle specific data. (**a**) Normalized nascent mRNA distributions of merged cell-cycle data were obtained by normalizing the signal intensity of the transcription site (defined as the brightest spot in the cell) by the median signal intensity of the cytoplasmic spots (shown in orange and zoom-in depicted in the inset). In all 4 datasets, approximately 90% of the detected cytoplasmic spots fell in the range 0.5× median – 1.5× median (grey bargraph). Black line in normalized distribution on the right represents best fit with delay telegraph model. (**b**) Nascent RNA distributions for cell-cycle-specific data. Black lines represent best fits with delay telegraph model. (**c**) Bar graphs comparing the transcriptional parameters, burst size, fraction of ON time and Fano factor for cell-cycle-specific and merged data. Error bars indicate standard deviation of the four datasets. (**d**) Normalized ACF plots of cell-cycle-specific and merged data. The ACF plots are generated by stochastic simulations using estimated parameters from merged and cell-cycle specific nascent mRNA data for each of the four data sets; these were compared with the ACF measured directly using live-cell data in ***Donovan et al., 2019*** (green line). (**e**) The sum of squared ACF residuals of merged and cell-cycle-specific data from each dataset (this is the sum of squared deviations between the measured and estimated normalised ACF where the sum was calculated over all time points).

## Inference from mature mRNA data: merged versus cell-cycle specific

The above analysis was performed using the merged data from all cells, irrespective of their position in the cell cycle. The inferred parameters of all four datasets are shown in *Figure 3g* (grey). To understand the effect of the cell cycle on these parameter estimates, we compared this inference with cell-cycle-specific data. We used the integrated nuclear DAPI intensity as a measure for DNA content to classify cells into G1 or G2 cells (*Figure 3f* (left)) to obtain separate mature mRNA distributions for G1 and G2 cells.

To infer the transcriptional parameters from mature mRNA data of cells in G1, the inference protocol remained the same. However for cells in the G2 stage, this protocol needed to be altered since G2 cells have two gene copies, whereas the solution of the telegraph model assumes one gene copy. Assuming the transcriptional activities of the two gene copies are independent, the distribution of the total molecule number is the convolution of the molecule number (obtained from the telegraph model) with itself for mature mRNA data. This convolved distribution was used in steps (ii) and (iii) of the inference algorithm in Methods Section Steps of the algorithm to estimate parameters from mature mRNA data. A difference between our method of estimating parameters in G2 from that in the literature (*Skinner et al., 2016*) is that we do not assume that the burst frequency is the only parameter that changes upon replication, and we estimated all transcription parameters simultaneously.

Note that the independence of gene copy transcription has been verified for genes in some eukaryotic cells (*Skinner et al., 2016*) where the two copies can be easily resolved. For yeast data, as we are analyzing in this paper, it is generally not possible to resolve the two copies of the allele in G2 because they are within the diffraction limit. However, in the absence of experimental evidence, the independence assumption is the simplest reasonable assumption that we could make (see later for a relaxation of this assumption).

For both G1 and G2 cells, we performed inference for cell-cycle specific mature mRNA data, the results of which are shown in *Figure 3f* (centre and right) and *Figure 3g* – see *Appendix 3—table 2* for the confidence intervals of the estimates calculated using profile likelihood. As expected, the mean number of mRNAs in G2 cells was larger than that in G1 cells. For both merged and cell-cycle specific data, the parameters ordered by increasing variability of the estimates from independent samples (the standard deviation divided by the mean) were: $\rho$, $f_{ON}$, $\sigma_{ON}$, burst size and $\sigma_{OFF}$, and the same order was predicted by the relative error (from ground truth values) from our synthetic experiments (compare with $f_{ON} = 0.50$ and $f_{ON} = 0.80$ in the middle and right panels of *Figure 2d*) and by sample variability (Appendix 1). In Appendix 3 and *Appendix 3—table 3* we show that the relaxation of the assumption of independence between the allele copies in G2 (by instead assuming perfect state correlation of the two alleles) had practically no influence on the inference of the two best estimated parameters ($\rho$, $f_{ON}$).

A comparison of the two types of data predicted different behaviour (*Figure 3g* bottom): merged data indicated behaviour consistent with the gene being ON half of the time and small burst sizes, while cell-cycle-specific data implied the gene is ON ≈80% of the time with large burst sizes. We note that the burst sizes have considerable sample variability, exemplifying burst size estimates of transcriptional parameters from mature mRNA distributions have to be treated with caution. Nevertheless, in line with this high fraction ON and large burst size, which start to approach constitutive expression, the variation introduced by the transcription kinetics is relatively modest with Fano factors not far from one: $2.43 \pm 0.21$ for merged data and $1.75 \pm 0.45$ for cell-cycle data (the slightly higher value for merged data likely was due to heterogeneity stemming from varying gene copy numbers per cell).

Comparing the mean rates between the G1 and G2 phases, we found that $\sigma_{off}$, $\sigma_{on}$, $\rho$ decreased while $f_{ON}$ and the burst size increased upon replication. However, taking into account the variability in estimates across the four datasets, the only two parameters which were well-separated between the two phases were $\sigma_{on}$ and $\rho$. These two parameters decreased by 65% and 21%, respectively, which suggests that upon replication, there are mechanisms at play which reduce the expression of each copy to partially compensate for the doubling of the gene copy number (gene dosage compensation) (*Skinner et al., 2016*).

In conclusion, what is particularly surprising in our analysis is the differences in the inference results using merged and cell-cycle specific data: the former suggests the gene spends only half of its time in the ON state while the latter implies the gene is mostly in its ON state.

# Inference from nascent mRNA data: cell cycle effects, experimental artifacts and comparison with mature mRNA inference

## Cell-cycle-specific versus merged data

To determine the number of nascent transcripts at the transcription site, we selected the brightest spot from each nucleus and normalized its intensity to the median intensity of the cytoplasmic spots. As the distribution of intensities of the cytoplasmic mRNAs followed a narrow unimodal distribution, its median likely represents the intensity of a single RNA (orange distribution in the central panel of *Figure 4a*). The inference of transcriptional parameters using the merged data was done using the algorithm described in Methods Section Steps of the algorithm to estimate parameters from nascent mRNA data.

Similar to above, to account for two gene copies in G2 cells, we assumed that the transcriptional activities of the two gene copies are independent. The distribution of the total fluorescent signal from both gene copies was the convolution of the signal distribution (obtained from the extended delay telegraph model, i.e. *Equation (1)*) with itself. This convolved distribution was then used in steps (ii) and (iii) of the inference algorithm.

The inference of transcriptional parameters from nascent RNA data was done using a fixed elongation time, which was measured previously at a related galactose-responsive gene (*GAL3*) at 65 bp/s (*Donovan et al., 2019*). Since the total transcript length is 3062 bp (see *Figure 1c*), the elongation time ($\tau$ in our model) is $\approx 47.1\,s \approx 0.785\,\min$. The fixed elongation rate enabled us to infer the absolute values of the three transcriptional parameters $\sigma_{\text{off}}, \sigma_{\text{on}}$ and $\rho$.

Best fits of the extended delay telegraph model to the distribution of signal intensity of nascent mRNAs at the transcription site are shown in *Figure 4a and b* for dataset 1; for the other datasets see *Appendix 4—figure 1*. The corresponding estimates of the transcriptional parameters are shown in *Appendix 4—table 1* and also illustrated by bar charts in *Figure 4c*. The confidence intervals of the transcriptional parameters (computed using the profile likelihood method) are shown in *Appendix 4—table 2*.

Comparing this estimation with that from mature mRNA, we observed that in both cases $f_{\text{ON}} \approx 0.5$ for merged data and in the range $0.7 - 0.8$ for cell-cycle-specific data. Also in both cases, the Fano factors of merged data were larger than those of cell-cycle-specific data. Hence, we are confident that not accounting for the cell cycle phase leads to an over-estimation of the time spent in the OFF state and of the Fano factor. In addition, comparing the burst sizes in *Figure 3g* and *Appendix 4—table 1*, we found that not taking into account post-transcriptional noise (by using mature mRNA data) led to an lower estimation of the burst size (2.6-fold, 2.6-fold, and 1.1-fold lower for inference from merged, G1 and G2 data, respectively). We note that it would be useful to directly compare the absolute estimates of the other transcriptional parameters from mature and nascent mRNA data. However, this was not possible because the telegraph model only estimates the switching rates and the initiation rate scaled by the degradation rate, and the latter is unknown. On the other hand, the estimates from nascent data were rates multiplied by the average elongation time, which is known and hence the absolute rates can be estimated from nascent mRNA data only. The only quantities that could be directly compared were the burst size and the fraction of ON time, since these are both non-dimensional.

Comparing the variability of the parameter estimates, we found that $\rho$ and $f_{\text{ON}}$ were the parameters with the smallest variability across samples for the nascent data, as for inference from mature data. However, the inferred parameter variability across samples was on average about 2.5-fold lower for nascent data compared to mature mRNA data (this was obtained by computing the standard deviation divided by the mean for each parameter and then averaging over all parameters and over merged, G1 and G2 data). Likely this is because nascent data does not suffer from post-transcriptional noise. Indeed, synthetic experiments suggested that the errors in parameter inference using nascent data are often less than those in mature data when $f_{\text{ON}} \approx 0.80$ (*Figure 2d*). In summary, we have more confidence in the parameter estimates from nascent data, in particular those from cell-cycle separated data.

To further investigate the hypothesis that estimates from cell-cycle-specific data are more accurate than merged data, we compared the estimates from merged and cell-cycle-specific data to previous live-cell transcription measurements of the same gene (*Donovan et al., 2019*). Because live-cell traces and simulated traces with the estimated transcriptional parameters are difficult to compare directly,

we instead compared their normalized autocorrelation functions (ACFs). Specifically, the live-cell traces displayed cell-to-cell variation in overall fluorescent intensities arising from differences in the PP7 coat protein expression level, precluding a direct comparison of the live-cell intensities with the smFISH distributions. The normalized ACFs are normalized per trace and thus can be used to directly compare the kinetics. For this, we fed the parameter estimates to the SSA to generate synthetic live-cell data and then calculated the corresponding ACF (Appendix 5). We found that the estimates from cell-cycle-specific data produced ACFs that match the live-cell data closer than that from the merged data (*Figure 4d*). This was also clear from the sum of squared residuals which for each dataset was smaller for the ACF computed using the cell-cycle-specific estimates rather than those from merged data (*Figure 4e*).

Using nascent data, we also reinvestigated the hypothesis that the gene exhibits dosage compensation. Comparing the mean rates between the G1 and G2 phases, we found that $\sigma_{\text{off}}$, $\sigma_{\text{on}}$, $\rho$, $f_{\text{ON}}$ decreased while the burst size increased upon replication. However, taking into account the variability in estimates across the four datasets, the only two parameters which were cleanly separated between the two phases were $\sigma_{\text{on}}$ and $f_{\text{ON}}$. These two decreased by 41% and 5%, respectively. These results had some similarity to those deduced from cell-cycle separated mature mRNA data (the decrease of $\sigma_{\text{on}}$) but they also displayed differences. Namely, from mature mRNA data it was predicted that $\rho$ decreased upon replication while from nascent data we predicted that $\rho$ did not change and it was rather $f_{\text{ON}}$ that decreased by a small degree. The decrease of the burst frequency $\sigma_{\text{on}}$ after replication has also been reported for some genes in mammalian cells (*Skinner et al., 2016*; *Padovan-Merhar et al., 2015*), indicating that this could be a general mechanism for gene dosage compensation. Our results are consistent with a population-based ChIP-seq study (*Voichek et al., 2016*) that showed DNA dosage compensation after replication in budding yeast. We note that our single-cell analysis only revealed partial dosage compensation, where the mean signal intensity of nascent mRNAs in G2 is not the same as in G1, but 1.7-fold higher in G2 than in G1 (*Figure 4c*).

## Correcting for experimental artefacts

Although inference on cell cycle separated data outperformed inference on merged data, we noticed that the corresponding best fit distributions did not match well to the experimental signal distributions in the lower bins (*Figure 4b* and *Appendix 4—figure 1*). In all cases, the experimental distributions showed high intensities in bins 1, 2, and 3, which was likely an artifact of the experimental data

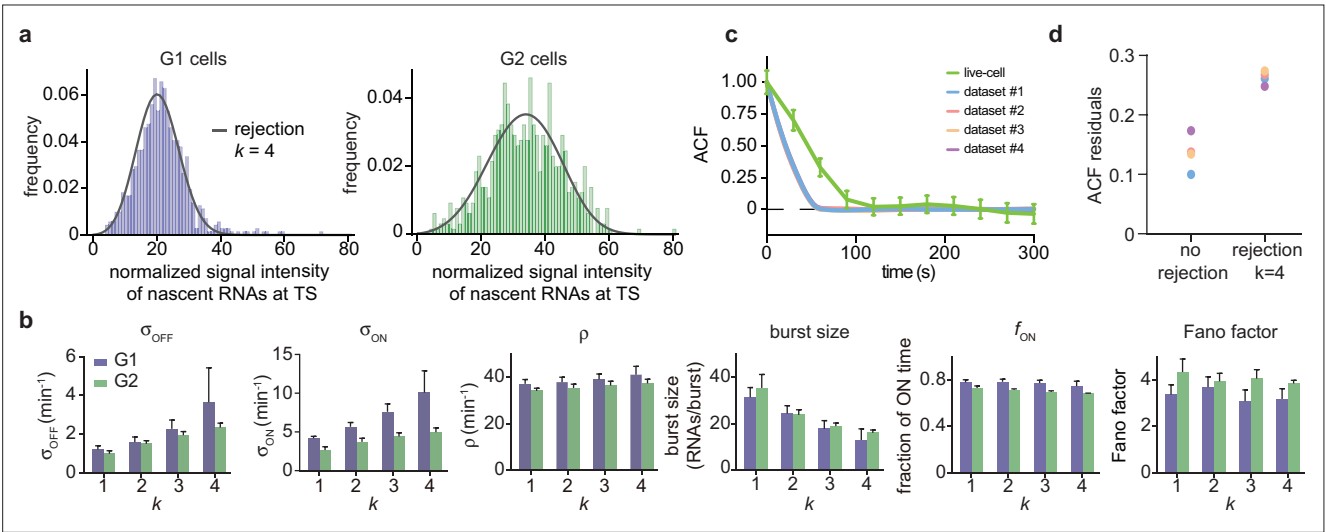

**Figure 5.** Inference results using the rejection method. (**a**) Nascent RNA distributions for cell-cycle-specific and merged data. Black lines represent best fits with delay telegraph model using the rejection method. (only the distributions for dataset #1 $k = 4$ are shown). (**b**) Estimated transcriptional parameters, burst size, fraction of ON time and Fano factor (mean values and standard deviation error bars of the four datasets) by rejecting the first $k$ bins with $k = 1, 2, 3, 4$. The estimated parameters are listed in *Appendix 4—table 3*. (**c**) Normalized autocorrelation function (ACF) predicted by stochastic simulations using the estimated parameters (for $k = 4$) for each of the four data sets versus that measured directly using live-cell data (green line). (**d**) The sum of squared residuals of the ACF of cell-cycle-specific data from each dataset without/with rejection when $k = 4$.

acquisition system. Since we defined the transcription site as the brightest spot, this implies that in the absence of a transcription site, a mature transcript can be misclassified as a nascent transcript. We therefore investigated two methods to correct for this, the 'rejection' method and the 'fusion' method.

The rejection method removed all data associated with the first $k$ bins of the experimentally obtained histogram of fluorescent intensities (*Figure 5a* shows the fits for dataset 1; for the other datasets see *Appendix 4—figure 2*). We found that the parameter estimates varied strongly when the number of bins from which data was rejected ($k$) was changed (*Figure 5b*; see also *Appendix 4— table 3*). Although the distributions fit well to the experimental histograms (*Appendix 4—figure 1*), comparison with the live-cell normalized ACF indicated that the estimates actually became worse than non-curated estimates, with a higher sum of squared residuals (*Figure 5c and d*). The rejection method therefore does not produce reliable estimates.

Next, we considered another data curation method which we call the fusion method. This works by setting to zero all fluorescent intensities in a cell population which were below a certain threshold. In other words, we fused or combined the first $k$ bins of the experimentally obtained histogram of fluorescent intensities, thereby taking into account that the true intensity of bin 0 was artificially distributed over some of the first bins.

*Figure 6* and *Appendix 4—table 4* show that the fusion method led to estimates that varied little with $k$ which enhanced our degree of confidence in them (note that $k = 1$ is the same as the uncurated data). The peak at the zero bin for both G1 and G2 was better captured using the fusion method than using non-curated data (compare *Figure 4b* and *Appendix 4—figure 1*, with *Figure 6b*). Comparison to the autocorrelation function of the live-cell data shows that correction with the fusion method also led to improved transcriptional estimates, as indicated by a reduction in the sum of the squared residuals for all four data sets (*Figure 6c*).

Overall, we conclude that for inferring parameters from the smFISH data, the optimal method is to use nascent cell-cycle-specific data, corrected by the fusion method. The optimally inferred parameters for the four data sets in our study are those given in *Appendix 4—figure 2d*. The profile likelihood estimates of the 95% confidence intervals of these parameters are shown in *Appendix 4—table 5*. Note that in line with our synthetic data study in *Figure 2*, the parameters suffering from the least sample variability were $f_{ON}$ and $\rho$. The rest of the parameters ($\sigma_{off}, \sigma_{on}$ and burst size) suffered more sample variability because the fraction of ON time was high; however since their standard deviation divided by the mean (computed over the four datasets) was not high (in the range of 10-20%), they still can be regarded as useful estimates. Note also that the previous prediction that gene dosage compensation involves regulation of the burst frequency did not change upon correction of the nascent data using the fusion method. All these results were deduced assuming that the two copies in G2 are independent from each other. Inferring rates under the opposite assumption of perfectly synchronized copies (*Appendix 4—table 6*) gave very similar estimates for $\rho$ and $f_{ON}$ (to be expected since according to the synthetic data study, these two are the most robustly estimated parameters for genes spending most of their time in the active state) but different estimates for the rest of the parameters. While such perfect synchronization of alleles is unlikely, some degree of synchronization is plausible and further improvement of the transcriptional parameters in the G2 phase will require its precise experimental quantification.

## Discussion

In this study, we compared the reliability of transcriptional parameter inference from mature and nascent mRNA distributions, with and without taking into account the cell cycle phase. Although these distributions come from the same experiment, we found that the different fits produced very different parameter estimates, ranging from small bursts to very large bursts. Comparison to live-cell data revealed that the optimal inference method is to use nascent mRNA data that is separated by cell cycle.

Our findings illustrate the risk of inferring transcriptional parameters from fitting of mRNA distributions. First of all, as we have shown, these fits are sensitive to the segmentation method which can lead to large errors in the estimates. Secondly, the most common method of parameter inference in the literature is fitting of mature mRNA distributions that are not separated by cell cycle (*Larsson et al., 2019*; *Raj et al., 2006*; *Zenklusen et al., 2008*). Obtaining such distributions is straightforward using

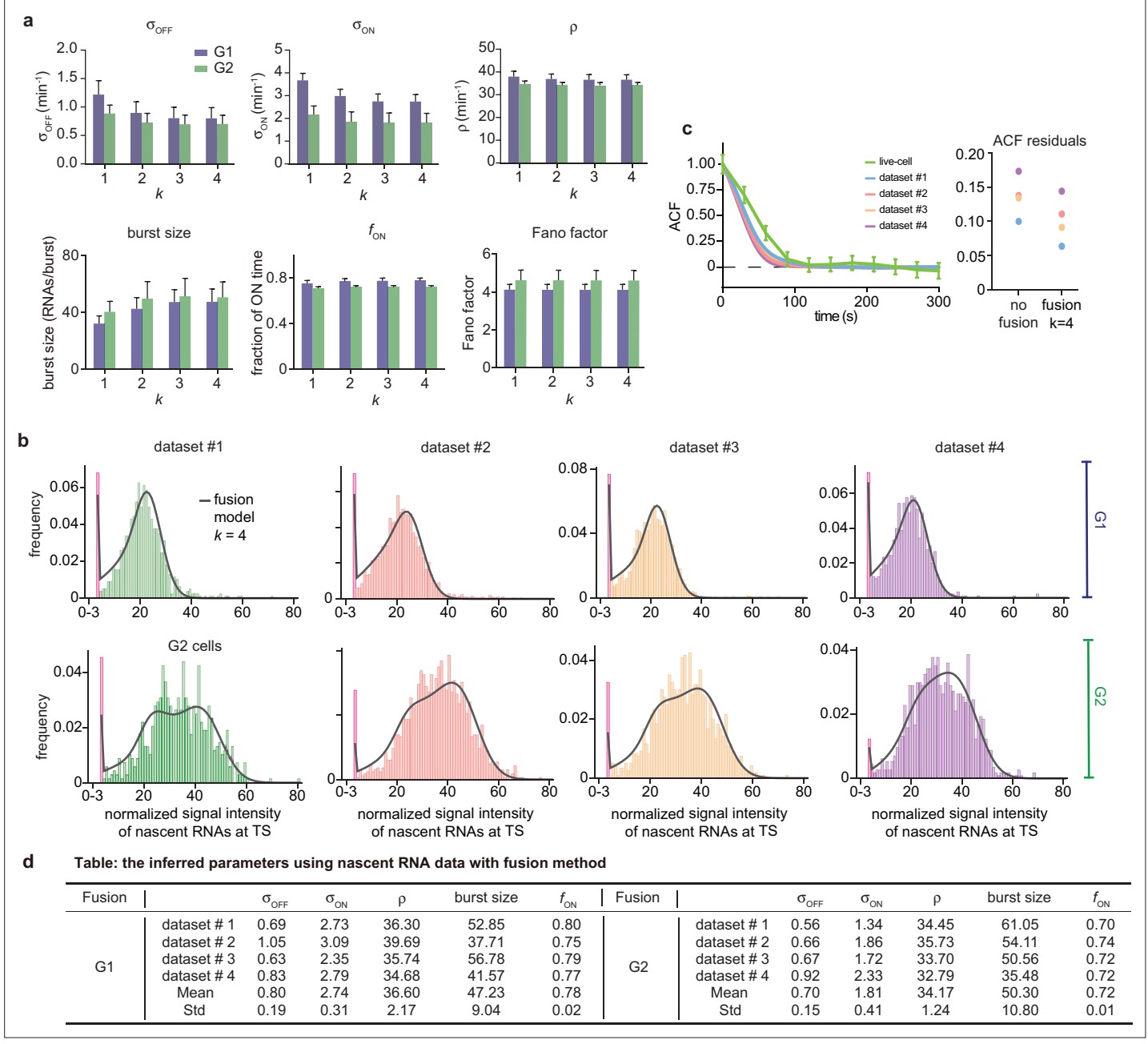

**Figure 6.** Inference results using the fusion method. (**a**) Estimated burst size, fraction of ON and Fano factor (mean values and standard deviation error bars of the four datasets) by combining the first $k$ bins with $k = 1, 2, 3, 4$. (**b**) Corresponding fitted distributions for G1 (top row) and G2 (bottom row) using delay telegraph model with the fusion method (only the distributions for $k = 4$ are shown). Magenta bar represents the combined bin 0–3 when $k = 4$. (**c**) Normalised autocorrelation function (ACF) predicted by stochastic simulations using the estimated parameters (for $k = 4$) for each of the four data sets versus that measured directly using live-cell data (green line). The sum of squared residuals of the ACF plots using cell-cycle specific data without/with fusion method when $k = 4$. (**d**) Estimated parameters of cell cycle specified data and merged data of nascent mRNAs with fusion method with $k = 4$ (fusing bins 0–3). These correspond to the fitted distributions in b. The elongation time $\tau$ is fixed to 0.785 min. See the inferred parameters in **Appendix 4—table 4** for all other values of $k$.

methods such as smFISH, where one can directly count the number of mRNAs per cell. Additionally, with the advance of single-cell mRNA sequencing technologies, it is possible to obtain mRNA distributions for many genes simultaneously and it is tempting to use these to estimate bursting behaviour across the genome (**Kim and Marioni, 2013**; **Larsson et al., 2019**). However, our comparisons on the same dataset show that the values obtained from mature mRNA fits (using merged data) can be

significantly different from the optimal values (using nascent cell-cycle separated data corrected using the fusion method), with underestimation of the burst sizes of almost 10-fold and underestimation of the active fraction of more than 1.5-fold. These results indicate that parameter inference from merged mature mRNA data should be treated with caution. There were smaller differences between the burst size and the active fraction inferred from cell-cycle separated mature and nascent data (only these two can be directly compared because these are non-dimensional); however the relative errors in the estimates (computed over the four datasets) were more than twofold higher for mature data likely due to post-transcriptional noise which nascent data is free from.

It is more common to fit mature distributions rather than nascent distributions because nascent distributions are technically more challenging to obtain. As nascent single-cell sequencing methods are still in the early phase (*Hendriks et al., 2019*), the only method available so far for nascent measurements is smFISH (*Patel et al., 2021*). In such smFISH experiments, intronic probes can be used to specifically label nascent RNA, although there may be some effects of splicing kinetics on the distribution (*Wan et al., 2021*). If introns are not present, like for most yeast genes, one can use exonic probes instead (*Zenklusen et al., 2008*). Since exonic probes label both nascent and mature mRNA transcript, it may be challenging to identify the nascent transcription site unambiguously, especially at lower transcription levels. We show in this manuscript that the fusion method can correct for this bias by combining bins below *k* RNAs, which results in an improvement of the parameter estimates.

Our analysis also emphasizes the importance of separately analyzing G1 and G2 cells (*Skinner et al., 2016*). It is important to note that for cell-cycle-specific analysis, experimental adjustments or cell-cycle synchronized cultures are not required. Although asynchronous cultures consist of a mix G1, S and G2 cells, the integrated DNA intensity of the nucleus of each cell, for example from a DAPI signal, can be used to separate these cells by cell cycle phase *in silico* (*Skinner et al., 2016*; *Roukos et al., 2015*). As most smFISH experiments already include a DNA-labelled channel, adding an extra analysis step should in principle not limit the incorporation of this step in future smFISH fitting procedures.

Even with our optimal fitting strategy, there is a residual error of the simulated ACF and the measured ACF from live-cell measurements. This difference may be the result of different experimental biases of the two measurements. For example, live-cell measurements have a detection threshold below which RNAs may not be detected. In addition, live-cell measurements include cells in S phase, which are not analyzed in smFISH. There could also be differences in the exact percentage of G1 and G2 cells, or other noise sources between live-cell and smFISH experiments. Alternatively, the fit may be imperfect because there might be parameter sets, others than the ones which our inference algorithm found, which provide an accurate fit of the nascent mRNA distribution and perhaps an even better fit to the ACF than we found. We cannot exclude this possibility because we estimated $f_{\mathrm{ON}}$ to be $0.7 - 0.8$ and using synthetic data we showed that the accuracy of some parameters ($\sigma_{\mathrm{on}}, \sigma_{\mathrm{off}}$ and the burst size) deteriorated as $f_{\mathrm{ON}}$ approached 1 (*Figure 2d*). Another factor which could explain the residual error between the simulated ACF and the measured ACF is that perhaps the two-state model may be too simplistic to cover the true promoter states in living cells and may therefore not be able to describe the true *in vivo* kinetics. The promoter may switch between more than 2 states, or there may be sources of extrinsic noise other than the cell cycle that contribute to the heterogeneity. Previous studies have for example identified extrinsic noise on the elongation rate (*Fritzsch et al., 2018*). However, these more complex transcription models also have more parameters, which in practice often means that very few will be identifiable with the current set of experimental observations. To fit these models, one requires temporal data on the transcription kinetics (*Fritzsch et al., 2018*; *Rodriguez et al., 2019*), or simultaneous measurements of various sources of extrinsic noise, such as single-cell transcription factor concentration and RNA polymerase number measurements, cellular volume, local cell crowding, etc, which are often not available in standard smFISH experiments (*Battich et al., 2015*; *Foreman and Wollman, 2020*). Nevertheless, given that there is no explicit time component in smFISH data, the closeness of the simulated ACF to the measured ACF provides confidence we are close to the real values.

The optimal parameter set (*Figure 6d*) indicates long ON promoter times of 75 s, during which almost 50 RNAs are produced in a burst. Large burst sizes (>70) have been previously reported for mouse embryonic stem cells (*Skinner et al., 2016*, mouse hepatocytes *Bahar Halpern et al., 2015* and human fibroblasts *Larsson et al., 2019*). The large burst size and high active fraction of 0.78 suggests

that *GAL10* expression is reaching its limit of maximal expression, which may not be surprising as it is already one of the most highly expressed genes in yeast. It is also interesting to note that the ON time of 75 s is longer than the residence time of a single transcript (47 s), which means that RNA polymerases in the beginning of a burst have already left the locus before the burst has finished.

The optimal parameter set (*Figure 6d*) also indicates partial gene dosage compensation. Specifically the burst frequency per gene copy ($\sigma_{on}$) in the G2 phase is 0.66 that in the G1 phase; the other transcriptional rates are not significantly different between the two cell cycle phases. The fold change in the burst frequency per gene copy was previously estimated for the *Oct*4 and *Nanog* genes to be 0.63 and 0.71 respectively, in mouse embryonic stem cells (*Skinner et al., 2016*). The similarity of our estimate of the fold change to those previously measured could be explained by the results of a recent study (*Jia et al., 2021*); using a detailed model of gene expression, it was shown that in the absence of a dependence of the initiation rate on cell volume, gene dosage compensation optimally leads to approximate mRNA concentration homeostasis when the fold change in the burst frequency upon DNA replication is $\sqrt{2}/2 \approx 0.71$.

In conclusion, obtaining kinetic information from static distributions can introduce biases. However, we show that it is possible to obtain reasonable estimates that agree with live-cell measurements, if one infers parameters from nascent mRNA distributions that are accounted for cell cycle phase.

## Methods

### Inference from mature mRNA data

#### Mathematical model

The steady-state solution of the telegraph model of gene expression (*Peccoud and Ycart, 1995*) gives mature mRNA distributions. The reaction steps in this model are illustrated in *Figure 1a*. Next we describe the generation of synthetic mature mRNA data and the algorithm used to infer parameters from this data.

#### Generation of synthetic mature mRNA data

We generate parameter sets on an equidistant mesh grid laid over the space:

$$(\sigma_{off}, \sigma_{on}, \rho) \in \left[ \mathrm{Uniform}(0, 150), \mathrm{Uniform}(0, 150), \mathrm{Uniform}(0, 250) \right] , \tag{2}$$

where the units are inverse minute. Furthermore we apply a constraint on the effective transcription rate

$$\hat{\rho} = \frac{\rho \sigma_{on}}{\sigma_{on} + \sigma_{off}} < 100.$$

In each of the three dimensions of the parameter space, we take 10 points that are equidistant, leading to a total of 1000 parameter sets which reduce to 789 after the effective transcription rate constraint is enforced.

We additionally fix the degradation rate $d = 1$ min$^{-1}$. Note that we choose not to vary the degradation rate (as we did for the other three parameters) since it is not possible to infer all four rates simultaneously – this is because the steady-state solution of the telegraph model is a function of the non-dimensional parameter ratios $\rho/d, \sigma_{off}/d$ and $\sigma_{on}/d$ (*Raj et al., 2006*).

Once a set of parameters is chosen, we use the stochastic simulation algorithm (SSA *Gillespie, 2007*) to simulate the telegraph model reactions in *Figure 1a* and generate $10^4$ samples of synthetic data. Note that each sample mimics a single cell measurement of mature mRNA.

#### Steps of the algorithm to estimate parameters from mature mRNA data

The inference procedure consists of the following steps: (i) select a set of random transcriptional parameters; (ii) use the solution of the telegraph model to calculate the probability of observing the number of mature mRNA measured for each cell; (iii) evaluate the likelihood function for the observed data; (iv) iterate the procedure until the negative log-likelihood is minimized; (v) the set of parameters that accomplishes the latter provides the best point-estimate of the parameters of the telegraph model that describes the measured mature mRNA fluctuations.

For step (i), we restrict the search for optimal parameters in the following region of parameter space

$$(\sigma_{\text{off}}, \sigma_{\text{on}}, \rho) \in \left[ \text{Uniform}(0, 250), \text{Uniform}(0, 250), \text{Uniform}(0, 300) \right] \ (\text{min}^{-1}) =: \Theta. \tag{3}$$

The degradation rate is fixed to $d = 1 \ \text{min}^{-1}$.

Step (ii) can be obtained either by computing the distribution from the analytical solution (*Peccoud and Ycart, 1995* or by using the finite state projection (FSP) method *Munsky and Khammash, 2006*). Here, for the sake of computational efficiency, we use the FSP method to compute the probability distribution of mature mRNA numbers.

For step (iii) we calculate the likelihood of observing the data given a chosen parameter set $\theta$

$$\mathcal{L}(\theta) = \prod_{j=1}^{N_{\text{cell}}} P(n_j; \theta), \tag{4}$$

where $P(n_j; \theta)$ is the probability distribution of mature mRNA numbers obtained from step (ii) given a parameter set $\theta$, $n_j$ is the total number of mature mRNA from cell $j$ and $N_{\text{cell}}$ is the total number of cells.

Steps (i) and (iv) involve an optimization problem. Specifically we use a gradient-free optimization algorithm, namely *adaptive differential evolution optimizer* (ADE optimizer) using *BlackBoxOptim. jl* (https://github.com/robertfeldt/BlackBoxOptim.jl; *Feldt and Stukalov, 2022*) within the *Julia* programming language to find the optimal parameters

$$\theta^* = \underset{\theta \in \Theta}{\arg\min} \left( -\sum_{j=1}^{N_{\text{cell}}} \log P(n_j; \theta) \right). \tag{5}$$

The minimization of the negative log-likelihood is equivalent to maximizing the likelihood. Note the optimization algorithm is terminated when the number of iterations is larger than $10^4$; this number is chosen because we have found that invariably after this number of iterations, the likelihood has converged to some maximal value. Note that the inference algorithm is particularly low cost computationally, with the optimal parameter values estimated in at most a few minutes.

Once the best parameter set $\theta^*$ is found, we calculate the mean relative error (MRE) which is defined as

$$\text{MRE} = \frac{1}{M} \sum_{i=1}^{M} \text{Relative error}(\theta_i^*, \theta_{\text{true},i}),$$
$$\text{Relative error}(\theta_i^*, \theta_{\text{true},i}) = \frac{|\theta_i^* - \theta_{\text{true},i}|}{|\theta_{\text{true},i}|} \tag{6}$$

where $\theta_i^*$ and $\theta_{\text{true},i}$ represent the $i$-th estimated and true parameters respectively, and $M$ denotes the number of the estimated parameters. Thus, the mean relative error reflects the deviation of the estimated parameters from the true parameters.

## Inference from nascent mRNA data

### Mathematical model

The steady-state solution of the delay telegraph model (*Xu et al., 2016*) gives the distribution of the number of bound Pol II. In Appendix 6, we present an alternative approach to derive the steady-state solution. The reaction steps are illustrated in *Figure 1a*.

The position of a Pol II molecule on the gene determines the fluorescence intensity of the mRNA attached to it. In particular for fluorescence data acquired from smFISH *PP7-GAL10*, the fluorescence intensity of a single mRNA on the DNA locus looks like a trapezoidal pulse (see *Figure 1b* for an illustration). This presents a problem because although we can predict the distribution of the number of bound Pol II using the delay telegraph model, we do not have any specific information on their spatial distribution along the gene. However, since the delay telegraph model implicitly assumes that a Pol II molecule has fixed velocity and that Pol II molecules do not interact with each other (via volume exclusion), it is reasonable to assume that in steady-state, the bound Pol II molecules are uniformly distributed along the gene. This hypothesis is confirmed by stochastic simulations of the delay telegraph

model where the position of a Pol II molecule is calculated as the product of the constant Pol II velocity and the time since its production.

By the uniform distribution assumption and the measured trapezoidal fluorescence intensity profile, it follows that the signal intensity of each bound Pol II has the density function $g$ defined by

$$g(s) = \tfrac{L_1}{L}\chi_{[0,1]}(s) + \tfrac{L_2}{L}\delta_1(s), \; s \in [0, 1],$$

where $L_1 = 862 \text{ bp}$ (base pairs), $L_2 = 2200 \text{ bp}$, $L = L_1 + L_2$ as defined in **Figure 1b**. The indicator function $\chi_{[0,1]}(s) = 1$ if and only if $s \in [0, 1]$ and $\delta_1(s)$ is the Dirac function at 1. The probability of the signal $s$ being between 0 and 1 is due to the first part of the trapezoid function and hence is multiplied by $L_1/L$ which is the probability of being in this region if Pol II is uniformly distributed. Similarly, the probability of $s$ being 1 is due to the $L_2$ part of the trapezoid and hence the probability is $L_2/L$ by the uniform distribution assumption. Note that the signal $s$ from each Pol II is at most 1 because in practice, the signal intensity from the transcription site is normalized by the median intensity of single cytoplasmic mRNAs (**Zenklusen et al., 2008**).

The total signal is the sum of the signals from each bound Pol II. Hence, the density function of the sum is given by the convolution of the signal densities from each bound Pol II. Defining $p(s|k)$ as the density function of the signal given there are $k$ bound Pol II molecules, we have that $p(s|k)$ is the $k$–th convolution power of $g$, that is

$$p(s|k) = (g * g \cdots * g)(s) = g^{*k}(s), \quad g^{*0}(s) = \delta_0(s), \tag{7}$$

where $\delta_0(s)$ is the Dirac function at. Finally we can write the total fluorescent signal density function as

$$p(s; \theta) = \sum_{k=0}^{\infty} p(s|k)P(k; \theta), \tag{8}$$

where $P(k; \theta)$ is the steady-state solution of the delay telegraph model giving the probability of observing $k$ bound Pol II molecules for the parameter set $\theta$. Hence **Equation (8)** represents the extension of the delay telegraph model to predict the smFISH fluorescent signal of the transcription site. Comparison to the algorithm in **Xu et al., 2016**. Both algorithms take into account the fact that the signal intensity depends on the position of Pol II on the gene, albeit this is done in different ways. In **Xu et al., 2016** a master equation is written for the joint distribution of gene state and the number of nascent mRNA. In this case the number of nascent RNAs can have non-integer values since it represents the experimentally measured signal from the (incomplete) nascent RNA. Solution of this master equation proceeds by (a) a discretization of the continuous nascent mRNA signal into bins which are much smaller than one; (b) solution using finite state projection (FSP). This approach can lead to a large state space which incurs a large computational cost. In contrast, in our method, we use FSP to solve for the delay telegraph model, i.e. the distribution of the discrete number of bound Pol II from which we construct (using convolution) the approximate distribution of the continuous nascent mRNA signal by assuming the Pol II is uniformly distributed on the gene. Since the state space of bound Pol II is typically not large, our method will typically be more computationally efficient than the one described in **Xu et al., 2016**.

## Generation of synthetic nascent mRNA data

We generated synthetic smFISH signal data by using the SSA, modified to include delay to simulate the delay telegraph model (**Fu et al., 2022**). Specifically, we use Algorithm 2 described in **Barrio et al., 2006**. One run of the algorithm simulates the fluctuating number of bound Pol II molecules in a single cell.

The total fluorescence intensity (mimicking smFISH) is obtained as follows. When a particular bound Pol II is produced by a firing of the transcription reaction $G \rightarrow G + N$, we record this production time; since the elongation rate is assumed to be constant, given the production time we can calculate the position of the Pol II molecule on the gene at any later time and hence using **Figure 1b** we can deduce the fluorescent signal due to this Pol II molecule.

Specifically we normalize each transcribing Pol II's position to $[0, 1]$ and map the position to its normalized signal by

$$q(x) = \begin{cases} x\frac{L}{L_1} & x \in \left[0, \frac{L_1}{L}\right], \\ 1 & x \in \left[\frac{L_1}{L}, 1\right], \end{cases}$$

where $x$ is the normalized position on the gene. Thus at a given time, the total fluorescent signal from the $n$-th cell (the $n$-th realization of the SSA) equals

$$q_n = \sum_{j=1}^{J_n} q(x_j),$$

where $J_n$ is the number of bound Pol II molecules in the $n$-th cell, and $\{x_j\}$ with $j = 1, \ldots, J_n$ is the vector of all Pol II positions on the gene. The total signal from each cell is a real number but it is discretized into an integer.

The kinetic parameters are chosen from the same region of parameter space as in (2), on the same equidistant mesh grid and with the same constraint on the effective transcription rate. Unlike the mature mRNA case, here there is no degradation rate; instead we have the elongation time, which we fix to $\tau = 0.5$ (min). Note that fixing this time is necessary since it is not possible to infer the three transcriptional parameters rates and the elongation time simultaneously because the steady-state solution of the delay telegraph model is a function of the non-dimensional parameter ratios $\rho\tau, \sigma_{\text{off}}\tau$ and $\sigma_{\text{on}}\tau$. Once a set of parameters is chosen, we use the modified SSA (as described above) to simulate the signal intensity in each of $10^4$ cells.

## Steps of the algorithm to estimate parameters from nascent mRNA data

The inference procedure is essentially the same as steps (i)-(v) described in mature mRNA inference except for the following points.

In step (ii), the probability of observing a total signal of intensity $i$ from a single cell is obtained by integrating $p(s; \theta)$ in **Equation (8)** on an interval $[i - 1, i]$ for $i \in \mathbb{N}$ which, in our numerical scheme, means

$$S(i; \theta) := \sum_{k=0}^{K} P(k; \theta) \int_{i-1}^{i} g^{*k}(x)\mathrm{d}x, \quad i = 1, 2, \ldots \tag{9}$$

Note that the integration over the interval of length 1 is to match the discretization of the synthetic data and $\theta \in \Theta$. Intuitively, one can always choose a positive integer $K$ such that $P(k) = 0$ for any $k \geq K$. The computation of the solution of the delay telegraph model $P(k)$ can be done either using the analytical solution (evaluated using high precision) or using the finite state projection algorithm (FSP) **Munsky and Khammash, 2006**. In **Appendix 6—figure 1** and **Appendix 6—table 1**, we show that the two methods yield comparable accuracy and CPU time.

For step (iii) we calculate the likelihood of observing the data given a chosen parameter set $\theta$

$$\mathcal{L}(\theta) = \prod_{j=1}^{N_{\text{cell}}} S(q_j; \theta), \tag{10}$$

where $q_j$ is the discretized total signal intensity from cell $j$ and $N_{\text{cell}}$ is the total number of cells. In the optimization, we aim to find

$$\theta^* = \arg\min_{\theta \in \Theta} \left( -\sum_{j=1}^{N_{\text{cell}}} \log S(q_j; \theta) \right).$$

The whole procedure (for both mature and nascent mRNA inference) is summarized by a flow-chart in **Figure 1c**.

## Experimental data acquisition and processing

A diploid yeast strain of BY4743 background with a single integration of 14xPP7 loops at the 5'UTR of *GAL10* (strain YTL047 **Donovan et al., 2019**) was used in this study. Four replicate yeast cultures were grown in synthetic complete media with 2% galactose to early mid-log (OD 0.5), fixed with 5% paraformaldehyde (PFA) for 20 min, permeabilized with 300 units of lyticase and hybridized with

7.5 pmol each of four PP7 probes labeled with Cy3 (Integrated DNA Technologies) as described in *Trcek et al., 2012* and *Lenstra et al., 2015*; *Patel et al., 2021*, resulting in four technical replicates. The PP7 probe sequences are: atatcgtctgctcctttcta, atatgctctgctggtttcta, gcaattaggtaccttaggat, aatg aacccgggaatactgc. Coverslips were mounted on microscope slides using mounting media with DAPI (ProLong Gold, Life Technologies).

The coverslips were imaged on a Zeiss AxioObserver (Zeiss, USA) widefield microscope with a Plan-Apochromat 40x1.4 NA oil DIC UV objective and a 1.25 x optovar. For Cy3, a 562 nm longpass dichroic (Chroma T562lpxr), 595/50 nm emission filter (Chroma ET595/50 m) and 550/15 nm LED excitation at full power (Spectra X, Lumencor) were used. For DAPI, a 425 nm longpass dichroic (Chroma T425lpxr) and a 460/50 nm emission filter (Chroma ET460/50 m) and LED excitation at 395/25 nm at 25% power (Spectra X, Lumencor) were used. The signal was detected on a Hamamatsu ORCA-Flash4.0 V3 Digital CMOS camera (Hamamatsu Photonics, Japan). For each sample and each channel, we utilized the Micro-Manager software (UCSF) to acquire at least 20 fields-of-view based on the DAPI channel. Each field-of-view consisted of 13 z-stacks (with a z-step of 0.5 μm) at 25ms exposure for DAPI and 250ms exposure for Cy3.

A custom python pipeline was used for analysis (https://github.com/Lenstralab/smFISH; *Pomp, 2022*). Maximum intensity projected images were used to segment the cell and nucleus using Otsu thresholding and watershedding (segmentation 1). In addition, we segmented cells using CellProfiler (segmentation 2). The diffraction-limited Cy3 spots were detected per z-slice using band-pass filtering and refined using iterative Gaussian mask localization procedure (*Crocker and Grier, 1996*; *Thompson et al., 2002*; *Larson et al., 2005*; *Larson et al., 2011* and *Coulon et al., 2014*). Cells in which no spots were detected were excluded from further analysis since a visual inspection indicated that these cells were not properly segmented or were improperly permeabilized.

Spots were classified as nuclear or cytoplasmic and the brightest nuclear spots were classified as transcription sites. The intensity of the brightest nuclear spot in a cell was normalized with the median fluorescence intensity of all the cytoplasmic spots in all cells. This is due to the fact that 90% of cytoplasmic mRNAs are isolated (*Figure 4a*), thus the median of the fluorescence signal of cytoplasmic mRNAs can be considered as the normalizing value. The distribution of the normalised intensity of the brightest nuclear spot, calculated over the cell population, is the experimental equivalent of the total fluorescent signal density function as given by the solution of the modified delay telegraph model, *Equation (8)*.

The number of mature mRNA in each cell is given by counting the number of spots in the entire cell, that is nuclear plus cytoplasmic. The transcription site is counted as 1 mRNA, regardless of its intensity. We show in *Figure 3c* that this has negligible influence on the estimated parameters since the mean number of mature mRNA is much greater than 1. The distribution of the number of spots is the experimental equivalent of the solution of the telegraph model, that is the marginal distribution of mature mRNA numbers in steady-state conditions.

The integrated nuclear intensity of each cell was calculated by summing the DNA content intensity (DAPI) of all the pixels within the nucleus mask. The distribution of the intensities was fit with a bimodal Gaussian distribution. Those cells whose intensity was within a standard deviation of the mean of the first (second) Gaussian peak was classified as G1 (G2) (see *Figure 3e* left). This gave similar results to a different cell cycle classication method using the Fried/Baisch model (*Johnston et al., 1978*) which was recently employed in *Skinner et al., 2016*. See *Appendix 7—figure 1* for a comparison of the two methods. We note that cells in late G2 may contain two separate transcription sites, one in the mother and one in the bud. When the nucleus moves into the bud, buds often contain less DNA than G1 cells, and mothers contain more DNA than G1 cells, both of which are excluded from the analysis. When the DNA content of the mother and daughter is similar, both mother and daughter are counted separately as G1 cells. We note that this late G2 subpopulation is very small.

We did four independent experiments with a total number of cells equal to 2510, 6411, 4592, 3181, respectively. After classification, the numbers of G1 cells are 766, 2111, 1495, 904 and the number of G2 cells are 683, 1657, 1209, 1143, whereas the rest were classified as undetermined.

## Data availability

The four smFISH datasets are available from https://osf.io/d5nvj/. These datasets include the maximum intensity projected images, the spot localization results, the nuclear and cellular masks used

for merged, G1 and G2 cells and the analyzed results of the mature and nascent data. The analysis code of the smFISH microscopy data is available at https://github.com/Lenstralab/smFISH; *Pomp, 2022*. The code for the the synthetic simulations and the parameter inference is available at https://github.com/palmtree2013/RNAInferenceTool.jl; *Fu, 2022*.

## Acknowledgements

ZC, XF, and LX acknowledge the support from Natural Science Foundation of China (NSFC No. 61988101, 62073137), Shanghai Action Plan for Technological Innovation Grant (No. 22ZR1415300, 22511104000) and Shanghai Center of Biomedicine Development. XF acknowledges the support from Shanghai Sailing Program (22YF1410700). TLL was supported by the Netherlands Organization for Scientific Research (NWO, gravitation program CancerGenomiCs.nl), Oncode Institute, which is partly financed by the Dutch Cancer Society, and the European Research Council (ERC Starting Grant 755695 BURSTREG). RG was supported by a Leverhulme Trust research award (RPG-2020–327).

## Additional information

### Funding

| Funder | Grant reference number | Author |
|---|---|---|
| National Natural Science Foundation of China | 61988101 | Zhixing Cao Xiaoming Fu Libin Xu |
| National Natural Science Foundation of China | 62073137 | Zhixing Cao Xiaoming Fu Libin Xu |
| H2020 European Research Council | 755695 BURSTREG | Tineke L Lenstra |
| Leverhulme Trust | RPG-2020-327 | Ramon Grima |
| Shanghai Action Plan for Technological Innovation Grant | 22ZR1415300 | Zhixing Cao Xiaoming Fu Libin Xu |
| Shanghai Action Plan for Technological Innovation Grant | 22511104000 | Zhixing Cao Xiaoming Fu Libin Xu |
| Shanghai Sailing Program | 22YF1410700 | Xiaoming Fu |
| Oncode Institute | | Tineke L Lenstra |
| Netherlands Organisation for Scientific Research | Gravitation program CancerGenomiCs.nl | Tineke L Lenstra |

The funders had no role in study design, data collection and interpretation, or the decision to submit the work for publication.

### Author contributions

Xiaoming Fu, Software, Formal analysis, Visualization, Writing – original draft, Writing – review and editing; Heta P Patel, Investigation, Visualization, Writing – original draft, Writing – review and editing; Stefano Coppola, Software, Formal analysis; Libin Xu, Formal analysis; Zhixing Cao, Formal analysis, Supervision, Writing – original draft, Project administration; Tineke L Lenstra, Conceptualization, Supervision, Methodology, Writing – original draft, Project administration, Writing – review and editing, Funding acquisition; Ramon Grima, Conceptualization, Supervision, Methodology, Writing – original draft, Project administration, Writing – review and editing

### Author ORCIDs

Xiaoming Fu http://orcid.org/0000-0003-4073-9822
Heta P Patel http://orcid.org/0000-0002-1618-951X
Zhixing Cao http://orcid.org/0000-0003-2600-5806

Tineke L Lenstra http://orcid.org/0000-0002-4440-9962
Ramon Grima http://orcid.org/0000-0002-1266-8169

**Decision letter and Author response**
Decision letter https://doi.org/10.7554/eLife.82493.sa1
Author response https://doi.org/10.7554/eLife.82493.sa2

## Additional files

### Supplementary files
• MDAR checklist

### Data availability

The four smFISH datasets are available from https://osf.io/d5nvj/. These datasets include the maximum intensity projected images, the spot localization results, the nuclear and cellular masks used for merged, G1 and G2 cells and the analyzed results of the mature and nascent data. The analysis code of the smFISH microscopy data is available at https://github.com/Lenstralab/smFISH (copy archived at swh:1:rev:b49af68653e9fdcab3fa48085f648fc86d8c659e). The code for the the synthetic simulations and the parameter inference is available at https://github.com/palmtree2013/RNAInferenceTool.jl (copy archived at swh:1:rev:be2fcc8f7a811a571a297d3e150395c0a73add09).

The following dataset was generated:

| Author(s) | Year | Dataset title | Dataset URL | Database and Identifier |
|---|---|---|---|---|
| Lenstra TL | 2022 | smFISH datasets for PP7-GAL10 in budding yeast | https://osf.io/d5nvj/ | Open Science Framework, d5nvj |

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

# Appendix 1

## Accuracy of inference from synthetic mature and nascent mRNA data

### Inference from synthetic mature mRNA data with external noise

In the main text, *Figure 2*, we showed how the addition of 5% external noise to synthetic mature mRNA data degrades the inference accuracy. In *Appendix 1—figure 1* we show how the addition of a larger amount of external noise (10%) causes an even larger loss of accuracy. In particular for 91% of the parameters, the inference accuracy is higher when using nascent mRNA data (*Appendix 1—figure 1a*) and the median relative errors become very high for most parameters, especially for $\rho$ and $\sigma_{\text{off}}$ in the limit of large $f_{\text{ON}}$ (*Appendix 1—figure 1b*).

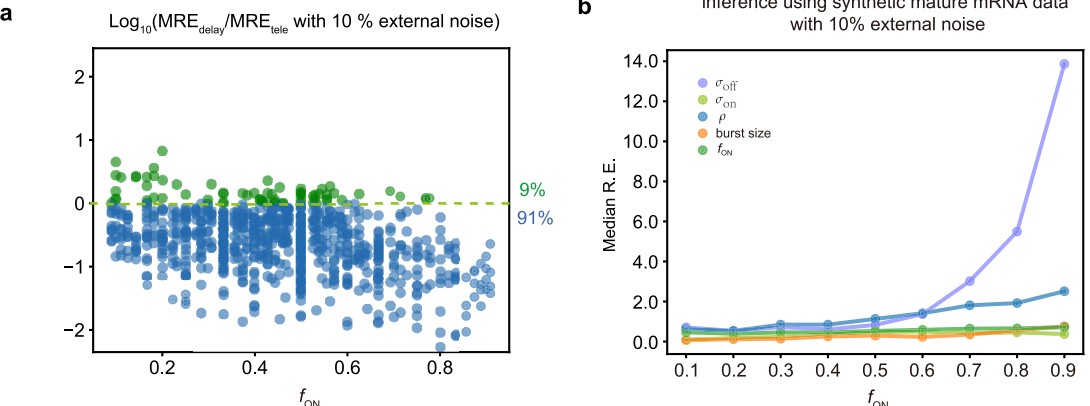

**Appendix 1—figure 1.** Comparing inference accuracy using synthetic nascent mRNA data and synthetic mature mRNA data with 10% external noise (log-normal distributed noise is added to the initiation rate $\rho$ to mimic external noise due to post-transcriptional processing that is only present in mature mRNA). (**a**) Ratio of the mean relative errors in the two types of data as a function of the true fraction of ON time, $f_{\text{ON}}$. For ≈91% (719/789) of the parameters, the inference accuracy is higher when using nascent mRNA data. (**b**) The median relative error of each transcriptional parameter as a function of the fraction of ON time using synthetic mature mRNA.

### Accuracy of distribution fits

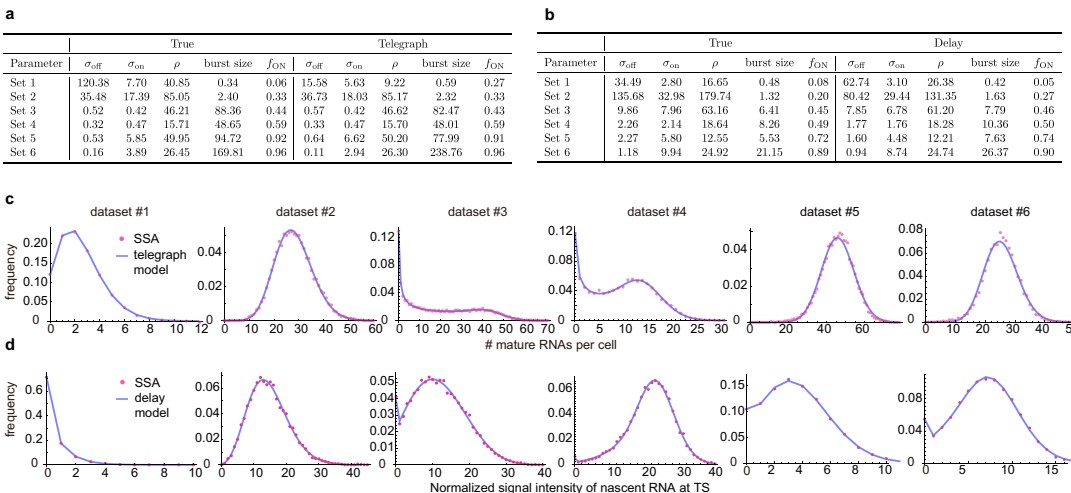

**Appendix 1—figure 2.** Inference with the telegraph model and delay telegraph model for six parameter sets. (**a**) Estimates using the inference algorithm with the telegraph model (with no external noise) for six parameter sets. For both the ground truth and the estimated parameters, we fix the degradation rate $d = 1\ \text{min}^{-1}$. (**b**) Estimates using the inference algorithm with the delay telegraph model for six parameter sets. For both the ground truth and

*Appendix 1—figure 2 continued*
the estimated parameters, we fix the delay $\tau = 0.5$ min. (**c**) Distributions from synthetic mature mRNA data fitted using the telegraph model. (**d**) Distributions from synthetic nascent mRNA data fitted using the delay telegraph model.

In the main text, *Figure 2*, we showed how the accuracy of parameter estimation is not uniform across parameter space. Here we investigate if there is any relationship between this accuracy and how well is a distribution of mature mRNA numbers / signal intensity fit by the inference algorithm. For 12 parameter sets (6 for the telegraph model – *Appendix 1—figure 2a*) and (6 for the delay telegraph model – *Appendix 1—figure 2b*), we evaluate the fits to the distribution of synthetic data in *Appendix 1—figure 2c–d*. The results show that independent of the accuracy of the parameters estimated by the inference algorithm, the fits of the delay telegraph and telegraph model distributions to the distributions generated from synthetic data are generally excellent.

## Testing the variability of the inference procedure

To obtain a better understanding of the variability of the inference procedure, for each of the six parameter sets in *Appendix 1—figure 2b* and i.e. for the inference using the delay telegraph model, we generated 10 independent sets of synthetic data and used maximum likelihood to infer the parameters for each dataset. The mean and standard deviation of the parameters (computed over all 10 datasets) are shown in *Appendix 1—table 1*. The means are close to the true parameter values (in *Appendix 1—figure 2b*); this shows that the inference procedure is working correctly and the deviations from ground truth values are mostly due to noise in the synthetic datasets. We can define the sample variability of a parameter as the standard deviation divided by the mean. Ordering parameters by this quantity, we find that for all six parameter sets the error is largest for $\sigma_{\text{off}}$. For small $f_{\text{ON}}$, the parameters ordered by increasing error are: $\sigma_{\text{on}}$, burst size, $\rho$, $f_{\text{ON}}$ and $\sigma_{\text{off}}$. While for large $f_{\text{ON}}$, the order is: $\rho$, $f_{\text{ON}}$, $\sigma_{\text{on}}$, burst size and $\sigma_{\text{off}}$. Note that this order is the same as we determined using the relative errors (from the ground truth) which is shown in *Figure 2* of the main text.

**Appendix 1—table 1.** Mean and standard deviation of the parameters estimated from 10 independent synthetic datasets, generated for each parameter set in *Appendix 1—figure 2*.

| Parameter | Mean | | | | | Standard deviation | | | | |
|---|---|---|---|---|---|---|---|---|---|---|
| | $\sigma_{\text{off}}$ | $\sigma_{\text{on}}$ | $\rho$ | burst size | $f_{\text{ON}}$ | $\sigma_{\text{off}}$ | $\sigma_{\text{on}}$ | $\rho$ | burst size | $f_{\text{ON}}$ |
| Set 1 | 31.85 | 2.69 | 15.68 | 0.53 | 0.10 | 19.07 | 0.22 | 7.68 | 0.07 | 0.05 |
| Set 2 | 92.67 | 30.47 | 141.42 | 1.56 | 0.25 | 23.76 | 1.51 | 21.61 | 0.14 | 0.04 |
| Set 3 | 9.94 | 8.01 | 63.35 | 6.39 | 0.45 | 0.72 | 0.19 | 1.94 | 0.25 | 0.01 |
| Set 4 | 2.30 | 2.16 | 18.76 | 8.17 | 0.48 | 0.12 | 0.05 | 0.31 | 0.31 | 0.01 |
| Set 5 | 2.26 | 5.80 | 12.57 | 5.59 | 0.72 | 0.19 | 0.30 | 0.11 | 0.41 | 0.01 |
| Set 6 | 1.22 | 10.01 | 24.97 | 20.53 | 0.89 | 0.10 | 0.43 | 0.16 | 1.65 | 0.01 |

## Confidence intervals using profile likelihood

We perform a profile likelihood study (*Kreutz et al., 2013*) on the 12 parameter sets of synthetic mature/nascent mRNA data described in *Appendix 1—figure 2a-b*. We obtain the 95% confidence interval for each parameter. The results are shown in *Appendix 1—table 2*. We also compare the relative errors for each parameter (computed as (estimated value - ground truth)/ground truth using the data in *Appendix 1—figure 2a and b*) with the profile likelihood error (computed as (upper bound - lower bound)/optimal estimate using the bounds in *Appendix 1—table 2* and the optimal values in *Appendix 1—figure 2a and b*). The results are shown in *Appendix 1—table 3*. Note that in most cases, the parameters ordered by relative error are in agreement with the parameters ordered by profile likelihood error.

**Appendix 1—table 2.** 95% confidence intervals of the 12 parameter sets (shown in **Appendix 1—figure 2a-b**).

| | Telegraph CI | | | Delay CI | | |
|---|---|---|---|---|---|---|
| Parameter | $\sigma_{\text{off}}$ | $\sigma_{\text{on}}$ | $\rho$ | $\sigma_{\text{off}}$ | $\sigma_{\text{on}}$ | $\rho$ |
| Set 1 | (6.76, 300.00) | (3.53, 8.49) | (5.59, 107.67) | (13.80, 300.00) | (1.85, 2.69) | (9.94, 160.51) |
| Set 2 | (17.22, 190.38) | (14.56, 23.92) | (61.14, 250.51) | (103.80, 268.35) | (30.00, 35.38) | (161.54, 300.00) |
| Set 3 | (0.54, 0.59) | (0.41, 0.43) | (46.08, 47.15) | (6.83, 8.55) | (6.42, 7.04) | (58.28, 63.18) |
| Set 4 | (0.31, 0.35) | (0.46, 0.49) | (15.50, 15.91) | (1.64,1.99) | (1.65,1.85) | (17.92,18.77) |
| Set 5 | (0.49, 0.85) | (5.82, 7.62) | (49.62, 50.92) | (1.56,2.59) | (4.58,5.85) | (12.08,12.97) |
| Set 6 | (0.08, 0.16) | (2.46, 3.54) | (26.10, 26.54) | (0.73,1.24) | (7.53,9.75) | (24.32,25.14) |

**Appendix 1—table 3.** Table showing the relative error against profile likelihood error of 12 parameter sets (shown in **Appendix 1—figure 2a and b**). See text for details.

| | Telegraph | | | | | | Delay | | | | | |
|---|---|---|---|---|---|---|---|---|---|---|---|---|
| | Relative error | | | Profile likelihood error | | | Relative error | | | Profile likelihood error | | |
| | $\sigma_{\text{off}}$ | $\sigma_{\text{on}}$ | $\rho$ | $\sigma_{\text{off}}$ | $\sigma_{\text{on}}$ | $\rho$ | $\sigma_{\text{off}}$ | $\sigma_{\text{on}}$ | $\rho$ | $\sigma_{\text{off}}$ | $\sigma_{\text{on}}$ | $\rho$ |
| Set 1 | 0.87 | 0.27 | 0.77 | 18.82 | 0.88 | 11.07 | 0.82 | 0.11 | 0.58 | 1.75 | 0.34 | 1.82 |
| Set 2 | 0.04 | 0.04 | 1.44E-03 | 4.72 | 0.52 | 2.22 | 0.41 | 0.11 | 0.27 | 0.81 | 0.16 | 0.55 |
| Set 3 | 0.08 | 0.01 | 0.01 | 0.09 | 0.06 | 0.02 | 0.20 | 0.15 | 0.03 | 0.22 | 0.09 | 0.08 |
| Set 4 | 0.01 | 0.01 | 1.11E-03 | 0.14 | 0.07 | 0.03 | 0.22 | 0.18 | 0.02 | 0.20 | 0.11 | 0.05 |
| Set 5 | 0.22 | 0.13 | 4.97E-03 | 0.55 | 0.27 | 0.03 | 0.30 | 0.23 | 0.03 | 0.64 | 0.28 | 0.07 |
| Set 6 | 0.29 | 0.24 | 0.01 | 0.74 | 0.37 | 0.02 | 0.20 | 0.12 | 0.01 | 0.55 | 0.25 | 0.03 |

## Effect of random perturbation of mature mRNA data on inference

To assess the reliability of the inference results due to errors in spot counting, we redid the inference with synthetic mRNA data (and using the telegraph model) perturbed randomly by minus 1/plus 1/ unchanged with probability 1/3. The results are shown in **Appendix 1—table 4**.

We found that when the fraction of ON time was very small (Set 1), there is a considerable effect of the perturbations on the values of the inferred parameters – this is because in this case, the mean number of mRNA is very small and hence a perturbation of one molecule is very significant. However as expected, the inference results are quite robust when the fraction of ON time is not too small (Sets 2–6).

**Appendix 1—table 4.** Effects of random perturbations on inference of parameters from mature mRNA data (using the telegraph model).

| | True | | | | | Unperturbed | | | | | −1/0/+1 stochastic perturbation | | | | |
|---|---|---|---|---|---|---|---|---|---|---|---|---|---|---|---|
| Parameter | $\sigma_{\text{off}}$ | $\sigma_{\text{on}}$ | $\rho$ | burst size | $f_{\text{ON}}$ | $\sigma_{\text{off}}$ | $\sigma_{\text{on}}$ | $\rho$ | burst size | $f_{\text{ON}}$ | $\sigma_{\text{off}}$ | $\sigma_{\text{on}}$ | $\rho$ | burst size | $f_{\text{ON}}$ |
| Set 1 | 120.38 | 7.70 | 40.85 | 0.34 | 0.06 | 15.58 | 5.63 | 9.22 | 0.59 | 0.27 | 0.59 | 1.17 | 3.70 | 6.28 | 0.66 |
| Set 2 | 35.48 | 17.39 | 85.05 | 2.40 | 0.33 | 36.73 | 18.03 | 85.17 | 2.32 | 0.33 | 24.13 | 15.79 | 70.89 | 2.94 | 0.40 |
| Set 3 | 0.52 | 0.42 | 46.21 | 88.36 | 0.44 | 0.57 | 0.42 | 46.62 | 82.47 | 0.43 | 0.61 | 0.46 | 47.17 | 76.74 | 0.43 |
| Set 4 | 0.32 | 0.47 | 15.71 | 48.65 | 0.59 | 0.33 | 0.47 | 15.70 | 48.01 | 0.59 | 0.39 | 0.54 | 16.09 | 41.17 | 0.58 |
| Set 5 | 0.53 | 5.85 | 49.95 | 94.72 | 0.92 | 0.64 | 6.62 | 50.20 | 77.99 | 0.91 | 0.68 | 6.72 | 50.35 | 74.48 | 0.91 |
| Set 6 | 0.16 | 3.89 | 26.45 | 169.81 | 0.96 | 0.11 | 2.94 | 26.30 | 238.76 | 0.96 | 0.13 | 3.06 | 26.42 | 203.20 | 0.96 |

## Effect of log-normal noise in nascent fluorescent signal on inference

Given the synthetic nascent mRNA signal data $\{S_i\}_{i=1}^{N}$ (where $N$ represents the number of samples) generated by delay telegraph model, for each $S_i$, we perform a random perturbation $\Phi$ under the following conditions

$$\Phi : S_i \rightarrow \Phi(S_i)$$

where $\Phi$ is a stochastic perturbation satisfying the following constraints
$\Phi(S_i)$ is a random variable sampled from the distribution Log-normal $(\alpha, \beta)$ whose mean equals $S_i$, and the standard deviation equals $0.1 * S_i$.
This means the random perturbation keeps the mean value of the signal $S_i$ unchanged but adds noise with a coefficient of variation equal to 0.1.

In *Appendix 1—table 5* we compare the results of inference using synthetic nascent mRNA data with the aforementioned stochastic perturbation and without. As for mature mRNA data, we find that the perturbation has only a significant impact when the fraction of ON time is very small.

**Appendix 1—table 5.** Inference using the delay telegraph model from synthetic nascent fluorescent data, with and without perturbation by log-normal noise.

| Parameter | True | | | | | Delay | | | | | Perturbation | | | | |
|---|---|---|---|---|---|---|---|---|---|---|---|---|---|---|---|
| | $\sigma_{\text{off}}$ | $\sigma_{\text{on}}$ | $\rho$ | burst size | $f_{\text{ON}}$ | $\sigma_{\text{off}}$ | $\sigma_{\text{on}}$ | $\rho$ | burst size | $f_{\text{ON}}$ | $\sigma_{\text{off}}$ | $\sigma_{\text{on}}$ | $\rho$ | burst size | $f_{\text{ON}}$ |
| Set 1 | 34.49 | 2.80 | 16.65 | 0.48 | 0.08 | 62.74 | 3.10 | 26.38 | 0.42 | 0.05 | 50.08 | 1.52 | 31.71 | 0.63 | 0.03 |
| Set 2 | 135.68 | 32.98 | 179.74 | 1.32 | 0.20 | 80.42 | 29.44 | 131.35 | 1.63 | 0.27 | 233.96 | 30.74 | 300.00 | 1.28 | 0.12 |
| Set 3 | 9.86 | 7.96 | 63.16 | 6.41 | 0.45 | 7.85 | 6.78 | 61.20 | 7.79 | 0.46 | 8.85 | 6.63 | 65.49 | 7.40 | 0.43 |
| Set 4 | 2.26 | 2.14 | 18.64 | 8.26 | 0.49 | 1.77 | 1.76 | 18.28 | 10.36 | 0.50 | 1.90 | 1.64 | 18.65 | 9.83 | 0.46 |
| Set 5 | 2.27 | 5.80 | 12.55 | 5.53 | 0.72 | 1.60 | 4.48 | 12.21 | 7.63 | 0.74 | 2.59 | 4.79 | 13.27 | 5.13 | 0.65 |
| Set 6 | 1.18 | 9.94 | 24.92 | 21.15 | 0.89 | 0.94 | 8.74 | 24.74 | 26.37 | 0.90 | 1.84 | 10.15 | 26.10 | 14.18 | 0.85 |

## Appendix 2

### Telegraph model versus delay telegraph model

In this section, we aim to precisely understand the differences between the telegraph and the delay telegraph models. For both models, we define the rate of switching from the ON state to OFF state as $\sigma_{\text{off}}$, the rate of switching from the OFF state to the ON state as $\sigma_{\text{on}}$ and the production rate of nascent mRNAs in the ON state as $\rho$. The first-order decay rate of nascent mRNA in the telegraph model is given by $d$ and the delay time between initiation and degradation in the delay telegraph model is $\tau$. Note that while the telegraph model was in the main text explained in terms of mature mRNA, in this section we use it as a model for nascent mRNA since we want to compare directly with the delay telegraph model. The telegraph and delay telegraph models can be solved exactly in steady-state conditions (*Peccoud and Ycart, 1995*; *Xu et al., 2016*) (an alternative derivation for the delay telegraph model is also given in Section F). From the generating function solution of the models, one can deduce expressions for the first and second centered moments in steady-state conditions:

$$
\begin{aligned}
\langle n \rangle_{\text{tele}} &= \frac{\rho \sigma_{\text{on}}}{d(\sigma_{\text{on}}+\sigma_{\text{off}})}, \\
\langle n \rangle_{\text{delay}} &= \frac{\rho \sigma_{\text{on}} \tau}{\sigma_{\text{on}}+\sigma_{\text{off}}}, \\
\text{Var}_{\text{delay}} &= \frac{\rho \sigma_{\text{on}} \left( (\sigma_{\text{off}}+\sigma_{\text{on}})^3 \tau + 2\rho \sigma_{\text{off}} \left( e^{-(\sigma_{\text{off}}+\sigma_{\text{on}})\tau} + (\sigma_{\text{off}}+\sigma_{\text{on}})\tau - 1 \right) \right)}{(\sigma_{\text{off}}+\sigma_{\text{on}})^4}, \\
\text{Var}_{\text{tele}} &= \frac{\rho \sigma_{\text{on}} \left( \rho \sigma_{\text{off}} + d(\sigma_{\text{off}}+\sigma_{\text{on}}) + (\sigma_{\text{off}}+\sigma_{\text{on}})^2 \right)}{d(\sigma_{\text{off}}+\sigma_{\text{on}}+d)(\sigma_{\text{off}}+\sigma_{\text{on}})^2},
\end{aligned}
\tag{11}
$$

where $\langle n \rangle_{\text{delay}}$ and $\langle n \rangle_{\text{tele}}$ are the mean number of nascent mRNA in the delay telegraph and telegraph models respectively, and $\text{Var}_{\text{delay}}$ and $\text{Var}_{\text{tele}}$ are the corresponding variances in molecule numbers. To understand the differences between the two models, we set the means of the two models to be the same (by choosing $d = 1/\tau$) and then compute the relative error in their variance predictions:

$$
R = \frac{\text{Var}_{\text{delay}} - \text{Var}_{\text{tele}}}{\text{Var}_{\text{delay}}} = \frac{\left(2(1+x)+e^x(x^2-2)\right)y}{(1+x)(2y+e^x(x+2(x-1)y))},
\tag{12}
$$

where $x$ and $y$ are non-dimensional variables defined as

$$
x = \tau(\sigma_{\text{off}} + \sigma_{\text{on}}), \quad y = (\rho/\sigma_{\text{off}})(\sigma_{\text{off}}/(\sigma_{\text{off}} + \sigma_{\text{on}}))^2.
$$

Note that $x$ is the ratio of the time for nascent mRNA to detach from the gene $\tau$ and the timescale of promoter switching $1/(\sigma_{\text{off}} + \sigma_{\text{on}})$. The non-dimensional parameter $y$ is a measure of the burstiness of gene expression since it increases with the mean burst size $\rho/\sigma_{\text{off}}$ and the fraction of time spent in the OFF state $\sigma_{\text{off}}/(\sigma_{\text{off}} + \sigma_{\text{on}})$. In fact as $\sigma_{\text{off}} \to 0$ which implies $y \to 0$, the telegraph model converges to a constitutive model where the time between two successive nascent mRNA production events is exponentially distributed. Note that the relative errors in the Fano factor and the coefficient of variation squared are the same as the error in the variance since the means of the two models are the same.

Using *Equation (12)*, it is easy to see that the relative error between the two models vanishes in the limit of $x \to 0$ (when the promoter switching timescales are much longer than the time spent by a polymerase on a gene) or in the limit of $y \to 0$ (when there is no burstiness in gene expression). Hence, the telegraph model is an accurate approximation of the delay telegraph model in these two limits. It can be shown that in the first case, the distribution of nascent mRNA numbers is well approximated by the sum of two Poisson distributions, whereas for the second case the distribution is a Poisson. Note that since $R$ is always positive, it follows that the *telegraph model systematically underestimates the size of noise in nascent mRNA numbers*. It can be further shown that $R$ increases monotonically with $x$ and $y$ and the maximum attainable value is $R = 1/2$ (when $x \to \infty, y \to \infty$), that is, the worst prediction of the telegraph model is that the variance is half that of the delay telegraph model.

In *Appendix 2—figure 1a and c* we contrast the distributions of nascent mRNA predicted by the delay telegraph model with those predicted by the telegraph model for the case $d = 1/\tau$ where the means are matched, as also assumed for the calculation of the relative error above. *Appendix 2—figure 1b* shows the effect of increasing $x$ (via $\tau$) and *Appendix 2—figure 1d* shows the effect of increasing $y$ (via $\rho$). The number distributions are constructed from the generating functions of the telegraph model *Peccoud and Ycart, 1995* and of the delay telegraph model (Section F). Note that the shapes of the distributions of the two models can be considerably different, e.g. the cases $\rho = 3$

and $\rho = 10$ in *Appendix 2—figure 1c* shows that the nascent distribution from the delay model is bimodal with peaks at 0 and at a non-zero value but it is unimodal with a non-zero peak from the telegraph model. Hence we conclude that if we are interested in accurately predicting nascent mRNA number distributions then generally the Markovian telegraph model is not a good approximation of the non-Markovian delay telegraph model.

In the main text, we used the delay telegraph model to infer the synthetic data from the synthetic data generated by the delay SSA. For comparison, we repeat the same but now we use the telegraph model (rather than the delay telegraph model) to calculate step (ii) in the inference algorithm, that is $P(k;\theta)$ in *Equation 8* in the main text is now chosen to be the steady-state solution of the telegraph model. Once the best parameter set $\theta^*$ is found, we calculate two scores: (i) the fitness given by the smallest negative log-likelihood value found by the optimizer normalized over the sample size. (ii) the mean relative error (for definition see Equation (4.5) in the main text). In *Appendix 2—figure 1*, we show both of these scores obtained for 20 independent numerical experiments – clearly the error using the delay telegraph model for the inference algorithm is significantly lower than if the telegraph model is used.

These observations are further reinforced in *Appendix 2—figure 1f* where we show the best fit distributions and the corresponding relative errors in the estimates of the burst size and the burst frequency (bar chart insets). These are computed using the formula:

$$\alpha = \text{Relative error in the burst frequency estimate} = \frac{|\sigma_{\text{on}}^* - \sigma_{\text{on}}|}{|\sigma_{\text{on}}^*|},$$
$$\beta = \text{Relative error in the burst size estimate} = \frac{|\rho^*/\sigma_{\text{off}}^* - \rho/\sigma_{\text{off}}|}{|\rho^*/\sigma_{\text{off}}^*|}. \tag{13}$$

We note that the distributions are well fit in all cases, using both telegraph and delay telegraph models. However the errors $\alpha$ and $\beta$ are considerably larger for the former.

In *Appendix 2—table 1* we show the true and estimated parameters for the 6 distributions in *Appendix 2—figure 1f*. The estimates for $\rho$ are accurate independent of the choice of model; however, the estimates for the promoter switching rates $\sigma_{\text{off}}, \sigma_{\text{on}}$ are far more accurate using the delay telegraph model. Hence we conclude that although inference using the telegraph model provides a histogram that fits well with the synthetic data, the inferred parameter values have little meaning because they are not an accurate reflection of the true parameter values.

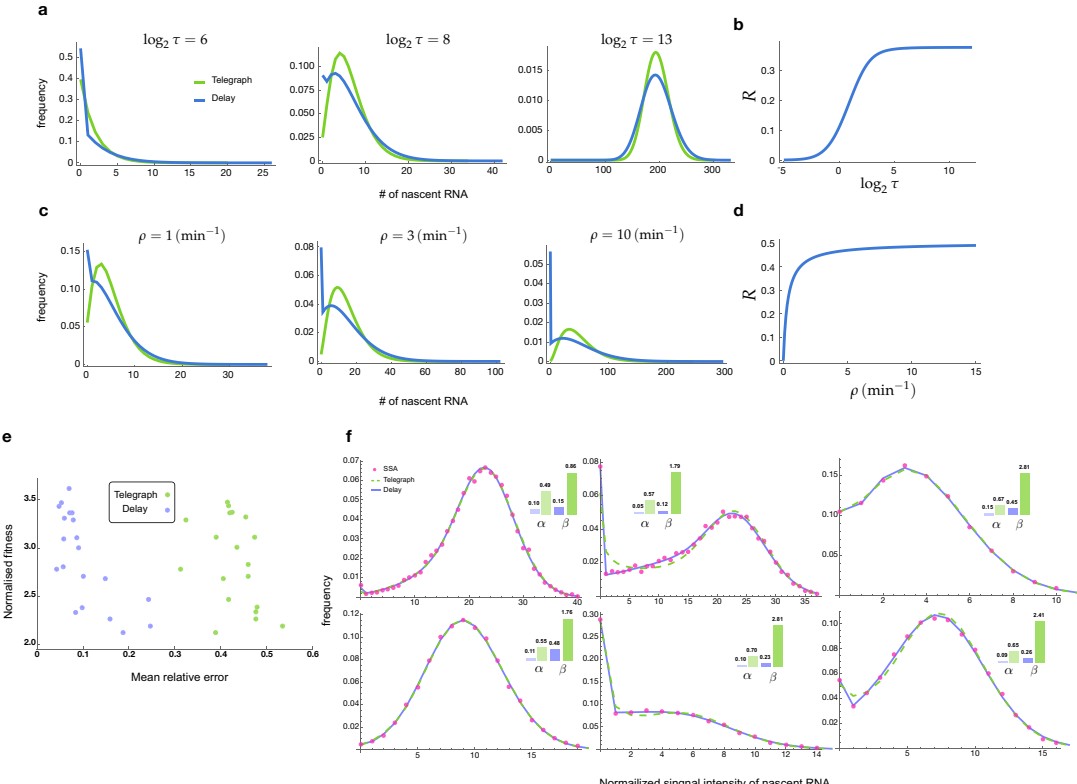

**Appendix 2—figure 1.** Distributions and mean errors of the transcriptional parameter inference. (**a-d**) Comparison of the stochastic properties of the delay telegraph model and the telegraph model. a. Distributions of the nascent mRNA predicted by the delay telegraph model and the telegraph model for various values of $\tau$. We fix the parameters $(\sigma_{\text{off}}, \sigma_{\text{on}}, \rho) = (0.6, 0.03, 1)$ which implies that the change of $\tau$ leads to a change in $x = \tau(\sigma_{\text{on}} + \sigma_{\text{off}})$ at constant $y = (\rho/\sigma_{\text{off}})(\sigma_{\text{off}}/(\sigma_{\text{off}} + \sigma_{\text{on}}))^2$. (**b**) Corresponding relative error $R$ between the variances of two models calculated as a function of $\tau$ using *Equation (12)*. (**c**) Distributions of the nascent mRNA predicted by the delay telegraph model and the telegraph model for various values of $\rho$. We fix the parameters $(\sigma_{\text{off}}, \sigma_{\text{on}}, \tau) = (0.6, 0.03, 100)$ which implies that the change of $\rho$ leads to a change in $y$ at constant $x$. (**d**) Corresponding relative error $R$ between the variances of two models calculated as a function of $\rho$ using *Equation (12)*. (**e-f**) Inference of transcriptional parameters using as input synthetic fluorescent signal data generated by SSA simulations of transcription and fluorescent tagging for $10^4$ cells (see Methods section of the main text). (**e**) Mean relative error and normalised fitness score (fitness/number of samples) plot for 20 sets of numerical experiments. The inference is done in two different ways, using either the telegraph model (green) or the delayed telegraph model (blue). (**f**) Distributions of total fluorescent intensity from synthetic data (red dots) fit using the inference algorithm with telegraph model (dashed green) or delayed telegraph model (blue) for 6 different parameter sets. The insets show the relative errors in the estimates of the burst frequency ($\alpha$) and of the burst size ($\beta$) calculated using Equation (13). Note that while both models provide a very good fit to the distribution from synthetic data, nevertheless parameter estimation is far more accurate using a delayed telegraph model. This is also reflected in (**a**) where we see low fitness scores for both models but a high mean relative error for estimates based on the telegraph model. The true and estimated parameters are shown in *Appendix 2—table 1*.

**Appendix 2—table 1.** Estimates using the inference algorithm with delay telegraph and telegraph models for the six parameter sets in *Appendix 2—figure 1f*.

| | True | | | | | Delay | | | | | Telegraph | | | | |
|---|---|---|---|---|---|---|---|---|---|---|---|---|---|---|---|
| Parameter | $\sigma_{\text{off}}$ | $\sigma_{\text{on}}$ | $\rho$ | burst size | $f_{\text{ON}}$ | $\sigma_{\text{off}}$ | $\sigma_{\text{on}}$ | $\rho$ | burst size | $f_{\text{ON}}$ | $\sigma_{\text{off}}$ | $\sigma_{\text{on}}$ | $\rho$ | burst size | $f_{\text{ON}}$ |
| Set 1 | 1.05 | 8.20 | 57.99 | 55.09 | 0.89 | 0.94 | 7.19 | 57.92 | 61.87 | 0.88 | 0.60 | 4.10 | 59.51 | 99.42 | 0.87 |
| Set 2 | 1.27 | 3.14 | 58.17 | 45.69 | 0.71 | 1.13 | 2.91 | 58.01 | 51.16 | 0.72 | 0.45 | 1.22 | 59.43 | 133.07 | 0.73 |
| Set 3 | 2.27 | 5.80 | 12.55 | 5.53 | 0.72 | 1.60 | 4.48 | 12.21 | 7.63 | 0.74 | 0.59 | 1.71 | 12.09 | 20.61 | 0.74 |

*Appendix 2—table 1 Continued on next page*

*Appendix 2—table 1 Continued*

| Parameter | True | | | | | Delay | | | | | Telegraph | | | | |
|---|---|---|---|---|---|---|---|---|---|---|---|---|---|---|---|
| | $\sigma_{\text{off}}$ | $\sigma_{\text{on}}$ | $\rho$ | burst size | $f_{\text{ON}}$ | $\sigma_{\text{off}}$ | $\sigma_{\text{on}}$ | $\rho$ | burst size | $f_{\text{ON}}$ | $\sigma_{\text{off}}$ | $\sigma_{\text{on}}$ | $\rho$ | burst size | $f_{\text{ON}}$ |
| Set 4 | 1.18 | 9.94 | 24.92 | 21.15 | 0.89 | 0.94 | 8.74 | 24.74 | 26.37 | 0.90 | 0.46 | 4.12 | 24.92 | 54.00 | 0.90 |
| Set 5 | 2.26 | 2.14 | 18.64 | 8.26 | 0.49 | 1.77 | 1.76 | 18.28 | 10.36 | 0.50 | 0.54 | 0.58 | 17.60 | 32.59 | 0.52 |
| Set 6 | 1.38 | 4.77 | 21.74 | 15.79 | 0.78 | 1.08 | 4.00 | 21.50 | 19.94 | 0.79 | 0.38 | 1.49 | 21.31 | 56.44 | 0.80 |

## Appendix 3

## Inference results using mature mRNA data
### Influence of segmentation on inference
In main text Section Inference from nascent mRNA data: cell cycle effects, experimental artifacts and comparison with mature mRNA inference, we used independent segmentation tools to study segmentation artifacts. The inference results under segmentation 1, segmentation 2 and segmentation 2 without counting the transcriptional site are summarised in *Appendix 3—table 1*.

**Appendix 3—table 1.** Inferred transcriptional parameters using merged mature mRNA data from segmentation 1, segmentation 2 and segmentation 2 without transcriptional site (TS).

| Parameter | segmentation 1 | | | | | segmentation 2 | | | | | segmentation 2 without TS | | | | |
| | $\sigma_{\text{off}}$ | $\sigma_{\text{on}}$ | $\rho$ | burst size | $f_{\text{ON}}$ | $\sigma_{\text{off}}$ | $\sigma_{\text{on}}$ | $\rho$ | burst size | $f_{\text{ON}}$ | $\sigma_{\text{off}}$ | $\sigma_{\text{on}}$ | $\rho$ | burst size | $f_{\text{ON}}$ |
|---|---|---|---|---|---|---|---|---|---|---|---|---|---|---|---|
| Set 1 | 21.18 | 4.73 | 47.15 | 2.23 | 0.18 | 3.27 | 3.36 | 20.62 | 6.31 | 0.51 | 2.92 | 2.45 | 21.05 | 7.22 | 0.46 |
| Set 2 | 19.53 | 5.44 | 40.11 | 2.05 | 0.22 | 3.22 | 3.87 | 19.17 | 5.95 | 0.55 | 2.46 | 2.68 | 18.35 | 7.46 | 0.52 |
| Set 3 | 16.53 | 4.77 | 39.48 | 2.39 | 0.22 | 3.67 | 4.05 | 20.09 | 5.47 | 0.52 | 3.00 | 2.87 | 19.73 | 6.57 | 0.49 |
| Set 4 | 28.72 | 5.16 | 50.00 | 1.74 | 0.15 | 5.79 | 4.58 | 23.04 | 3.98 | 0.44 | 5.27 | 3.27 | 24.33 | 4.61 | 0.38 |
| Mean | 21.49 | 5.02 | 44.19 | 2.10 | 0.19 | 3.99 | 3.97 | 20.73 | 5.43 | 0.50 | 3.41 | 2.82 | 20.87 | 6.46 | 0.46 |
| Std | 5.19 | 0.34 | 5.21 | 0.28 | 0.03 | 1.06 | 0.43 | 1.43 | 0.89 | 0.04 | 1.26 | 0.35 | 2.56 | 1.29 | 0.06 |

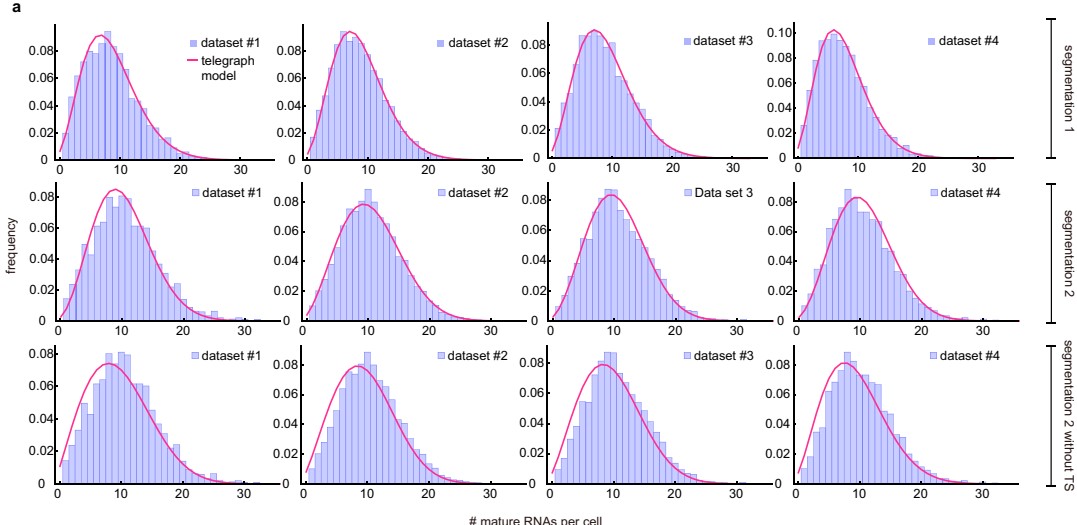

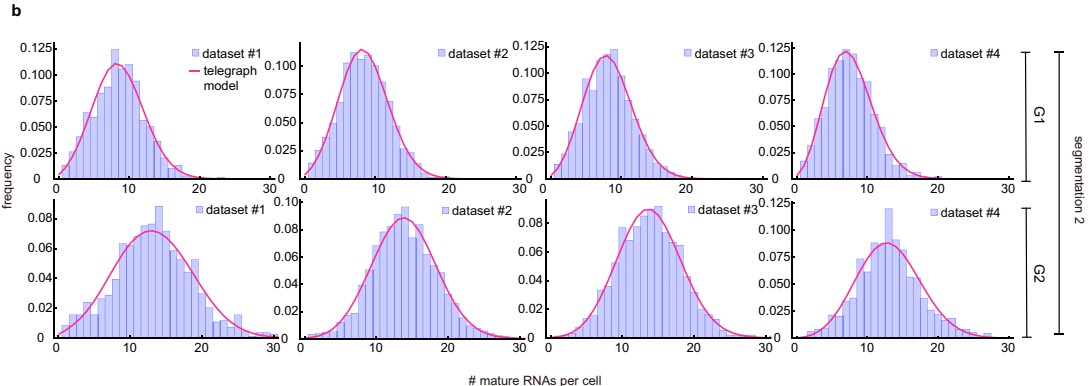

**Appendix 3—figure 1.** Merged and cell-cycle specific mature mRNA count distributions. (**a**) Merged mature mRNA count distribution (purple) under segmentation method 1 and with/without counting the transcriptional site (TS) under segmentation method 2 with a best fit obtained from the telegraph model (magenta curves). (**b**) Cell-

cyle specific mature mRNA count distribution (purple) under segmentation method 2 with a best fit obtained from the telegraph model (magenta curves).

## Confidence intervals for estimates from merged and cell-cycle specific mRNA data using segmentation 2

We obtain 95% confidence intervals for the parameters estimated from mature mRNA data under segmentation 2 (shown in *Figure 3f* of the main text). The confidence intervals are shown for $\sigma_{\text{off}}, \sigma_{\text{on}}, \rho$ in *Appendix 3—table 2*.

**Appendix 3—table 2.** 95% confidence interval intervals for the estimates from experimental mature mRNA data using segmentation 2.

| mature | | $\sigma_{\text{off}}$ | $\sigma_{\text{on}}$ | $\rho$ | CI for $\sigma_{\text{off}}, \sigma_{\text{on}}, \rho$ | | |
|---|---|---|---|---|---|---|---|
| | Set 1 | 3.27 | 3.36 | 20.62 | (2.08, 6.08) | (2.81, 4.16) | (18.03, 25.97) |
| | Set 2 | 3.22 | 3.87 | 19.17 | (2.38, 4.69) | (3.41, 4.44) | (17.51, 21.62) |
| | Set 3 | 3.67 | 4.05 | 20.09 | (2.54, 5.92) | (3.51, 4.79) | (18.03, 23.67) |
| merged | Set 4 | 5.79 | 4.58 | 23.04 | (3.33, 13.33) | (3.79, 5.85) | (19.36, 33.51) |
| | Set 1 | 0.69 | 2.76 | 10.71 | (0.27, 2.35) | (1.73, 4.96) | (9.77, 12.85) |
| | Set 2 | 0.60 | 2.80 | 10.27 | (0.35, 1.24) | (2.13, 3.92) | (9.77, 11.15) |
| | Set 3 | 0.88 | 3.41 | 10.33 | (0.38, 2.84) | (2.23, 5.77) | (9.58, 12.35) |
| G1 | Set 4 | 2.46 | 5.21 | 11.12 | (0.46, 300.00) | (2.68, 18.86) | (8.96, 216.67) |
| | Set 1 | 0.42 | 0.99 | 9.45 | (0.22, 0.91) | (0.65, 1.48) | (8.64, 10.86) |
| | Set 2 | 0.14 | 1.01 | 7.92 | (0.05, 0.36) | (0.56, 1.72) | (7.61, 8.46) |
| | Set 3 | 0.18 | 1.24 | 7.91 | (0.05, 0.72) | (0.57, 2.59) | (7.49, 8.85) |
| G2 | Set 4 | 0.42 | 1.68 | 8.19 | (0.12, 2.07) | (0.76, 3.50) | (7.41, 10.28) |

## Effect of synchronization of gene copies in G2 phase on parameter inference

In the main text, we assumed that the transcription of two allele copies in the G2 phase are independent. Here we consider the opposite scenario where the two allele copies in G2 phase are perfectly synchronized with each other, i.e. when one copy switches from on to off, the other copy also switches from on to off. Note that the time at which mRNA is transcribed from each copy is however not the same. Using this modified simulation algorithm, we re-perform the inference using the experimental data under segmentation 2. The results are shown in *Appendix 3—table 3*. Comparing these values with those estimated for phase G2 under the assumption of allele independence (main text *Figure 3f* right panel), we find that $\rho$ and $f_{\text{ON}}$ are very similar but the other parameters vary considerably.

**Appendix 3—table 3.** Inferred transcriptional rate (normalized) per gene copy for the G2 cell cycle phase under the assumption that the two gene states are perfectly synchronized.

| mature | | $\sigma_{\text{off}}$ | $\sigma_{\text{on}}$ | $\rho$ | burst size | $f_{\text{ON}}$ |
|---|---|---|---|---|---|---|
| | Set 1 | 0.62 | 2.17 | 8.45 | 13.69 | 0.78 |
| | Set 2 | 0.37 | 3.26 | 7.72 | 21.05 | 0.90 |
| | Set 3 | 0.67 | 4.42 | 7.92 | 11.85 | 0.87 |
| | Set 4 | 0.55 | 3.45 | 7.55 | 13.62 | 0.86 |
| | Mean | 0.55 | 3.32 | 7.91 | 15.05 | 0.85 |
| G2 sync | Std | 0.13 | 0.92 | 0.39 | 4.09 | 0.05 |

## Appendix 4

### Inference results using nascent mRNA data

#### Inference using non-curated data

We show the inference results of the experimental non-curated nascent mRNA data (with and without taking into consideration the cell-cycle) for the four datasets (*Appendix 4—table 1*) and their 95% confidence interval using the profile likelihood estimate (*Appendix 4—table 2*). The best fit distributions for merged and cell-cycle-specific data are shown in *Appendix 4—figure 1*.

**Appendix 4—table 1.** Estimated parameters from the non-curated distribution of the normalized intensity of the brightest nuclear spot (nascent mRNA data) constructed by merging all data or else specific to the cell cycle phases G1 and G2.

The elongation time $\tau$ is estimated to be 0.785 min, based on measurements of the elongation speed.

| nascent | | $\sigma_{\text{off}}$ | $\sigma_{\text{on}}$ | $\rho$ | burst size | $f_{\text{ON}}$ |
|---|---|---|---|---|---|---|
| | Set 1 | 5.24 | 5.07 | 77.46 | 14.78 | 0.49 |
| | Set 2 | 5.58 | 5.13 | 82.11 | 14.71 | 0.48 |
| | Set 3 | 5.11 | 5.17 | 76.09 | 14.90 | 0.50 |
| | Set 4 | 5.96 | 6.13 | 73.23 | 12.30 | 0.51 |
| | Mean | 5.47 | 5.37 | 77.22 | 14.17 | 0.50 |
| merged | Std | 0.38 | 0.50 | 3.70 | 1.25 | 0.01 |
| | Set 1 | 1.11 | 3.76 | 37.83 | 34.10 | 0.77 |
| | Set 2 | 1.53 | 3.94 | 41.33 | 27.06 | 0.72 |
| | Set 3 | 0.95 | 3.23 | 36.79 | 38.56 | 0.77 |
| | Set 4 | 1.28 | 3.76 | 36.04 | 28.09 | 0.75 |
| | Mean | 1.22 | 3.67 | 38.00 | 31.95 | 0.75 |
| G1 | Std | 0.25 | 0.31 | 2.34 | 5.39 | 0.02 |
| | Set 1 | 0.74 | 1.69 | 35.00 | 47.30 | 0.70 |
| | Set 2 | 0.82 | 2.18 | 36.30 | 44.37 | 0.73 |
| | Set 3 | 0.91 | 2.19 | 34.54 | 37.90 | 0.71 |
| | Set 4 | 1.08 | 2.61 | 33.27 | 30.76 | 0.71 |
| | Mean | 0.89 | 2.17 | 34.78 | 40.08 | 0.71 |
| G2 | Std | 0.15 | 0.38 | 1.25 | 7.35 | 0.01 |

**Appendix 4—table 2.** 95% confidence intervals for non-curated data estimated using the profile likelihood method.

| nascent | | $\sigma_{\text{off}}$ | $\sigma_{\text{on}}$ | $\rho$ | burst size | $f_{\text{ON}}$ | CI for $\sigma_{\text{off}}, \sigma_{\text{on}}, \rho$ | | |
|---|---|---|---|---|---|---|---|---|---|
| | Set 1 | 5.24 | 5.07 | 77.46 | 14.78 | 0.49 | (4.42, 6.23) | (4.68, 5.45) | (73.48, 82.92) |
| | Set 2 | 5.58 | 5.13 | 82.11 | 14.71 | 0.48 | (5.05, 6.15) | (4.97, 5.37) | (79.38, 85.53) |
| | Set 3 | 5.11 | 5.17 | 76.09 | 14.90 | 0.50 | (4.53, 5.71) | (4.98, 5.38) | (73.49, 79.64) |
| merged | Set 4 | 5.96 | 6.13 | 73.23 | 12.30 | 0.51 | (5.19, 7.22) | (5.87, 6.57) | (69.34, 78.37) |
| | Set 1 | 1.11 | 3.76 | 37.83 | 34.10 | 0.77 | (0.77, 1.66) | (3.04, 4.63) | (36.26, 39.96) |
| | Set 2 | 1.53 | 3.94 | 41.33 | 27.06 | 0.72 | (1.29, 1.88) | (3.64, 4.40) | (40.18, 42.85) |
| | Set 3 | 0.95 | 3.23 | 36.79 | 38.56 | 0.77 | (0.77, 1.17) | (2.86, 3.60) | (36.05, 37.69) |
| G1 | Set 4 | 1.28 | 3.76 | 36.04 | 28.09 | 0.75 | (0.99, 1.73) | (3.18, 4.34) | (34.68, 37.75) |

*Appendix 4—table 2 Continued on next page*

*Appendix 4—table 2 Continued*

| nascent | | $\sigma_{\text{off}}$ | $\sigma_{\text{on}}$ | $\rho$ | burst size | $f_{\text{ON}}$ | CI for $\sigma_{\text{off}}, \sigma_{\text{on}}, \rho$ | | |
|---|---|---|---|---|---|---|---|---|---|
| | Set 1 | 0.74 | 1.69 | 35.00 | 47.30 | 0.70 | (0.54, 1.02) | (1.36, 2.08) | (33.64, 36.72) |
| | Set 2 | 0.82 | 2.18 | 36.30 | 44.37 | 0.73 | (0.66, 1.07) | (1.85, 2.52) | (35.35, 37.40) |
| | Set 3 | 0.91 | 2.19 | 34.54 | 37.90 | 0.71 | (0.70, 1.23) | (1.85, 2.61) | (33.38, 36.05) |
| G2 | Set 4 | 1.08 | 2.61 | 33.27 | 30.76 | 0.71 | (0.75, 1.71) | (2.00, 3.41) | (31.91, 35.40) |

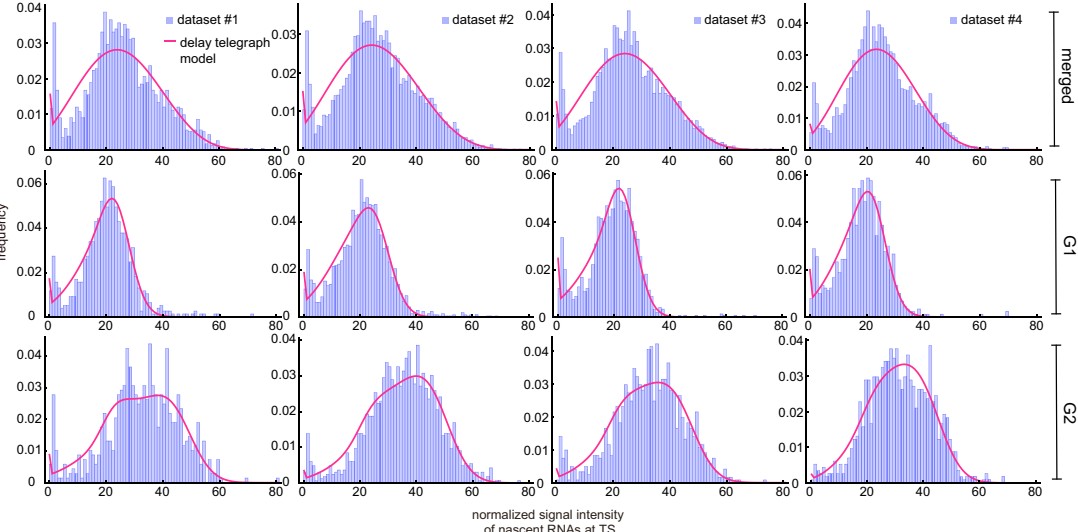

**Appendix 4—figure 1.** Inference results using merged and cell-cycle specific nascent data. Experimental distributions (purple) are fit using the delay telegraph model (magenta curves).

## Inference using curated data

We show the inference results using nascent mRNA data curated with the rejection method in *Appendix 4—table 3* and *Appendix 4—figure 2*, and with the fusion method in *Appendix 4—table 4*. We also used the profile likelihood method to obtain the 95% confidence intervals for the fusion method with $k = 4$; the results are shown in *Appendix 4—table 5*. We also reinferred the parameters in the G2 phase for fusion corrected data by assuming that the allele states are perfectly synchronised; these are shown in *Appendix 4—table 6*.

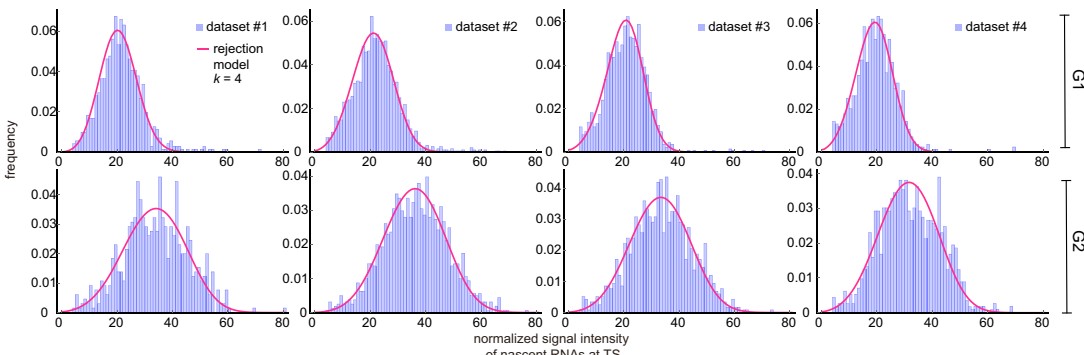

**Appendix 4—figure 2.** Inference results using cell-cycle specific data curated with the rejection method (only the distributions for $k = 4$ are shown). Corresponding fitted distributions (purple) for G1 (top row) and G2 (bottom row) using the delay telegraph model (magenta curves).

**Appendix 4—table 3.** (Rejection method) Estimated parameters by discarding the first $k$ signal bins of the experimental distribution of the signal intensity (and renormalizing afterwards). Inference is done for each of the four data sets. The elongation time is fixed to $\tau \approx 0.785$ min.

| $k = 1$ | | | $\sigma_{\text{off}}$ | $\sigma_{\text{on}}$ | | burst size | $f_{\text{ON}}$ |
|---|---|---|---|---|---|---|---|
| | | Set 1 | 1.06 | 4.33 | 36.62 | 34.63 | 0.80 |
| | | Set 2 | 1.45 | 4.42 | 39.84 | 27.51 | 0.75 |
| | | Set 3 | 1.04 | 3.84 | 36.66 | 35.19 | 0.79 |
| | | Set 4 | 1.27 | 4.12 | 35.41 | 27.78 | 0.76 |
| | | Mean | 1.21 | 4.18 | 37.13 | 31.28 | 0.78 |
| G1 | | Std | 0.19 | 0.26 | 1.90 | 4.20 | 0.02 |
| | | Set 1 | 0.84 | 2.10 | 34.54 | 41.20 | 0.71 |
| | | Set 2 | 0.97 | 2.94 | 35.49 | 36.73 | 0.75 |
| | | Set 3 | 0.94 | 2.53 | 33.68 | 35.99 | 0.73 |
| | | Set 4 | 1.20 | 3.07 | 32.81 | 27.28 | 0.72 |
| | | Mean | 0.99 | 2.66 | 34.13 | 35.30 | 0.73 |
| G2 | | Std | 0.15 | 0.44 | 1.15 | 5.82 | 0.02 |
| $k = 2$ | | | $\sigma_{\text{off}}$ | $\sigma_{\text{on}}$ | $\rho$ | burst size | $f_{\text{ON}}$ |
| | | Set 1 | 1.58 | 6.33 | 37.77 | 23.89 | 0.80 |
| | | Set 2 | 1.93 | 5.93 | 40.86 | 21.16 | 0.75 |
| | | Set 3 | 1.28 | 5.15 | 37.07 | 29.07 | 0.80 |
| | | Set 4 | 1.55 | 5.29 | 35.92 | 23.10 | 0.77 |
| | | Mean | 1.59 | 5.67 | 37.91 | 24.30 | 0.78 |
| G1 | | Std | 0.27 | 0.55 | 2.11 | 3.38 | 0.02 |
| | | Set 1 | 1.43 | 3.42 | 35.83 | 25.11 | 0.71 |
| | | Set 2 | 1.69 | 4.39 | 37.42 | 22.10 | 0.72 |
| | | Set 3 | 1.33 | 3.38 | 34.64 | 26.08 | 0.72 |
| | | Set 4 | 1.56 | 3.68 | 33.70 | 21.61 | 0.70 |
| | | Mean | 1.50 | 3.72 | 35.39 | 23.72 | 0.71 |
| G2 | | Std | 0.16 | 0.47 | 1.61 | 2.20 | 0.01 |
| $k = 3$ | | | $\sigma_{\text{off}}$ | $\sigma_{\text{on}}$ | $\rho$ | burst size | $f_{\text{ON}}$ |
| | | Set 1 | 2.66 | 9.03 | 39.73 | 14.92 | 0.77 |
| | | Set 2 | 2.55 | 7.41 | 42.04 | 16.46 | 0.74 |
| | | Set 3 | 1.64 | 6.67 | 37.69 | 22.92 | 0.80 |
| | | Set 4 | 2.22 | 7.32 | 37.02 | 16.65 | 0.77 |
| | | Mean | 2.27 | 7.61 | 39.12 | 17.73 | 0.77 |
| G1 | | Std | 0.46 | 1.00 | 2.26 | 3.54 | 0.02 |

*Appendix 4—table 3 Continued on next page*

*Appendix 4—table 3 Continued*

| k = 1 | | $\sigma_{\text{off}}$ | $\sigma_{\text{on}}$ | | **burst size** | $f_{\text{ON}}$ |
|---|---|---|---|---|---|---|
| | Set 1 | 2.01 | 4.35 | 37.23 | 18.50 | 0.68 |
| | Set 2 | 2.16 | 5.10 | 38.56 | 17.85 | 0.70 |
| | Set 3 | 1.70 | 4.05 | 35.55 | 20.85 | 0.70 |
| | Set 4 | 1.88 | 4.16 | 34.48 | 18.30 | 0.69 |
| | Mean | 1.94 | 4.41 | 36.45 | 18.88 | 0.69 |
| G2 | Std | 0.19 | 0.48 | 1.80 | 1.34 | 0.01 |

| k = 4 | | $\sigma_{\text{off}}$ | $\sigma_{\text{on}}$ | $\rho$ | burst size | $f_{\text{ON}}$ |
|---|---|---|---|---|---|---|
| | Set 1 | 6.10 | 14.18 | 44.41 | 7.28 | 0.70 |
| | Set 2 | 3.72 | 9.57 | 43.96 | 11.81 | 0.72 |
| | Set 3 | 2.02 | 7.99 | 38.25 | 18.89 | 0.80 |
| | Set 4 | 2.75 | 8.60 | 37.78 | 13.76 | 0.76 |
| | Mean | 3.65 | 10.09 | 41.10 | 12.94 | 0.74 |
| G1 | Std | 1.77 | 2.80 | 3.57 | 4.81 | 0.04 |
| | Set 1 | 2.12 | 4.50 | 37.48 | 17.68 | 0.68 |
| | Set 2 | 2.59 | 5.68 | 39.55 | 15.25 | 0.69 |
| | Set 3 | 2.47 | 5.14 | 37.27 | 15.08 | 0.68 |
| | Set 4 | 2.18 | 4.55 | 35.17 | 16.13 | 0.68 |
| | Mean | 2.34 | 4.97 | 37.37 | 16.04 | 0.68 |
| G2 | Std | 0.23 | 0.56 | 1.79 | 1.19 | 0.01 |

**Appendix 4—table 4.** (Fusion method) Estimated parameters by combining the first $k$ signal bins of the experimental distribution of the signal intensity.
Inference is done for each of the four data sets. The elongation time is fixed to $\tau \approx 0.785$ min.

| k = 1 | | $\sigma_{\text{off}}$ | $\sigma_{\text{on}}$ | $\rho$ | **burst size** | $f_{\text{ON}}$ |
|---|---|---|---|---|---|---|
| | Set 1 | 1.11 | 3.76 | 37.83 | 34.10 | 0.77 |
| | Set 2 | 1.53 | 3.94 | 41.33 | 27.06 | 0.72 |
| | Set 3 | 0.95 | 3.23 | 36.79 | 38.56 | 0.77 |
| | Set 4 | 1.28 | 3.76 | 36.04 | 28.09 | 0.75 |
| | Mean | 1.22 | 3.67 | 38.00 | 31.95 | 0.75 |
| G1 | Std | 0.25 | 0.31 | 2.34 | 5.39 | 0.02 |
| | Set 1 | 0.74 | 1.69 | 35.00 | 47.30 | 0.70 |
| | Set 2 | 0.82 | 2.18 | 36.30 | 44.37 | 0.73 |
| | Set 3 | 0.91 | 2.19 | 34.54 | 37.90 | 0.71 |
| | Set 4 | 1.08 | 2.61 | 33.27 | 30.76 | 0.71 |
| | Mean | 0.89 | 2.17 | 34.78 | 40.08 | 0.71 |
| G2 | Std | 0.15 | 0.38 | 1.25 | 7.35 | 0.01 |

| k = 2 | | $\sigma_{\text{off}}$ | $\sigma_{\text{on}}$ | $\rho$ | burst size | $f_{\text{ON}}$ |
|---|---|---|---|---|---|---|

*Appendix 4—table 4 Continued on next page*

*Appendix 4—table 4 Continued*

| $k = 1$ | | | $\sigma_{\text{off}}$ | $\sigma_{\text{on}}$ | $\rho$ | **burst size** | $f_{\text{ON}}$ |
|---|---|---|---|---|---|---|---|
| | | Set 1 | 0.78 | 2.99 | 36.65 | 46.99 | 0.79 |
| | | Set 2 | 1.14 | 3.27 | 40.01 | 35.06 | 0.74 |
| | | Set 3 | 0.71 | 2.58 | 36.00 | 51.00 | 0.79 |
| | | Set 4 | 0.96 | 3.09 | 35.08 | 36.52 | 0.76 |
| | | Mean | 0.90 | 2.98 | 36.94 | 42.39 | 0.77 |
| G1 | | Std | 0.19 | 0.30 | 2.15 | 7.82 | 0.02 |
| | | Set 1 | 0.54 | 1.30 | 34.37 | 63.20 | 0.70 |
| | | Set 2 | 0.67 | 1.87 | 35.74 | 53.70 | 0.74 |
| | | Set 3 | 0.74 | 1.88 | 33.96 | 45.75 | 0.72 |
| | | Set 4 | 0.94 | 2.36 | 32.84 | 34.89 | 0.72 |
| | | Mean | 0.72 | 1.85 | 34.23 | 49.38 | 0.72 |
| G2 | | Std | 0.17 | 0.44 | 1.20 | 12.01 | 0.01 |
| $k = 3$ | | | $\sigma_{\text{off}}$ | $\sigma_{\text{on}}$ | $\rho$ | burst size | $f_{\text{ON}}$ |
| | | Set 1 | 0.71 | 2.79 | 36.38 | 51.39 | 0.80 |
| | | Set 2 | 1.07 | 3.14 | 39.77 | 37.01 | 0.75 |
| | | Set 3 | 0.63 | 2.35 | 35.74 | 56.84 | 0.79 |
| | | Set 4 | 0.80 | 2.71 | 34.58 | 43.09 | 0.77 |
| | | Mean | 0.80 | 2.75 | 36.62 | 47.08 | 0.78 |
| G1 | | Std | 0.19 | 0.32 | 2.23 | 8.78 | 0.02 |
| | | Set 1 | 0.52 | 1.25 | 34.30 | 65.86 | 0.71 |
| | | Set 2 | 0.65 | 1.84 | 35.70 | 54.75 | 0.74 |
| | | Set 3 | 0.71 | 1.81 | 33.85 | 47.67 | 0.72 |
| | | Set 4 | 0.91 | 2.31 | 32.76 | 35.91 | 0.72 |
| | | Mean | 0.70 | 1.80 | 34.15 | 51.05 | 0.72 |
| G2 | | Std | 0.16 | 0.43 | 1.22 | 12.57 | 0.01 |
| $k = 4$ | | | $\sigma_{\text{off}}$ | $\sigma_{\text{on}}$ | $\rho$ | burst size | $f_{\text{ON}}$ |
| | | Set 1 | 0.69 | 2.73 | 36.30 | 52.85 | 0.80 |
| | | Set 2 | 1.05 | 3.09 | 39.69 | 37.71 | 0.75 |
| | | Set 3 | 0.63 | 2.35 | 35.74 | 56.78 | 0.79 |
| | | Set 4 | 0.83 | 2.79 | 34.68 | 41.57 | 0.77 |
| | | Mean | 0.80 | 2.74 | 36.60 | 47.23 | 0.78 |
| G1 | | Std | 0.19 | 0.31 | 2.17 | 9.04 | 0.02 |
| | | Set 1 | 0.56 | 1.34 | 34.45 | 61.05 | 0.70 |
| | | Set 2 | 0.66 | 1.86 | 35.73 | 54.11 | 0.74 |
| | | Set 3 | 0.67 | 1.72 | 33.70 | 50.56 | 0.72 |
| | | Set 4 | 0.92 | 2.33 | 32.79 | 35.48 | 0.72 |
| | | Mean | 0.70 | 1.81 | 34.17 | 50.30 | 0.72 |
| G2 | | Std | 0.15 | 0.41 | 1.24 | 10.80 | 0.01 |

**Appendix 4—table 5.** Inference of the kinetic parameters $(\sigma_{\mathrm{off}}, \sigma_{\mathrm{on}}, \rho)$ using nascent data curated with the fusion method.

Inferred values and the corresponding 95% confidence intervals of G1 and G2 cell-cycle-specific data calculated using the profile likelihood method.

| nascent | | $\sigma_{\mathrm{off}}$ | $\sigma_{\mathrm{on}}$ | $\rho$ | burst size | $f_{\mathrm{ON}}$ | CI for $\sigma_{\mathrm{off}}, \sigma_{\mathrm{on}}, \rho$ | | |
|---|---|---|---|---|---|---|---|---|---|
| | Set 1 | 0.69 | 2.73 | 36.30 | 52.85 | 0.80 | (0.46, 1.05) | (2.12, 3.52) | (35.06, 37.91) |
| | Set 2 | 1.05 | 3.09 | 39.69 | 37.71 | 0.75 | (0.86, 1.29) | (2.73, 3.47) | (38.77, 40.92) |
| | Set 3 | 0.63 | 2.35 | 35.74 | 56.78 | 0.79 | (0.49, 0.79) | (1.99, 2.71) | (35.06, 36.58) |
| G1 | Set 4 | 0.83 | 2.79 | 34.68 | 41.57 | 0.77 | (0.62, 1.16) | (2.27, 3.41) | (33.54, 36.01) |
| | Set 1 | 0.56 | 1.34 | 34.45 | 61.05 | 0.70 | (0.40, 0.81) | (1.04, 1.73) | (33.35, 35.82) |
| | Set 2 | 0.66 | 1.86 | 35.73 | 54.11 | 0.74 | (0.51, 0.85) | (1.57, 2.21) | (34.87, 36.77) |
| | Set 3 | 0.67 | 1.72 | 33.70 | 50.56 | 0.72 | (0.50, 0.91) | (1.40, 2.11) | (32.78, 34.87) |
| G2 | Set 4 | 0.92 | 2.33 | 32.79 | 35.48 | 0.72 | (0.62, 1.45) | (1.74, 3.09) | (31.65, 34.68) |

**Appendix 4—table 6.** Inferred transcriptional rate (normalized) per gene copy for the G2 cell cycle phase using nascent data curated with the fusion method under the assumption that the two gene states are perfectly synchronized.

| nascent fusion | | $\sigma_{\mathrm{off}}$ | $\sigma_{\mathrm{on}}$ | $\rho$ | burst size | $f_{\mathrm{ON}}$ |
|---|---|---|---|---|---|---|
| | Set 1 | 1.84 | 4.42 | 33.84 | 18.42 | 0.71 |
| | Set 2 | 2.18 | 5.78 | 35.85 | 16.44 | 0.73 |
| | Set 3 | 2.15 | 5.36 | 33.61 | 15.63 | 0.71 |
| | Set 4 | 3.53 | 7.45 | 34.16 | 9.69 | 0.68 |
| | Mean | 2.42 | 5.75 | 34.36 | 15.05 | 0.71 |
| G2 sync | Std. | 0.75 | 1.26 | 1.01 | 3.76 | 0.02 |

# Appendix 5

## Calculating the autocorrelation function (ACF) from stochastic simulations

Intensity traces of *GAL10* transcription were generated by a stochastic simulation algorithm (SSA) following the delay telegraph model presented in the main manuscript. Briefly, for each yeast cell, the ON and OFF *GAL10* promoter states are simulated such that they have an exponentially distributed lifetime with means $1/\sigma_{\text{off}}$ and $1/\sigma_{\text{off}}$, respectively. When the promoter was ON, the transcription of single *GAL10* mRNAs was initiated following a Poisson process with mean equal to the initiation rate $\rho$. The fluorescence trace $I(t)$, as measured experimentally over time, was obtained by convolving the train of mRNA initiation events with the trapezoidal intensity pulse of the PP7 tagging system (shown in *Figure 1c* of the main text). It is important to note that this last step is necessary to accurately model the fluorescent intensities that are experimentally measured.

To account for cell cycle, we simulated an asynchronous population, where cells could switch from G1 to G2. In the latter, to account for two independent transcription sites, two independent intensity traces were generated and summed over time. The ratio of G1 to G2 cells was 1.15:1, 1.27:1, 1,24:1 and 0.80:1 for datasets 1, 2, 3, and 4, respectively, as determined from the smFISH experiments. It is important to note that: (i) live-cell experiments (as in *Donovan et al., 2019*) are performed in asynchronous populations, without any sorting based on cell cycle phase; (ii) the average G1 phase duration for mother and daughter cells ($t_{\text{mother}} = 2400\,s$ and $t_{\text{daughter}} = 6480\,s$, respectively – unpublished data) is on the order of the typical total acquisition time (e.g. $1800\,s$). Hence, in our simulations, cells that were G1 at the start of the intensity trace (i.e. live-cell experiment) were allowed to switch to G2 before the end of the experiment. Trivially, cells that were G2 at the start remained so until the end of the experiment.

The autocorrelation function (ACF) was calculated as:

$$G(\Delta t) = \frac{\langle \delta I(t) \delta I(t+\Delta t) \rangle}{\langle I(t) \rangle^2} - 1, \tag{14}$$

where $\Delta t$ is the time-lag (e.g. $30\,s$ in *Donovan et al., 2019*), $\langle \cdot \rangle$ denotes the temporal average over $t$, and $\delta I(t) = I(t) - \langle I(t) \rangle$. *Appendix 5—figure 1* shows the ACFs from *Figure 6c* of the main manuscript, in which a linear fit is performed to correct for non-stationary or slowly-varying effects, i.e. switching from G1 to G2 during the experiment, bleaching, heterogeneous galactose induction.

*Appendix 5—figure 1 continued on next page*

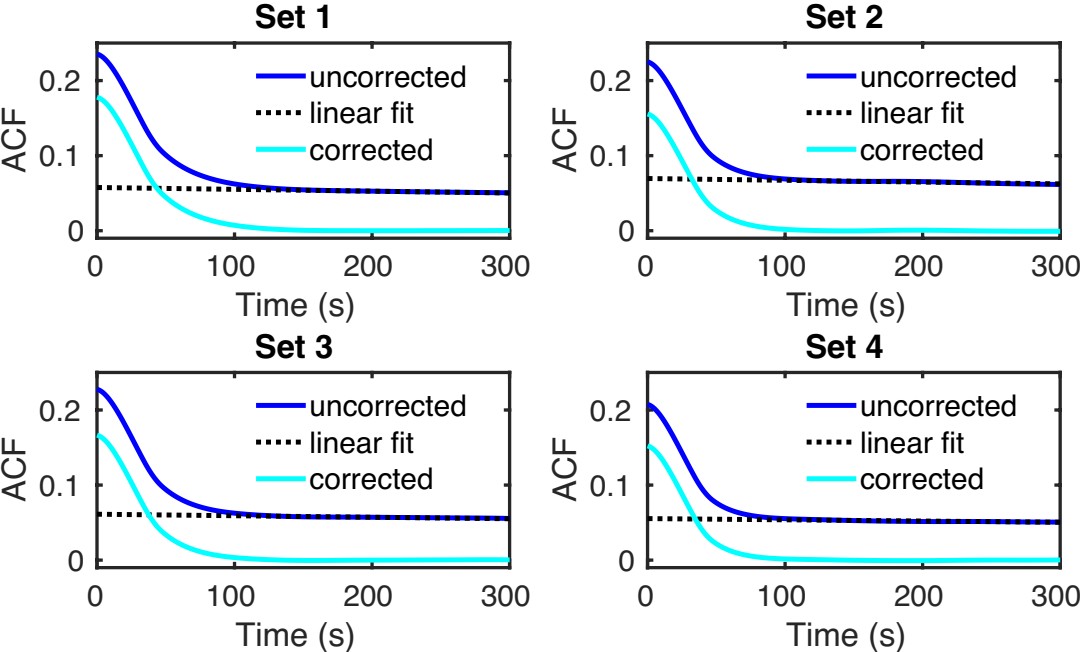

**Appendix 5—figure 1.** Autocorrelation functions of $10^4$ simulated *GAL10* intensity traces (solid blue lines). The transcriptional parameters for G1 and G2 cells in the four sets of experimental data were obtained using the fusion method (see *Figure 6d* of the main manuscript). A linear fit (dashed black line) was subtracted to correct the ACFs for switching from G1 to G2 (solid cyan lines).

# Appendix 6

## Derivation of the exact solution for the master equation of the delay telegraph model and finite state projection

### Exact solution

The delay telegraph model includes four reactions

$$G \xrightarrow{\rho} G + N, \quad G \xrightarrow{\sigma_{\text{off}}} G^\star, \quad G^\star \xrightarrow{\sigma_{\text{on}}} G, \quad N \underset{\tau}{\Rightarrow} \varnothing, \tag{15}$$

where $G$ and $G^\star$ stand for the active (ON) and inactive (OFF) gene state, respectively, and $\sigma_{\text{off}}$ and $\sigma_{\text{on}}$ are the activation and inactivation rates, respectively. Once a nascent mRNA is produced, it will be removed from the system after a deterministic time $\tau$. We aim to find the quantitative description of the kinetics of nascent mRNA ($N$). We proceed by deriving the delay chemical master equation (CME) of *Equation (15)*. The same delay models have also been studied by other authors .

Let $P(0, n, t)$ and $P(1, n, t)$ be the probability of observing $n$ nascent mRNAs while the gene state is inactive and active at time $t$, respectively. Consequently, we can state

$$\begin{aligned} P(i, n, t + \Delta t) \quad &= \{\text{Part A: Probability of one instant reaction occurring during } \Delta t\} \\ &+ \{\text{Part B: Probability of one delayed reaction occurring during } \Delta t\} \\ &+ \{\text{Part C: Probability of no reactions occurring during } \Delta t\}. \end{aligned} \tag{16}$$

Note that by instant reactions, we mean all reactions except the delayed reaction that removes nascent mRNA. Since Part A includes all Markovian reactions, it follows from standard arguments *Gillespie, 2007* that

$$\begin{aligned} P(0, n, t + \Delta t; 1, n, t) \quad &= \sigma_{\text{off}} \Delta t P(1, n, t) + o(\Delta t), \\ P(1, n, t + \Delta t; 0, n, t) \quad &= \sigma_{\text{on}} \Delta t P(0, n, t) + o(\Delta t), \\ P(1, n, t + \Delta t; 1, n - 1, t) \quad &= \rho \Delta t P(1, n - 1, t) + o(\Delta t). \end{aligned} \tag{17}$$

The contribution of Part B requires a careful consideration of the history of the process: (i) $(1, n', t - \tau)$ leads to $(1, n' + 1, t - \tau + \Delta t)$ when the gene was ON; (ii) $(1, n' + 1, t - \tau + \Delta t)$ leads to $(i, n + 1, t)$ where the gene state $i$ can be either OFF or ON; (iii) $(i, n + 1, t)$ leads to $(i, n, t)$. Indeed, since the removal of nascent mRNA is a delayed reaction, (iii) is due to a production reaction in (i) a time interval $\tau$ earlier. The probability of (i) occurring is $\rho \Delta t P(1, n', t - \tau)$. The probability of (iii) occurring is 1 since every production event is followed by a removal event a time $\tau$ later. The probability of (ii) occurring is:

$$P(i, n\mathbf{+1}, t | 1, n'\mathbf{+1}, t - \tau + \Delta t) \tag{18}$$

As for *Equation (18)*, the two '+1's means that the new nascent mRNA was produced during the time interval $(t - \tau, t - \tau + \Delta t)$, and it did not participate in any other reactions, thereby not influencing the probability at time $t + \Delta t$. In short,

$$P(i, n\mathbf{+1}, t | 1, n'\mathbf{+1}, t - \tau + \Delta t) = P(i, n, t | 1, n', t - \tau + \Delta t). \tag{19}$$

Furthermore, all $n'$ nascent mRNAs must leave the system prior to time $t$ as they were all born before time $t - \tau$. This implies that

$$P(i, n, t | 1, n', t - \tau + \Delta t) = P(i, n, t | 1, 0, t - \tau + \Delta t). \tag{20}$$

Hence the probability that event B occurs is given by the product of the probability of events (i), (ii) and (iii) and summing over all values of $n'$

$$\begin{aligned} P(i, n, t + \Delta t; i, n + 1, t) \quad &= \rho \Delta t \sum_{n'} P(1, n', t - \tau) P(i, n, t | 1, 0, t - \tau + \Delta t) \\ &= \rho \Delta t P(1, t - \tau) P(i, n, t | 1, 0, t - \tau + \Delta t), \end{aligned} \tag{21}$$

where we used $\sum_{n'} P(1, n', t - \tau) = P(1, t - \tau)$.

Finally, the contribution of Part C is obtained from simple probability arguments

$$\begin{aligned}
P(0,n,t;0,n,t) &= P(0,n,t) - P(1,n,t;0,n,t) - P(0,n-1,t;0,n,t) \\
&= P(0,n,t) - \sigma_{\text{on}}\Delta t P(0,n,t) - \rho\Delta t P(1,t-\tau)P(0,n-1,t|1,0,t-\tau+\Delta t), \\
P(1,n,t;1,n,t) &= P(1,n,t) - P(0,n,t;1,n,t) - P(1,n+1,t;1,n,t) - P(1,n-1,t;1,n,t) \\
&= P(1,n,t) - \sigma_{\text{off}}\Delta t P(1,n,t) - \rho\Delta t P(1,n,t) - \rho(1,t-\tau)P(1,n-1,t|1,0,t-\tau+\Delta t),
\end{aligned}$$

$$(22)$$

where we used the same argument as in **Equation (21)**. Using **Equations 16; 17; 21; 22** and taking the limit of small $\Delta t$, we finally obtain the set of delay CMEs

$$\begin{cases}
\frac{dP(0,n,t)}{dt} = -\sigma_{\text{on}}P(0,n,t) + \sigma_{\text{off}}P(1,n,t) \quad + (E-1)P(0,n-1,t|1,0,t-\tau)\rho P(1,t-\tau), \\
\frac{dP(1,n,t)}{dt} = \sigma_{\text{on}}P(0,n,t) - \sigma_{\text{off}}P(1,n,t) + \rho(E^{-1}-1)P(1,n,t) \quad + (E-1)P(1,n-1,t|1,0,t-\tau)\rho P(1,t-\tau)
\end{cases}$$

$$(23)$$

where $E^k P(i,n,t) = P(i,n+k,t)$ is the step operator. Summing over all possible $n$ for the two equations in **Equation (23)**, we obtain

$$\begin{cases}
\frac{dP(0,t)}{dt} = -\sigma_{\text{on}}P(0,t) + \sigma_{\text{off}}P(1,t), \\
\frac{dP(1,t)}{dt} = \sigma_{\text{on}}P(0,t) - \sigma_{\text{off}}P(1,t),
\end{cases}$$

initiated with the inactive state, in which $P(0,t)$ and $P(1,t)$ are the probabilities of finding a cell at the inactivated and activated state at time $t$, and their solutions are

$$P(0,t) = \frac{\sigma_{\text{off}}}{\sigma_{\text{off}}+\sigma_{\text{on}}} + \frac{\sigma_{\text{on}}}{\sigma_{\text{off}}+\sigma_{\text{on}}}e^{-(\sigma_{\text{on}}+\sigma_{\text{off}})t}, \quad P(1,t) = \frac{\sigma_{\text{on}}}{\sigma_{\text{off}}+\sigma_{\text{on}}}\left(1 - e^{-(\sigma_{\text{on}}+\sigma_{\text{off}})t}\right).$$

Besides, it is noted that in **Equation (23)**

$$P(1,t-\tau) = \frac{\sigma_{\text{on}}}{\sigma_{\text{off}}+\sigma_{\text{on}}}\left(1 - e^{-(\sigma_{\text{on}}+\sigma_{\text{off}})(t-\tau)}\right) =: h(t-\tau), \qquad (24)$$

for $t \geq \tau$, and $h(t-\tau) = 0$ if $t < \tau$. The $n$ nascent mRNAs of the two conditional probabilities $P(i,n,t|1,0,t-\tau)$ for $i = 0,1$ in **Equation (23)** are produced during time $(t-\tau,t)$, and the pertinent dynamics are that of a reaction system only composed of the three instant reactions in **Equation (15)**. Specifically, we have

$$\begin{cases}
\frac{d\widetilde{P}(0,n,t)}{dt} = -\sigma_{\text{on}}\widetilde{P}(0,n,t) + \sigma_{\text{off}}\widetilde{P}(1,n,t), \\
\frac{d\widetilde{P}(1,n,t)}{dt} = \sigma_{\text{on}}\widetilde{P}(0,n,t) - \sigma_{\text{off}}\widetilde{P}(1,n,t) + \rho(E^{-1}-1)\widetilde{P}(1,n,t),
\end{cases} \qquad (25)$$

where the initial values are $\widetilde{P}(0,n,0) = 0$ for any $n$, $\widetilde{P}(1,n,0) = 1$ for $n = 0$ and equal to 0 otherwise, as well as $P(i,n,t|1,0,t-\tau) = \widetilde{P}(i,n,\tau)$ for any $i = 0,1$.

Let $G_i(u,t) = \sum_n (u+1)^n P(i,n,t)$ and $\widetilde{G}_i(u,t) = \sum_n (u+1)^n \widetilde{P}(i,n,t)$. We particularly define the generating function in such a form to simplify the notation. Then, using **Equations 23; 24** we obtain

$$\begin{cases}
\partial_t G_0 = -\sigma_{\text{on}}G_0 + \sigma_{\text{off}}G_1 - \rho h(t-\tau)u\mathbb{1}_{[\tau,\infty)}\widetilde{G}_0^\tau, \\
\partial_t G_1 = \rho u G_1 + \sigma_{\text{on}}G_0 - \sigma_{\text{off}}G_1 - \rho h(t-\tau)u\mathbb{1}_{[\tau,\infty)}\widetilde{G}_1^\tau,
\end{cases} \qquad (26)$$

and

$$\begin{cases}
\partial_t \widetilde{G}_0 = -\sigma_{\text{on}}\widetilde{G}_0 + \sigma_{\text{off}}\widetilde{G}_1, \\
\partial_t \widetilde{G}_1 = \rho u \widetilde{G}_1 + \sigma_{\text{on}}\widetilde{G}_0 - \sigma_{\text{off}}\widetilde{G}_1,
\end{cases} \qquad (27)$$

where the arguments $u$ and $t$ in the generating functions are suppressed for clarity, and the superscript $\tau$ is used to emphasize the generating function $\widetilde{G}_i$ up to a particular time $\tau$. The initiation condition of **Equation (27)** is $\widetilde{G}_0 = 0$ and $\widetilde{G}_1 = 1$ when $t = 0$.

Under the condition of steady-state as $t \to \infty$, **Equation (26)** reduces to

$$\begin{cases} 0 = & -\sigma_{\text{on}} G_0 + \sigma_{\text{off}} G_1 - \rho \bar{h} u \widetilde{G}_0^\tau, \\ 0 = & \rho u G_1 + \sigma_{\text{on}} G_0 - \sigma_{\text{off}} G_1 - \rho \bar{h} u \widetilde{G}_1^\tau, \end{cases} \tag{28}$$

with $\bar{h} = \sigma_{\text{on}}/(\sigma_{\text{on}} + \sigma_{\text{off}})$.

Therefore, by solving **Equation (28)** we obtain

$$G(u, \infty) = G_0(u, \infty) + G_1(u, \infty)$$
$$= \frac{e^{-\frac{1}{2}\tau\left(\sqrt{\Delta(u)} + \sigma_{\text{off}} + \sigma_{\text{on}} - u\rho\right)}}{2\sqrt{\Delta(u)}(\sigma_{\text{off}} + \sigma_{\text{on}})}$$
$$\times \left[ (\sigma_{\text{off}} + \sigma_{\text{on}}) \left( \sqrt{\Delta(u)} + e^{\sqrt{\Delta(u)}\tau} \left( \sqrt{\Delta(u)} + \sigma_{\text{off}} + \sigma_{\text{on}} \right) - \sigma_{\text{off}} - \sigma_{\text{on}} \right) - u \left( e^{\sqrt{\Delta(u)}\tau} - 1 \right) \rho \left( \sigma_{\text{off}} - \epsilon \right. \right.$$

$$\tag{29}$$

where we define $\Delta(u) = \left(\sigma_{\text{off}} - u\rho\right)^2 + 2\sigma_{\text{on}}\left(\sigma_{\text{off}} + u\rho\right) + \sigma_{\text{on}}^2$. Finally, we obtain the probability distribution by the following relation

$$P(n, t) = \frac{1}{n!} \partial_u^n G(u, \infty)|_{u=-1}, \quad \text{for all } n \in \mathbb{N}.$$

We checked that the obtained distribution agrees exactly with the nascent mRNA distribution of Equation 7 in **Xu et al., 2016**.

## Finite State Projection

The finite state projection (FSP) method **Munsky and Khammash, 2006** is an efficient numerical method that can be implemented to solve the CME (23) to any desired degree of accuracy. Because **Equation (23)** is coupled with **Equation (25)**, here we show how to use a two-step FSP method to obtain the probability distribution.

By assuming that there exists a large $N$ such that $P(i, n, t) = 0$ and $\tilde{P}(i, n, t) = 0$ for any $i = 0, 1$ and $n \geq N + 1$, we denote

$$\mathbf{P}(t) = (P(0,0,t), P(0,1,t), \ldots, P(0,N,t), P(1,0,t), P(1,1,t), \ldots, P(1,N,t))^T$$

as a $N^2$-dimensional vector and $\tilde{\mathbf{P}}(t) \in \mathbb{R}^{N^2}$ similarly. We first use FSP method to solve **Equation (25)** with the following ordinary differential equation (ODE) up to time

$$\frac{\mathrm{d}}{\mathrm{d}t} \mathbf{P}(t) = \mathbf{A}\mathbf{P}(t) \tag{30}$$

where $\mathbf{A}$ is defined as

$$\mathbf{A} = \begin{pmatrix} -\sigma_{\text{on}} \mathbf{I}_N & 0 \\ 0 & \sigma_{\text{off}} \mathbf{I}_N + \mathbf{B} \end{pmatrix}_{2(N+1) \times 2(N+1)}, \quad \mathbf{B} = \begin{pmatrix} -\rho & 0 & \cdots & 0 & 0 \\ \rho & -\rho & \cdots & 0 & 0 \\ \vdots & \vdots & \ddots & \vdots & \vdots \\ 0 & 0 & \cdots & \rho & -\rho \end{pmatrix}_{N+1 \times N+1}$$

with the initial condition

$$\mathbf{P}(0) = (\underbrace{0, 0, \ldots, 0}_{N+1}, \underbrace{1, 0, \ldots, 0}_{N+1})^T.$$

Secondly, we solve the following inhomogeneous ODE

$$\frac{\mathrm{d}}{\mathrm{d}t} \mathbf{P}(t) = \mathbf{A}\mathbf{P}(t) + h(t - \tau)\left(\mathbf{P}(\tau) - \mathcal{S}\mathbf{P}(\tau)\right), \tag{31}$$

where $\mathcal{S}$ is the right-shift operator, i.e. for any $\mathbf{x} = (x_1, x_2, \ldots, x_N)$, $\mathcal{S}\mathbf{x} = (0, x_1, x_2, \ldots, x_{N-1})$ and

$$h(t) = \begin{cases} \frac{\rho \sigma_{\text{on}}}{\sigma_{\text{off}} + \sigma_{\text{on}}}\left(1 - e^{-(\sigma_{\text{on}} + \sigma_{\text{off}})t}\right) & t \geq 0, \\ 0 & t < 0, \end{cases}$$

with the initial condition

$$\mathbf{P}(0) = (\underbrace{1, 0, \ldots, 0}_{N+1}, \underbrace{0, 0, \ldots, 0}_{N+1})^T.$$

Hence, the solution of *Equation (31)* gives the approximate probability distribution $\mathbf{P}(t)$ for any $t > 0$.

Note that if one uses the generating function *Equation 29* to compute the probability distribution, the computation for the high-order derivatives can cause numerical instabilities due to lack of arithmetic precision – compare the exact solutions with precision 85 and 300 in the left and right panels of *Appendix 6—figure 1*. Because FSP avoids the computation of the higher-order derivatives, it can be as numerically stable as computing the exact solution with high precision (right panel of *Appendix 6—figure 1*). In *Appendix 6—table 1*, we show the computational efficiency of FSP versus numerical evaluation of the exact solution – FSP and the exact solution with high precision are hence comparable both in terms of accuracy and efficiency. In the main text, we use FSP.

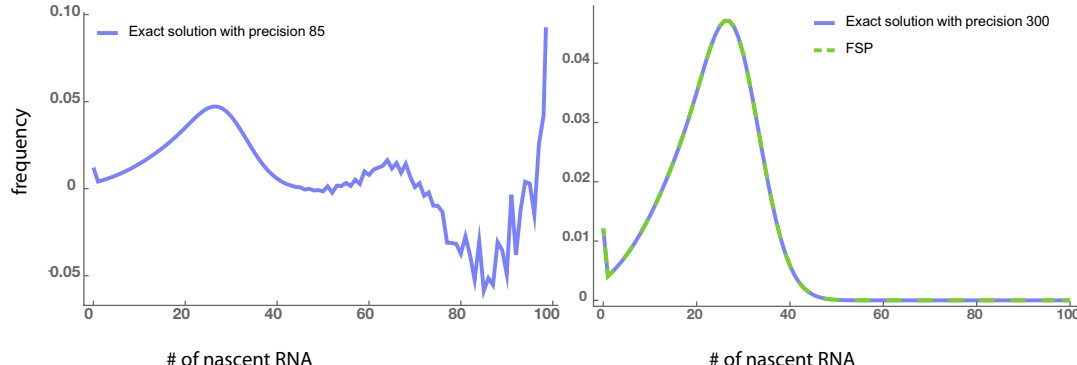

**Appendix 6—figure 1.** Left panel: Numerical instabilities due to the calculation of higher-order derivatives in the exact solution appear when the arithmetic precision is not very high (85). Right panel: These instabilities disappear when the precision increases to 300. The exact solution with such high precision agrees well with the FSP method using double-precision floating-point (Float64) type. The parameters are, $\sigma_{\text{off}} = 1.71$ $\sigma_{\text{on}} = 5.82,, \rho = 53.74$ and. $\tau = 0.56$

**Appendix 6—table 1.** Comparison of the performance of three methods to compute the probability distribution.

Parameters same as in *Appendix 6—figure 1*. The time was calculated using the *Julia* package *BenchmarkTools.jl*.

|  | Exact solution: precision = 85 | Exact solution: precision = 300 | FSP method |
|---|---|---|---|
| Minimum time: | 6.422ms | 9.277ms | 8.317ms |
| Median time: | 6.868ms | 9.562ms | 9.092ms |
| Mean time: | 8.279ms | 11.482ms | 9.415ms |
| Maximum time: | 16.791ms | 16.919ms | 14.203ms |
| # of Simulations: | 604 | 436 | 531 |

# Appendix 7

## Classification of the cell cycle phase: bimodal Gaussian distribution method versus the Fried/Baisch model

The distribution of DNA content intensities (DAPI) was fit using a bimodal distribution; those cells whose intensity was around one peak were classified as G1 and those around the second peak were classified as G2 (main text *Figure 3e*). We also did the classification using an alternative method, based on the Freid/Baisch model, which was recently employed in *Skinner et al., 2016*. In *Appendix 7—figure 1* we show the distributions of fluorescent signal intensity for cells in the G1 phase (top row) and for those in the G2 phase (bottom row) for the four data sets described in the main text. Note that the method (bimodal Gaussian or Freid/Baisch) used to classify the cells in G1/G2 does not alter the intensity distribution hence verifying the robustness of our classification method.

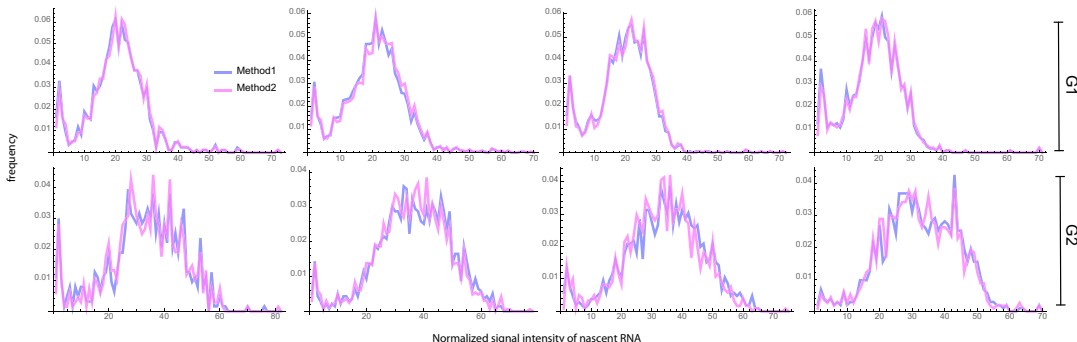

**Appendix 7—figure 1.** Blue curves (Method 1) show the fluorescent intensity distributions of the four experimental data sets after the classification of cells into G1 and G2 phases using the Fried/Baisch model. Magenta curves (Method 2) show the same but using a bimodal Gaussian, as described in the main text.

