## [Editor Report]

This important paper tackles a core problem in systems biology – how to quantify kinetic parameters from incomplete and noisy experimental data. The study makes a convincing methodological contribution to the field, and its potential usefulness is demonstrated using experimental data in yeast.

---

## [Decision Letter]

[Editors' note: this paper was reviewed by Review Commons.]

Thank you for submitting your article "Quantifying how post-transcriptional noise and gene copy number variation bias transcriptional parameter inference from mRNA distributions" for consideration by eLife. Your article has been reviewed by 3 peer reviewers at Review Commons, and the evaluation at eLife has been overseen by Anna Akhmanova as the Senior Editor in consultation with two of the three original reviewers. We apologize for the delay in coming back to you, which was due to the fact that the reviewers have taken some time to respond.

The authors made substantial modifications to the text to accommodate reviewer comments and changed the title to better describe their results. The claims are now easier to assess in the context of the existent literature. Additionally, they also modified figures to more explicitly back their claims. Overall, these changes make the manuscript easier to understand.

However, there are some remaining issues that need to be addressed, as outlined below:

1. A main concern that remains is the discrepancy between nascent and mature mRNAs. The authors argue that this is presumably due to large mRNA turnover rates, which may be expected when the coat protein is not expressed. First, it would be great if the authors could back up this statement with appropriate references. Second, if I take the inferred transcription rates of ~36mRNA/min and the fraction the gene is on ~0.8, this leads to an effective transcription rate of roughly 30mRNA/min (the elongation delay should only lead to a static delay, but not affect the average amount of mature mRNAs that we observe at steady state). Now, in order to obtain an average steady state of mature mRNA around 10 transcripts (e.g., Fig. 3f), I require the mRNA degradation rate to be around 3min^-1. This would give a half-life of roughly log(2)/3=0.2min, which seems to be exceedingly low. According to literature, mRNA half-lives in yeast are typically in the order of 30min and only for very few mRNAs falls below 10mins (see e.g., Geisberg et al., Cell 2014). If we trust this simple calculation above, that would mean that the mRNA half-live is substantially smaller than the lowest reported values that I could find in the literature. I therefore still worry that there might be an issue with the data/spot extraction. At the very least, I would expect such mismatch and its potential origins to be discussed critically in the paper.

2. The second remaining concern relates to the validation against time-series data. I appreciate that the discussion around the calculation of the ACF has been extended. The authors argue that a comparison between statistics obtained from smFISH and live-cell data is not possible due to a potential detection threshold in the former. But why am I then allowed to compare the autocorrelation function (that is derived from calibration against smFISH data) with the autocorrelation of the live-cell data, which also is a statistic of the same transcriptional process? Or conversely, if it was the case that the mRNA mean and variance don't match between experiments, why should we expect the autocorrelation to match? I would be grateful if the authors could clarify these questions. Moreover, I still believe that the term "non-stationary effects" is misleading, since the slowly varying components to the ACF still need to be assumed to be in stationarity in order to capture them by an ACF with just a single time parameter (e.g., the linear fit). Perhaps "slowly-varying" would be a better term.

---

## [Author Response]

We thank all referees for their constructive comments which has helped us to improve the manuscript significantly. Below we provide a point-by-point response.

Reviewer 1:Referee Summary and SignificanceSummary:In this study, the authors consider the problem of inferring transcription dynamics from smFISH data. They distinguish between two important experimental situations. The first one considers measurements of mature mRNAs, while the second one considers measurements of nascent mRNA through fluorescent probes targeting PP7 stem loops. The former problem has been previously dealt with extensively, but less work has been done on the context of the latter. The inference approaches are based on maximum likelihood estimation, from which point estimates for promoter-switching and transcription rates are obtained. The study focuses on steady state measurements only. The authors perform several analyses using synthetic data to understand the limitations of both approaches. They find that inference from nascent mRNA is more reliable than inference from mature mRNA distributions. Moreover, they show that accounting for different cell-cycle stages (G1 vs G2) is important and that pooling measurements across the cell-cycle can lead to quantitatively and even qualitatively different inferences. Both approaches are then used to analyze transcription in an experimental system in yeast, for which they find evidence of gene dosage compensation. I consider this an interesting and relevant study, which will appeal to the systems- and computational biology community. The paper is well written and the (computational) methods are described in detail. The experimental description is quite minimal and could profit from further details / explanations. I have several technical criticisms and questions, which I believe should be addressed before publication. Since I am a theorist, I will comment predominantly on the statistical / computational aspects.Significance:Quantifying kinetic parameters from incomplete and noisy experimental data is a core problem in systems biology. I therefore consider this manuscript to be very relevant to this field. The contribution of this manuscript is largely methodological, although its potential usefulness is demonstrated using experimental data in yeast.

Thank you for the overall positive assessment of our paper. Below, we respond to each comment in detail.

Referee Comment (Major)– A key reference that is missing is Fritzsch et al. Mol Syst Biol (2018). In this work, the authors have used nascent mRNA distributions and autocorrelations (obtained from live-imaging) to infer promoter- and transcription dynamics. I believe this work should be appropriately cited and discussed.

Thank you for pointing out this important reference. We have now added it to the Introduction and the Discussion.

Referee Comment (Major)Synthetic case study:– Inference and point estimates. The authors use a maximum-likelihood framework to extract point estimates of the parameters. Subsequently, relative absolute differences are used to assess the accuracy of the inference. However, as far as I have understood, this is performed for only a single simulated dataset, for each considered parameter configuration. The resulting metric, however, does not really capture the inference accuracy, since it is based on a single (random) realization of the MLE. I would recommend to at least repeat the inference multiple times for different realizations of the simulated dataset (per parameter configuration) to get a better feeling of the distribution of the MLE (e.g., its bias / variance). Alternatively, identifiability analyses based on the Fisher information could be performed for (some of) the different parameter configurations although this may be computationally more demanding.

We thank the reviewer for the suggestion. In SI Section 1.3 we have added numerical experiments to obtain a better understanding of the variability of the inference procedure. For each of the 6 parameter sets in SI Figure 2b, i.e. for the inference using the delay telegraph model, we generated 10 independent sets of synthetic data and used maximum likelihood to infer the parameters for each dataset. The mean and standard deviation of the estimated parameters (computed over all 10 datasets) are shown in SI Table 1. We can define the sample variability of a parameter as the standard deviation divided by the mean. Ordering parameters by this quantity, we find that for all 6 parameter sets the error is largest for σ_off_. For small f_ON_, the parameters ordered by increasing error are: σ_on_, burst size, ρ, f_ON_ and σ_off_. While for large f_ON_, the order is: ρ, f_ON_, σ_on_, burst size and σ_off_. Note that this order is the same as we determined using the relative errors of each of the parameters (from the ground truth) which we have added to Figure 2 of the main text. Note further that the same order is found by using profile likelihood error (SI Section 1.4, SI Tables 2 and 3). Since from experimental data, only the sample variability and the profile likelihood error are available, it follows that the results of our synthetic data study in Figure 2 based on relative error from the ground truth have wide practical applicability. We also note that the means of the parameters (calculated over the parameters inferred from 10 independent synthetic datasets) in SI Table 1 is close to the true parameters (in SI Figure 2b); this shows that the inference procedure is working correctly and the deviations from ground truth values are mostly due to noise in the synthetic datasets.

Referee Comment (Major)– It would be useful to include confidence intervals based on profile likelihoods also for the synthetic case study, in particular for the 6 reported datasets. I would also find it helpful to see comprehensive profile likelihood plots for the key results / parameter inferences in the supplement. This would also provide useful insights into the identifiability of the parameters.

As suggested by the reviewer, we have performed a comprehensive profile likelihood study for the inference on synthetic data and obtained the 95% confidence intervals. We added these to SI Section 1.4. We have also provided confidence intervals for estimates from merged and cell-cycle specific mRNA data in SI Section 3.2 and Table 8; for merged nascent non-curated data in SI Section 4.1 and Table 11; for cell-cycle specific nascent data curated using the fusion method in SI Section 4.2 and Table 14.

Referee Comment (Major)Experimental case study:– Validation against live-cell data. In the simulation of the autocorrelation function, what was the ratio of cells initialized in G1 / G2, respectively? I’d expect this to have direct influence on the simulated ACF. Moreover, a linear fit is used to correct for ”non-stationary effects” in the ACF that supposedly stem from cell-cycle dynamics. First, I don’t think this terminology is really accurate, since non-stationarity would lead to an ACF that depends on two parameters (τ_1_ and τ_2_). I suppose the goal of the linear correction is to remove slow / static population heterogeneity? If yes, wouldn’t it be easier / more direct to also change the simulations to non-synchronized cell-cycles? In this case, they should also display the very slow / static components as displayed in the data, which would eliminate the need for the post-hoc correction. I was also wondering whether other statistics (e.g., mean, variance, distributions) match between the simulations and the live-cell experiment? This could provide further validation of the inferred parameters.

The ratio of G1 to G2 cells was 1.15:1, 1.27:1, 1,24:1 and 0.80:1 for datasets 1, 2, 3 and 4 respectively, as determined from the smFISH experiments. As correctly pointed out by the reviewer, we correct for non-stationary affects in the ACF to remove slow extrinsic heterogeneity, which includes but is not limited to cell cycle. In addition to cell cycle, there are many non-stationary effects, such as bleaching or heterogeneous galactose induction. Since each of these non-stationary effects has its own kinetics and contribution to the ACF, we are limited by post-hoc correction of the ACF. We clarified these points in SI Section 5. Furthermore we note that while in principle, the mean, variance and the distributions of intensities in the live-cell data should be the same as the smFISH data, in reality for the former there is often a detection threshold, and hence the statistics of both distributions cannot be matched directly.

Referee Comment (Major)– If I understood correctly, the signal intensity of the measured transcription spot is normalized by the median cytoplasmic spot brightness. Since the normalized intensity of a single complete transcript is 1, the cumulative intensity should give a lower bound on the nascent mRNAs. The histograms in Figure 4b show intensity values in the range of 30, which would mean that at least 30 transcripts contribute to the transcription spot. The total number of nucleoplasmic and cytoplasmic mRNA, however, is in the range of 10 (Figure 3a). I am probably missing something but how can we reconcile these numbers? The authors mention that the brightest spot just counts for one transcript, but argue that this has negligible influence on mature RNA counts. Could this be a possible explanation for the mismatch?

The difference between mature mRNA count and the nascent transcript count is likely the result of high mRNA turnover. Addition of PP7 loops to the gene is known to affect mRNA stability, when the PP7 coat protein is not expressed. We note that high mRNA turnover should aid in determination of kinetics, as it more closely matches the transcriptional activity. In addition, low mRNA counts allow for better fitting of spots to determine their intensity, in the small volume of the yeast cell. These advantages allow us to make a proper comparison between nascent and mature distributions. We clarified this in the main text (Section 2.2.1). To test the contribution of counting the transcription site as one transcript on the estimated parameters, we now compared two methods, where we either include or exclude the transcription site from the mature RNA distribution. As shown in the main text Figure 3c,d, subtracting the transcription site (shown as seg2-TS) has only a small influence on the fitted parameters (compare to seg2). A discussion can be found in the last paragraph of Section 2.2.1.

In the experimental case study, the authors argue that the ”correct” inference result is the one that accounts for cell-cycle stage, while the other one termed ”incorrect” I find this terminology too strong, since every estimate is subject to uncertainty.We agree and changed to phrasing to ”optimal” or ”matched the live-cell data better”.

Page 2: ”… in a asynchronous population” → ”… in an asynchronous population”

We corrected this typo.

Page 7: ”…parameters sets 3 and 4” → ”…parameter sets 3 and 4”We corrected this typo.

Figures 5a and 6a: parameter names and units should go on the y-axis.

In the new figures, the equivalent of the previous Figures 5a and 6a are Figures 5b and 6a, respectively. Note that we have now changed these to bar graphs.With this format, we have included the parameter names and units on the y-axis and we also added the parameter names on top to make it visually more clear.

In the experimental case study, the authors argue that the ”correct” inference result is the one that accounts for cell-cycle stage, while the other one termed ”incorrect”. I find this terminology too strong, since every estimate is subject to uncertainty.

Reviewer 2:Referee Summary and SignificanceSummary:In the manuscript Fu and co-authors compare accuracy for 2 models that infer kinetics of the transcription from synthetic and experimental data. Specifically, they compare the telegraph model for mRNA and the delayed telegraph model for nascent RNA. They first provide the comparison for synthetically simulated data, and derive that the latter exhibits higher accuracy. Next they apply the model to experimental data from smFISH for PP7-GAL10 strain, and provide the framework to estimate the number of mRNAs and use the intensity at the transcriptional site to infer the number of bounds of polymerase during the transcription (nascent RNA). For the latter, I appreciate that they account for the fact that intensity throughout the transcription will depend on ’spatial’ position of polymerase and incorporate this into the framework to infer nascent RNA levels. Additionally, for the experimental data they infer kinetics with and without accounting for cell cycle (accordingly 1 or 2 gene copies), and through comparing to life imaging data from Donovan et al., 2019, they suggest that the model that best describes experimental data is delayed telegraph for nascent RNA when accounted for cell cycle. Finally they provide 2 approaches – called rejection and fusion – to account for potential artifacts in estimation of nascent RNA levels from the intensity at transcriptional sites, and provide the comparison of how this approaches affect the overall fit.Whereas it is important to have a systematic understanding/comparison for both models as well as for how accounting of cell cycle might improve the overall accuracy, some of the aspects of the results/estimation of values from experimental data require more thorough analysis.Significance:This paper is somewhat outside my core expertise, although closer to the expertise of my postdoc who assisted with the review.The work is interesting but the generalisability of the conclusions is somewhat limited, partially by the lack of experimental validation. Nevertheless, there are interesting aspects of the study and the area of research is important.Overall, the main statements of the paper – that cell cycle specific inference from the experimental data using delayed telegraph model from nascent RNA performs best (compared to telegraph model from mRNA or not cell cycle specific) are supported, and I agree that understanding of the limitations of the currently popular models (telegraph for mRNA and/or not accounting for cell cycle) is an important addition to the field. I would be happy to further proceed with the revision/acceptance of the paper if the (major) comments above are addressed/considered.

Thank you for the overall positive assessment of our paper. Below, we respond to each comment in detail.

Referee Comment (Major)Comparison of the models for simulated data.In the first two chapters of the results the authors compare simulations/parameter inference from the synthetic data for the telegraph-based model for mRNA and delayed telegraph model for nascent RNA, and conclude that the latter provides better accuracy. However, based on the relationship for mean relative error distribution as a function of fON, it seems to me that both models show very similar results, and the support of better accuracy for nascent RNA seems unclear to me. Additionally, simulations are performed for the concise number of parameter sets, and it is unclear how well/uniformly the chosen sets cover the parameter space.I suggest that more thorough analysis is required. One way to do so would be to perform simulations on the same set of parameters that comprehensively cover the parameter space for both models and compare mean error rates in pairwise fashion. Additionally, it might be worth considering comparing error rate for each parameter separately (i.e. for σ-on, σ-off and the production rate of mRNAs when promoter is on).

We agree and we now report a detailed study to take into account all the points raised above. Specifically, we calculate the relative errors (from the true values) for each inferred parameter from 789 synthetic nascent and mature mRNA datasets. Each of these datasets corresponds to an independent parameter set sampled on a grid. We generate parameter sets on an equidistant mesh grid laid over the space:

(*σ_off_, σ_on_, ρ*) ∈ [Uniform(0, 150), Uniform(0, 150), Uniform(0, 250)] , (1)

where the units are inverse minute. Furthermore we apply a constraint on the effective transcription rate ρ^=ρσonσon+σoff<100. In each of the three dimensions of the parameter space, we take 10 points that are equidistant, leading to a total of 1000 parameter sets which reduce to 789 after the effective transcription rate constraint is enforced. The results are presented in Figure 2 of the main text and SI Figure 1. Note that for the telegraph model, besides the usual model, we consider an additional version where we add noise to the initiation rate to model post-transcriptional noise (due to splicing, export, etc) in mature mRNA data that is not present in the nascent mRNA data. In Figure 2b, c and SI Figure 1 we show that the mean relative error is generally smaller for the delay telegraph model (inference from synthetic nascent data) than using the telegraph model (inference using synthetic mature mRNA data with or without post-transcriptional noise). Furthermore in Figure 2d we show plots of the median relative error (for each parameter individually) as a function of the fraction of ON time. Several conclusions can be made including: (i) the errors in σ_on_ (the burst frequency), σ_off_ and the burst size ρ/σ_off_ tend to increase with f_ON_ while for the rest of the parameters (ρ and the estimated value of f_ON_) there is a decreasing tendency; (ii) for small f_ON_, the best estimated parameters are the burst frequency and size while for large f_ON_, it was ρ and the estimated value of f_ON_. The worst estimated parameter was σ_off_, independent of the value of f_ON_; (iii) the addition of noise to mature mRNA data (to model post-transcriptional noise) had a small impact on inference for small f_ON_; in contrast, for large f_ON_ the noise appreciably increased the relative error in σ_off_ and to a lesser extent the error in the other parameters too.

Referee Comment (Major)An additional analysis of the accuracy of the estimated values from the experimental data.

We have now carried such an analysis by (i) calculating the 95% confidence intervals using the profile likelihood method. For merged and cell-cycle specific mRNA data these can be found in SI Section 3.2 and Table 8; for merged nascent non-curated data in SI Section 4.1 and Table 11; for cell-cycle specific nascent data curated using the fusion method in SI Section 4.2 and Table 14; (ii) to test the contribution of counting the transcription site as one transcript on the estimated parameters, we now compared two methods, where we either include or exclude the transcription site from the mature RNA distribution. As shown in the main text Figure 3c,d, subtracting the transcription site (shown as seg2-TS) has only a small influence on the fitted parameters (compare to seg2). A discussion can be found in the last paragraph of Section 2.2.1.

Referee Comment (Major)When it comes to experimental data, the overall fit of any proposed model will depend on both the suitability/correctness of a model to explain the process in question as well as the reliability of the estimates (inputs for the model) from the experiments. Specifically, it is possible that a model (either telegraph for mRNA or delayed telegraph for nascent RNA or both) to explain transcriptional kinetics is fairly accurate, but the input estimates (for accordingly mRNA or nascent RNA) are biased (due to technical artifacts from the experiment and/or the approach towards estimating those values), thus affecting the overall fit of a model and interpretation of the results.I appreciate that authors address one potential artifact in estimating nascent RNA, where it is possible that the intensity of nascent RNA is overestimated if it is mistakenly confused with mRNA. I suggest that the more detailed analysis of the accuracy for both the number of mRNA molecules and the intensity of nascent RNA is required to provide better insight in how reliably those values are estimated and accordingly whether models might perform poorly due to biased estimates.

We thank the reviewer for the good suggestion. Numerical experiments testing the reliability of the inference procedure to random perturbation of mature and nascent mRNA data is now reported in SI Sections 1.5 and 1.6 and the associated SI Tables 4, 5. Specifically for mature mRNA we randomly changed the number by minus 1/plus 1/unchanged with probability 1/3. For nascent mRNA, the perturbation we considered consisted of the addition of lognormal noise to the fluorescent signal. In all cases, we found the inference results were practically unchanged, except for cases where the gene spends a huge amount of time in the OFF state. This is not the case for our experimental data and hence we are confident that the inference results are robust. In the main text we have also added a new section Section 2.2.1 to discuss how various experimental artefacts (overlapping spots, segmentation method, mistaking a transcription site with mature mRNA) may affect the inference – see the response to the next comment.

Referee Comment (Major)Mature mRNA:More detailed method section covering the estimate for background signal and spot detection.A potential proximity of mRNA molecules resulting in underestimation of the total number of mRNAs, and how this might affect the fit of the telegraph model. Even though smFISH has been widely used to estimate the number of mRNA molecules (as a total number of spots), the technique has been mostly applied to mammalian cells with considerably bigger cell size. Additionally, the usage of the total number of mRNA molecules in order to estimate transcriptional kinetics from the telegraph model seemingly requires a highly accurate estimate of the total number of molecules. Combined, it is not obvious if potential underestimation of mRNAs (specifically in cells with high number of mRNAs) via smFISH in budding yeast cells might lead to the misleading interpretation of the results. One way to assess whether such ’merging’ takes place is to look into the distribution of intensities for cytoplasmic spots (per cell and/or all the cells in the whole field of view). If those distributions frequently show bi/multi-modal behavior, it is worth considering whether a proposed way to estimate mRNA number is suitable in for given model organism/growth conditions/gene, and further extend the analysis on simulated data to provide the robustness of the fit of the telegraph model for mRNAs in cases whether number of mRNAs is underestimated.

The reviewer is correct that a highly accurate estimate of the total number of mRNAs is optimal for proper estimates of the transcriptional parameters. We now added an extra section (2.2.1) to describe these possible artifacts and their potential effect on the parameter estimates. Briefly, since cytoplasmic PP7-GAL10 has a high turnover, the number of RNAs allows us to accurately identify all spots, without ”merging” of spots. This is also apparent from the distribution of the intensities of cytoplasmic spots, which follows a unimodal distribution (orange colored distribution in the centre of Figure 4a) where ≈ 90 percent of the detected spots fall within half a standard deviation of the median. Another possible source of error is cell segmentation. To test how cell segmentation errors contribute to the mature mRNA distribution and the error estimates, we included a comparison of two independent segmentation tools. The inferred results are reported in Figure 3b,c,d. Segmentation 1 often results in missed spots, and underestimation of the mRNA count, which reduces the burst size and f_ON_ compared to segmentation 2. Because of the higher accuracy of the latter we used segmentation 2 for the inference from merged and cell-cycle specific mature mRNA data. This analysis further illustrates the challenges associates with parameter estimation from mature mRNA counts, as nascent mRNA data is not affected by the segmentation method.

Referee Comment (Major)A more minor issue, but authors state that, for each cell, the highest intensity of the nuclear spot will count as one mRNA, and that it has a negligible influence. I would appreciate a more thorough analytical explanation for this or an additional analysis on the simulated data to support how random +/-1 of mRNAs might affect results of the fit, specifically for cases with low average mRNA estimate.

We analyzed how removal of the transcription site from the experimental mRNA count affects the inferred parameters. As shown in Figure 3c and d subtracting the transcription site has only a small influence on the estimated parameters. We included a description of this analysis in Section 2.2.1. As mentioned in response to a previous comment, numerical experiments testing the reliability of the inference procedure to random perturbation of mature and nascent mRNA data are now reported in SI Sections 1.5 and 1.6 and the associated SI Tables 4, 5. In all cases, we found the inference results were practically unchanged, except for cases where the gene spends a huge amount of time in the OFF state, which is not the case for our experimental data.

Referee Comment (Major)Nascent RNA:I might be missing something, but it seems that for cells in late G2 phase where nucleus is either strongly elongated (and looks like a sand clock) or even exhibits 2 separate nuclei connected with the chromatin bridge – 2 copies of the gene can be spatially resolved and therefore it might happen that 2 independent/separate brightest spots (one per each cell) amount to total estimate of nascent RNA in cases where promoter is on simultaneously in both copies? If so, depending on estimated in the study/prior literature-based estimates for σ-on/off, the probability of simultaneous transcription might vary and this should be taken into account? This also might partially explain the phenomenon of lower transcriptional activity in G2 which is currently suggested to be explained with dosage compensation? Or are those cells considered as 2 cells in G1? If so, it needs to be specified in the text.

The reviewer is correct that cells in late G2 have elongated nuclei which may contain 2 transcription sites. When the nucleus moves into the bud, buds often contain less DNA than G1 cells, and mothers contain more DNA than G1 cells, both of which are excluded from the analysis. When the DNA content of the mother and daughter is similar, both mother and daughter are counted separately as G1 cells. We note that this late G2 subpopulation is very small. We included a description of this to Methods Section 4.3. For the cells in early G2, the two gene copies cannot be easily resolved in yeast cells since they are within the diffraction limit. In our manuscript, we assumed that these two transcription sites are independent. In SI Section 3.3 and SI Table 9 we show that the relaxation of the assumption of independence between the allele copies in G2 (by instead assuming the opposite case of perfect state correlation of the two alleles) has practically no influence on the inference of the two best estimated parameters (ρ, f_ON_) from mature RNA data (compare the values for phase G2 in Figure 3f and those in SI Table 9). The same is shown for inference from nascent mRNA data in SI Table 15. We note at the end of Section 2 in the main text that while such perfect synchronization of alleles is unlikely, some degree of synchronization is plausible and further improvement of the transcriptional parameters in the G2 phase will require its precise quantification. This is however beyond the scope of the current manuscript since as previously mentioned it is currently difficult to reliably resolve two alleles in yeast cells.

Referee Comment (Major)Additionally, I suggest that images from microscopy can be provided as a supplement to aid clarity in how cell cycle, number of mRNAs and intensity for nascent RNA were estimated.

We have uploaded the data to https://osf.io/d5nvj/. The analysis code of the smFISH microscopy data is available at https://github.com/Lenstralab/smFISH. The code for the synthetic simulations and the parameter inference is available at https://github.com/palmtree2013/RNAInferenceTool.jl. In addition, we adjusted the text and figure to provide clarity on how the data was processed. We included the DAPI intensity distribution in Figure 3e and updated the description in the Methods (section 4.3) to explain how this distribution was used for the selection of G1 and G2 cells. We also included images of the cells in Figure 3a,b and c to show how the number of RNAs was counted and how the segmentation was performed to obtain the mature RNA distributions. Lastly, we added Figure 4a to show how the intensity distribution of the cytoplasmic spots was used to estimate nascent transcripts at the transcription site.

Referee Comment (Major)The analysis of the experimental data consists of the (I presume highly comparable with Donovan et al., 2019) single condition (i.e. galactose concentration, glu/galactose ratio) resulting in a single parameter set for transcriptional kinetics. Specifically, it is estimated that σ on and off will be comparable for the given set up, and therefore, based on simulated data, the estimates will be somewhat reliable for the cell cycle accounted delayed telegraph for nascent RNA. I wonder how in practice (i.e. estimated from the experiments) the same model will perform for a different set of parameters/different conditions. Ideally, I would suggest performing the similar experiment, but where σ on/σ off is expected to be different. One way to achieve this with the GAL10 / galactose set up is to tune the glu/gal ratio of the media. Even without a comparison to live-cell tracing, the analysis of estimated parameters for merged and cell cycle specific data can shed light on how suitable the model is for alternative parameters.Alternatively, if the experiment is currently not feasible, I would appreciate a more extensive discussion of the practical suitability of the cell-cycle specific delayed telegraph model for nascent RNA for alternative sets of transcriptional parameters. Considering that the comparison was performed only against ’simple’ telegraph model and in introduction authors mention a variety of ’improved’ models for mRNA, that account for various sources of heterogeneity, they might be more suitable for alternative set of transcriptional parameters, and might be more suitable that cell cycle specific delayed telegraph for nascent RNA.

Although we appreciate the added value of an extra experimental dataset in different conditions, we question whether this gain outweighs the required time investment. We feel that this questions is already sufficiently addressed by our novel thorough analysis across a large parameters space using synthetic data, described above and in Figure 2, which indicates that inference from nascent transcription distributions performs better for the majority of the parameters space. As suggested by the reviewer, we included a section in the discussion on the suitability of more complex models models. We discuss that these more complex transcription models, although perhaps capturing the kinetics better, they also have many more parameters, meaning that in practice very few will be identifiable with the current set of experimental observables. To use these models, one would thus need the acquisition of temporal data or simultaneous measurement of various sources of extrinsic noise (single cell transcription factor and RNAP number measurements, cellular volume, local cell crowding, etc) which influence gene expression. This is far beyond the scope of the current manuscript, but will be interesting to explore in future studies.

Referee Comments (Minor)Current method section is lacking the description of the growth media, which is an important aspect to specify when it comes to budding yeast (particularly when the sugar source is different from the standard glucose and/or results are compared to another publication).

We agree this should be included and added the growth media (synthetic complete media with 2 % galactose) to the Methods section 4.3

In the figure 2b I find the cartoon a little misleading – specifically why polymerase is bound when the promoter is off? If it is to illustrate the case when transcription/polymerase bound occured after promoter is switched off, why there are no polymerase to the right from the current one (as in in the case where promoter is on)?

We agree and we changed the cartoon (Figure 1c), by showing no Pol II when the promoter is off.

In table1 – there is a typo in the 2nd meta-row – I suspect it should say G2?

This is corrected (SI Table 10).

Reviewer 3:Referee Summary and SignificanceSummary:In this work, the authors investigate the effect of using mature mRNAs instead of only nascent mRNA (located at the transcription site) when estimating transcriptional kinetics parameters from single-molecule fluorescent in situ hybridization (smFISH) experiments. The authors find that using nascent mRNA and correcting for cell cycle effects yields more accurate parameter estimates than using mature mRNAs. The author performs smFISH experiments of the GAL10 gene in yeast to test their findings. Also, the authors test different methods to obtain parameter estimates in cases where there is no information about the location of the transcription site.Significance:As described above, some claims do not seem novel considering the references in this manuscript. This is not a problem; the authors can soften their claims to novelty without compromising their other claims. Previous works that estimated mRNA transcription kinetic parameters by quantifying nascent mRNA recognized that using mature mRNA would incur in parameter estimation errors. They considered it evident that quantifying the process closer to the transcription site would improve estimates. Similarly, it was also apparent that adding missing information (the gene copy number based on cell cycle information) would improve parameter estimates. That is why the authors presenting those arguments as findings is unnecessary. However, it is true that here the authors are interested in the level of error, not the fact that getting more accurate (or relevant) measurement will improve estimates.An item that the authors may want to emphasize is their finding that it is possible to correct for measurements where the identity of the transcription site is unknown. All the works that they cite where nascent mRNA is measured using some method to localize the position of the transcription site. I mammalian cells and fly embryos, it is possible to label introns to identify mRNA located at the transcription site. That is not possible in many yeast genes or other microorganisms.Which audience would be most interested in this work? I think those searching for methods to quantify transcriptional kinetics in organisms where the identity of the transcription site cannot be measured by smFISH or other novel methods such as Cas-FISH.I performed studies of transcriptional kinetics in bacteria during my doctorate, and I continue utilizing smFISH in my research.Overall comment:Overall, I think the current experiments are sufficient to support their claims. Also, the description of methods and references is appropriate to allow other researchers to reproduce their observations. Finally, the experiments are replicated, and enough cells are analyzed to provide enough statistical significance to their claims.

Thank you for the overall positive assessment of our paper. Below, we respond in detail to each comment.

Referee Comment (Major)1. The authors make multiple claims of novelty that conflict with work described in some of their references, particularly: Skinner et al., eLife, 2016; Xu et al., Nature Methods, 2015 and Physical Review Letters, 2016 (References #26,27 and 24 in their manuscript). I could find several instances where the scope of their claims was unclear. Below I describe some cases:a. The title of this paper, “accurate inference of stochastic gene expression from nascent transcript heterogeneity” could also be the summary conclusion of the three works cited above. However, later in the Introduction of the manuscript, the authors state that their goal is to “understand the impact of post-transcriptional noise and cell-to-cell variability on the accuracy of transcriptional parameters inferred from mature mRNA data,” a related yet different topic. I would change the title of the manuscript to reflect their main goal better.b. I would make their claims of novelty more specific. For example, at the end of the abstract, the authors claim that “our novel data curation method yields a quantitatively accurate picture of gene expression.” Quantifying nascent mRNA using smFISH to obtain transcription kinetic parameters has been done before (the references above are an example) also developing the modeling tools to do so (for example, in Xu et al., Physical Review Letters, 2016). What is, exactly, the novelty in their approach? They need to make that explicit or soften their claims.c. In the Introduction, when discussing the effect of the cell cycle in parameter estimation, they write: “Since estimation of all transcriptional parameters (…) from nascent data as a function of the cell cycle phase has not been reported”. However, the work they reference (Skinner et al., eLife, 2016) shows such measurements for multiple transcriptional parameters for different cell cycle stages. The original work may not have gone as far as the current work, but it is unclear what has been done before from the way the authors describe earlier literature.d. The authors develop a new formulation of the delay telegraph model to obtain kinetic parameters from the nascent RNA copy number statistics. They state in the SI that “Similar delay models have also been studied by other authors,” however, the authors do not explain in which way their model differs from previous work. Does their approach have advantages over previously published models?

We have now clarified the claims of novelty, as follows.

a. We agree with the reviewer. We have now changed the title of the manuscript to “Quantifying how post-transcriptional noise and gene copy number variation bias transcriptional parameter inference from mRNA distributions”.

b. We agree and have made our claims of novelty more specific and have changed parts of the abstract and introduction. We now describe in the introduction that previous work by Xu et al., PRL 2016 and Skinner et al., *eLife* 2016 develop methods to estimate parameters from nascent mRNA data where the transcription site is localized, but have not considered cases where the transcription sites are unknown; in addition, a quantitative comparison between parameters estimated from mature and nascent data is also not considered because from their point of view it is evident that the estimates using nascent mRNA data should be more accurate. Nevertheless, it could be the case that the estimates using nascent data are only marginally better than those from mature data. In addition, we have removed the sentence “Our novel data curation method yields a quantitatively accurate picture of gene expression”. Furthermore, in the abstract we have mentioned that while it is presumed in the literature that post-transcriptional noise and gene copy number variation may affect parameter estimation, however the size of the errors incurred is presently unclear and this is what we focus on. Lastly, we explicitly mention in the abstract that the fusion method corrects for measurement noise due to the uncertainty in the localization of the transcription site when introns cannot be labelled. We have also added a paragraph to the Introduction (the one before the last) clarifying what other studies before us have done and what remains to be addressed.

c. We agree that from our description, although technically correct, it was unclear what has been done before by Skinner et al. and what is novel. In Skinner et al., *eLife* (2016) (Supplementary File 2 Table B ), all parameters are reported, but it was assumed that the burst frequency is the only parameter that changed upon replication, i.e. not all parameters were simultaneously estimated pre and post-replication. In our manuscript, we estimate all parameters without this assumption. In the Introduction we have now changed the sentence to “The estimation of transcriptional parameters from nascent mRNA data for pre- and post-replication phases of the cell cycle has, to the best of our knowledge, only been reported in Skinner et al. *eLife* (2016)”. In the Introduction, we also point out that in this reference (and others) there is no direct comparison between estimates from nascent and mature mRNA data. In Section 2.2.2, we have made the differences between our method and that in Skinner et al. more explicit by stating: “A difference between our method of estimating parameters in G2 from that in the literature (Skinner et al) is that we do not assume that the burst frequency is the only parameter that changes upon replication, i.e. we estimate all transcription parameters simultaneously.”.

d. Our algorithm and that in Xu et al. PRL (2016) both take into account the fact that the signal intensity depends on the position of Pol II on the gene, albeit this is done in different ways. In Xu et al. PRL (2016), a master equation is written for the joint distribution of gene state and the number of nascent mRNA. In this case the number of nascent RNAs can have non-integer values since it represents the experimentally measured signal from the (incomplete) nascent RNA. Solution of this master equation proceeds by (a) a discretization of the continuous nascent mRNA signal into bins which are much smaller than one; (b) solution using finite state projection (FSP). This approach can lead to a large state space which incurs a large computational cost. In contrast in our method, we use FSP to solve for the delay telegraph model, i.e. the distribution of the discrete number of bound Pol II from which we construct (using convolution) the approximate distribution of the continuous nascent mRNA signal by assuming Pol II is uniformly distributed on the gene. Since the state space of bound Pol II is typically not large, our method will typically be more computationally efficient than the one described in Xu et al. PRL (2016). We note that in Skinner, et al. *eLife* (2016) there is also mentioned that they used a uniform distribution assumption for the Pol II positions but we cannot find any details of the algorithm in the paper to compare directly to ours. However, note that our paper is not about a novel inference algorithm but about the systematic study of how post-transcriptional and gene copy number noise affect the accuracy of parameter inference – this is the main novelty. We have added a discussion of the above points to the Methods Section 4.2.1.

Referee Comment (Major)2. There is a particular choice during their analysis that I find problematic. In section 2.3, the authors state “The transcription site is counted as 1 mRNA, regardless of its intensity, but has a negligible influence since the mean number of mature mRNA is much greater than 1” (the number should be spelled). It is unclear that statement is true for all possible kinetic parameters. It is also hard to evaluate that claim because the authors do not show images of transcription sites that would support it. Trying to find more information, I saw images from previous work from one of the authors (“Optimized protocol for single-molecule RNA FISH to visualize gene expression in *S. cerevisiae*”, figure 4). Those images suggest that the opposite is the case: in the cell shown, the number of mRNAs in the transcription site is not negligible but instead seems to contain most of the mRNAs in the cell. Solving this problem would require the authors to remake their analysis without making this assumption.

The reviewer is correct that the number of nascent transcripts is substantial. As these transcripts are not mature, their count should however not contribute to the mature transcript distribution. We previously counted this transcription site as one mRNA. As also described in the reviewer comments to reviewers 1 and 2, we now analyzed how removal of the transcription site from the experimental mRNA count affects the inferred parameters. As shown in Figure 3c and d subtracting the transcription site has only a small influence on the estimated parameters (compare seg 2 and seg2-TS). See also SI Table 7 and SI Figure 4a. We included a description of this analysis in Section 2.2.1. To see if this insensitivity to TS inclusion is true for all parameter sets, we used synthetic data to randomly change the number of mature mRNA by minus 1/plus 1/unchanged with probability 1/3. In all cases, we found the inference results were practically unchanged, except for cases where the gene spends a huge amount of time in the OFF state. The results are described in SI Section 1.5 and SI Table 4

Referee Comments (Minor)1. In section 2.1.3, the authors mention using an optimization package written in Julia programing language. A reference to the package needs to be included, either an academic article or the website to the package.

We now added the webpage for BlackBoxOptim.jl https://github.com/robertfeldt/BlackBoxOptim.jl in Methods Section 4.1.3.

2. In the discussion, the authors state “In addition, live-cell measurements include cells in S phase, which are excluded in smFISH.” I do not think that statement is correct. One would expect that a large enough sample of cells assayed with smFISH will contain a subpopulation containing cells in the S-phase.

The reviewer is correct. smFISH contains cells in S phase, which are the cells in Figure 3e with a DNA content in between the G1 and G2 population. In this manuscript, we do not analyze this S phase population. They are however measured in live-cell imaging. We updated the text to “which are not analyzed in smFISH”.

3. I find the overall presentation of figures and the analysis performed not optimal to convey their points. Below are some suggestions regarding presentation (and in some cases, analysis).– Text suggestions:a. The meaning of the word “inference” seems to change across the manuscript. In the title, I understand that inference means “estimation,” or more explicitly, estimating model parameters from experimental or simulated data. However, in the methods section, the authors write “Mature mRNA inference” and “Nascent mRNA inference.” Do they mean “Estimating/Inferring model parameters from synthetic/experimental mature/nascent mRNA datasets”?

To avoid confusion and to be clearer, we have changed the titles of the subsections in Methods. With the word “inference” we mean the inference of parameters from mRNA data. The algorithms for inference from mature and nascent data in Methods are the same for synthetic and experimental data.

b. In the Introduction, the authors use three different terms for cell cycle (cell cycle position, cell cycle stage, and cell cycle phase). It is unclear to me if they are referring to the same concept.We have now used the phrase “cell cycle phase” throughout the manuscript.– Presentation suggestions:c. I would remove Figure 2C and put it in the Supplementary information. It shows procedure details that are not fundamental to understanding their claims.

We appreciate the suggestion but we have left the new version of Figure 2c (presently Figure 1d) in the main text. The reason is that the study from synthetic data is still part of the story in the main text and we believe the figure aids its comprehension.

d. I would also relegate the tables in their six datasets in figure 1 and 2 to the Supplementary material. Tables are not very effective methods to present information.

We agree and moved these tables to the supplement. We have also replaced all relevant tables in the manuscript with bargraphs.

e. I do not think that figures 1c and 2d are needed. Comparing the results from stochastic simulations and the predictions from the models is an internal control that the researchers should do to test the accuracy of their SSA implementation; it does not convey a message related to the main conclusions of their work.

We agree and moved these figures to the supplement.

f. I like figure 4a; it conveys one of the main points: not correcting for cell cycle can lead to considerable errors in parameter estimation. **I would like to see a similar plot that conveys the difference in parameter estimation when using nascent vs. mature mRNA.**

We agree this comparison would be very interesting. We note however that for nascent versus mature, a direct comparison, parameter by parameter, is not possible. This is because using the telegraph model it is only possible to estimate the switching rates and the initiation rate scaled by the degradation rate, and the latter is is unknown. On the other hand, the estimates from nascent data are rates multiplied by the elongation time – the latter is known and hence the absolute rates can be estimated from nascent mRNA data only. The only quantities that can be directly compared are the burst size and the fraction of ON time, since these are both non-dimensional. We now included bar charts in Figure 3f and 4c to compare these two and we comment on the comparison in Section 2.3.1.

g. Why do the authors have table 1 separated from figure 4 while adding the tables to figures 1 and 2? I would be consistent and move all tables to the supplementary material.

We have considerably reorganised the table and figure order in the manuscript. Most tables are now in the SI. However, we have left some of the tables in the main text, i.e. Figure 3f for mature mRNA data and Figure 6d for nascent mRNA data corrected for the fusion method, since these enable an easy way to assess the impact of transcriptional noise and cell-cycle phase on inferred values.

[Editors' note: further revisions were suggested prior to acceptance, as described below.]

However, there are some remaining issues that need to be addressed, as outlined below:1. A main concern that remains is the discrepancy between nascent and mature mRNAs. The authors argue that this is presumably due to large mRNA turnover rates, which may be expected when the coat protein is not expressed. First, it would be great if the authors could back up this statement with appropriate references. Second, if I take the inferred transcription rates of ~36mRNA/min and the fraction the gene is on ~0.8, this leads to an effective transcription rate of roughly 30mRNA/min (the elongation delay should only lead to a static delay, but not affect the average amount of mature mRNAs that we observe at steady state). Now, in order to obtain an average steady state of mature mRNA around 10 transcripts (e.g., Fig. 3f), I require the mRNA degradation rate to be around 3min^-1. This would give a half-life of roughly log(2)/3=0.2min, which seems to be exceedingly low. According to literature, mRNA half-lives in yeast are typically in the order of 30min and only for very few mRNAs falls below 10mins (see e.g., Geisberg et al., Cell 2014). If we trust this simple calculation above, that would mean that the mRNA half-live is substantially smaller than the lowest reported values that I could find in the literature. I therefore still worry that there might be an issue with the data/spot extraction. At the very least, I would expect such mismatch and its potential origins to be discussed critically in the paper.

We agree that the half-life of PP7-tagged GAL10 is very fast compared to other endogenous mRNA. We have added a discussion on this in the results section, as well as added the reference suggested by the reviewer: “In fact, in our experiments, the number of detected mature mRNA transcripts per cell was lower than expected based on the number of nascent transcripts (compare Fig. 3 with Fig. 4). This discrepancy between nascent and mature transcripts likely arises because the addition of the PP7 loops to the GAL10 RNA results in a very fast mRNA turnover, which is much faster than the turnover of most endogenous RNAs [1, 2, 3, 4]. Previously, both shorter and longer mRNA half-lives from the addition of stem loops have been observed, which may be caused because changes in the 5’ UTR length or sequence changes its recognition by the mRNA degradation machinery [5, 6, 7].”

2. The second remaining concern relates to the validation against time-series data. I appreciate that the discussion around the calculation of the ACF has been extended. The authors argue that a comparison between statistics obtained from smFISH and live-cell data is not possible due to a potential detection threshold in the former. But why am I then allowed to compare the autocorrelation function (that is derived from calibration against smFISH data) with the autocorrelation of the live-cell data, which also is a statistic of the same transcriptional process? Or conversely, if it was the case that the mRNA mean and variance don't match between experiments, why should we expect the autocorrelation to match? I would be grateful if the authors could clarify these questions. Moreover, I still believe that the term "non-stationary effects" is misleading, since the slowly varying components to the ACF still need to be assumed to be in stationarity in order to capture them by an ACF with just a single time parameter (e.g., the linear fit). Perhaps "slowly-varying" would be a better term.

We apologize that we misunderstood the reviewer previously. The reason that we cannot directly compare the mean and variance of the live-cell data with the estimated distribution derived from the smFISH, is because of cell-to-cell variation in the intensities of the live-cell data. Specifically, the live-cell traces showed cell-to-cell variation in overall fluorescent intensity of each trace arising from differences in the PP7 coat protein expression level that was expressed from a plasmid. This means that the scaling factor of how much intensity represents a single RNA varies between cells, but the traces are not long enough to obtain this scaling factor (or the full intensity distribution) for each cell. The normalized ACF however, normalizes the intensity per trace, and thus allows comparison between the kinetics even if the absolute intensity differs per cell. We now added a statement to the manuscript explaining why we use the normalized ACF for comparison rather than the intensity distribution: ”Specifically, the live-cell traces showed cell-to-cell variation in overall fluorescent intensity arising from differences in the PP7 coat protein expression level, precluding a direct comparison of the live-cell intensities with the smFISH distributions. The normalized ACFs are normalized per trace and thus can be used to directly compare the kinetics.”

We agree that the term ”non-stationary effects” does not accurately encompass all corrected slowvarying effects, such a bleaching. We note that we also use the same linear subtraction method to correct for the non-stationary fast induction from raffinose to galactose in the live-cell data, which in our hands is well-described by a linear long time component in the ACF. We have adapted our description and nomenclature in the supplemental methods (last sentence of Section 5) to make this more accurate.

References

[1] Miller, C. et al. Dynamic transcriptome analysis measures rates of mrna synthesis and decay in yeast. Molecular systems biology 7, 458 (2011).

[2] Wang, Y. et al. Precision and functional specificity in mrna decay. Proceedings of the National Academy of Sciences 99, 5860–5865 (2002).

[3] Holstege, F. C. et al. Dissecting the regulatory circuitry of a eukaryotic genome. Cell 95, 717–728 (1998).

[4] Geisberg, J. V., Moqtaderi, Z., Fan, X., Ozsolak, F. & Struhl, K. Global analysis of mrna isoform half-lives reveals stabilizing and destabilizing elements in yeast. Cell 156, 812–824 (2014).

[5] Heinrich, S., Sidler, C. L., Azzalin, C. M. & Weis, K. Stem–loop rna labeling can affect nuclear and cytoplasmic mrna processing. Rna 23, 134–141 (2017).

[6] Tutucci, E. et al. An improved ms2 system for accurate reporting of the mrna life cycle. Nature methods 15, 81–89 (2018).

[7] Garcia, J. F. & Parker, R. Ms2 coat proteins bound to yeast mrnas block 5 to 3 degradation and trap mrna decay products: implications for the localization of mrnas by ms2-mcp system. Rna 21, 1393–1395 (2015).